# SIMBA: single-cell embedding along with features

Huidong Chen [1,2,3], Jayoung Ryu [1,2,4], Michael E. Vinyard [1,2,3,5], Adam Lerer [6] ✉ & Luca Pinello [1,2,3] ✉

Most current single-cell analysis pipelines are limited to cell embeddings and rely heavily on clustering, while lacking the ability to explicitly model interactions between different feature types. Furthermore, these methods are tailored to specific tasks, as distinct single-cell problems are formulated differently. To address these shortcomings, here we present SIMBA, a graph embedding method that jointly embeds single cells and their defining features, such as genes, chromatin-accessible regions and DNA sequences, into a common latent space. By leveraging the co-embedding of cells and features, SIMBA allows for the study of cellular heterogeneity, clustering-free marker discovery, gene regulation inference, batch effect removal and omics data integration. We show that SIMBA provides a single framework that allows diverse single-cell problems to be formulated in a unified way and thus simplifies the development of new analyses and extension to new single-cell modalities. SIMBA is implemented as a comprehensive Python library (https://simba-bio.readthedocs.io).

Recent advances in single-cell omics technologies have enabled the individual and joint profiling of cellular measurements. The emergence of single-cell multi-omics technologies allows for the measurements of multiple cellular layers, including genomics, epi-genomics, transcriptomics and proteomics. Such assays have greatly enhanced our ability to understand cell states and the molecular machinery behind development and disease. Despite the potential of these technologies, computational challenges remain in fully harnessing their capabilities.

Many single-cell computational methods have been developed for the analysis of one modality (for example, single-cell RNA sequencing (scRNA-seq) or single-cell sequencing assay for transposase-accessible chromatin (scATAC-seq))[1–4]. Common to these methods is a workflow that includes routine steps such as feature selection, dimension reduction, clustering and differential feature detection. These 'cluster-centric' analysis methods rely on accurately defined clustering solutions to discover meaningful marker features. Unfortunately, clustering solutions may range widely within the space of the user-defined clustering resolution (number of clusters) and the chosen clustering

algorithm, potentially leading to inconsistent and inaccurate biological annotations[5]. Although initial efforts have been made recently to develop clustering-free approaches to discover informative genes, they are specifically designed for extracting gene signatures[6,7] or identifying perturbations between experimental conditions[8] from scRNA-seq data, and are therefore limited to single-modality and single-task analysis.

Computational methods have also been proposed for multi-batch and cross-modality analysis, such as multimodal analysis (distinct cellular parameters are measured in the same cell)[9], batch correction (the same cellular parameter is measured in different batches)[10–12] and integration of multi-omics datasets (distinct cellular parameters are measured in different cells)[11,12]. However, these tasks require development of new approaches owing to differences in task formulation. Also, most current methods cannot exploit relations between multiple cellular features directly. Furthermore, these methods for identifying marker features rely on clustering and therefore are limited to clustering solutions. Additionally, instead of directly identifying marker features in the integrated space, current state-of-the-art batch correction

[1]Molecular Pathology Unit, Center for Cancer Research, Massachusetts General Hospital, Boston, MA, USA. [2]Department of Pathology, Harvard Medical School, Boston, MA, USA. [3]Broad Institute of Harvard and MIT, Cambridge, MA, USA. [4]Department of Biomedical Informatics, Harvard Medical School, Boston, MA, USA. [5]Department of Chemistry and Chemical Biology, Harvard University, Cambridge, MA, USA. [6]Facebook AI Research, New York, NY, USA. ✉e-mail: adam.lerer@gmail.com; lpinello@mgh.harvard.edu

and multi-omics integration methods[10–12] need to first detect marker features in the uncorrected or unintegrated original space of each batch or modality independently and then combine them, thus resulting in potentially inconsistent interpretations between batches or modalities.

To overcome these limitations, we propose SIMBA (single-cell embedding along with features), a versatile single-cell embedding method that co-embeds cells and features such as genes, peaks and DNA sequences into a common latent space, allowing for the execution of various tasks in a unified manner. Unlike existing methods that require featurization of cells, SIMBA directly encodes the cell–feature or feature–feature relations into a large multi-relation (or heterogenous, that is, multiple node and edge types) graph. For each task, SIMBA constructs a graph, wherein differing entities (that is, cells and features) are represented as nodes, and relations between these entities are encoded as edges. Once the graph is constructed, SIMBA then applies a multi-relation graph embedding algorithm derived from social-networking technologies[13,14], as well as a Softmax-based transformation to embed the nodes or entities into a common low-dimensional space wherein cells and features can be analyzed on the basis of their distance. Hence the SIMBA embedding space containing cells and all the features can be viewed as an informative database of entities. Depending on the task, biological queries can be defined on the 'SIMBA database' by considering neighboring entities of either a cell (or cells) or a feature (or features) at the individual-cell and individual-feature level (Methods). For example, the query for a cell's neighboring features can be used to identify marker features (for example, marker genes or peaks) or to study the interaction between features (for example, peak-gene), while the query for features' neighboring cells can be used to annotate cells. This is fundamentally different from recently proposed single-cell embedding methods (Supplementary Note 1 and Supplementary Table 1).

SIMBA can solve various single-cell tasks in a unified framework, including: (1) dimensionality reduction; (2) clustering-free marker detection; (3) multimodal analysis; and (4) batch correction and omics integration. SIMBA can be adapted to these diverse tasks by simply modifying the input graph constructed from the single-cell data. SIMBA has been extensively tested on multiple scRNA-seq, scATAC-seq and dual-omics datasets, outperforming or performing comparably to current state-of-the-art methods developed specifically for each task.

Importantly, we have developed a scalable and comprehensive Python package that enables seamless interaction between graph construction, training with PyTorch for graph embedding and post-training analysis. SIMBA is a self-contained framework; however, it is also compatible with popular single-cell analysis tools, such as Scanpy[2]. SIMBA with detailed documentation and tutorials is available at https://simba-bio.readthedocs.io.

## Results

### Overview of SIMBA

SIMBA is a single-cell embedding method that supports single- or multi-modality analyses. It leverages recent graph embedding techniques[13,14] to embed cells and genomic features into a shared latent space. Unlike existing methods that primarily focus on learning cell states, SIMBA treats both cells and features as nodes in the same graph and thus solves various single-cell tasks through a unified procedure. Importantly, SIMBA introduces several crucial procedures, including Softmax transformation, weight decay for controlling overfitting and entity-type constraints to generate comparable embeddings (co-embeddings) of cells and features and to address unique challenges in single-cell data.

SIMBA first encodes different types of entities, such as cells, genes, open chromatin regions (peaks or bins), transcription factor (TF) motifs and k-mers (short sequences of a specific length, k), into a single graph (Fig. 1 and Methods), in which each node represents an individual entity and edges indicate relations between entities. In SIMBA, edges

may be added in two ways by being (1) measured experimentally or (2) inferred computationally. For edges that are measured experimentally, each cell–feature edge corresponds to a single-cell measurement. For example, if a gene is expressed in a cell, an edge is created between the gene and cell. The weight of this edge is determined by the gene expression level. Similarly, an edge is added between a cell and a chromatin region if the region is open in this cell. Edges are also allowed between different features to capture and model the underlying regulatory mechanisms. For example, an edge between a chromatin region and a TF motif (or k-mer) captures the notion that a TF may bind to a regulatory region containing a specific DNA sequence. Edges that cannot be directly measured are inferred computationally by summarizing features of the same or different types (Methods). Each edge between cells of different batches or modalities indicates the cellular functional or structural similarity. Figure 1 summarizes potential relations represented by edges and the semantics for the analyses presented in this study, namely: (cell–gene), a cell expresses a given gene; (cell–peak), a cell has an accessible chromatin region; (peak–TF motif), a peak sequence contains a putative binding site for a given TF; (peak–k-mer), a peak sequence contains a given k-mer sequence; and (cell–cell), cells of different batches or modalities are functionally or structurally similar.

Once the input graph is constructed, SIMBA computes a low-dimensional representation of graph nodes using an unsupervised graph embedding method, leveraging the PyTorch-BigGraph framework[14] that can scale to millions of cells. The resulting joint embedding of cells and features not only reconstructs the heterogeneity of cells, but also allows for the discovery of the defining features for each cell without relying on clustering, separating cell-type-specific (informative) features from the non-cell-type-specific (non-informative) features. The proximity in the SIMBA embedding reflects edge probability, which is the likelihood of an edge existing in the graph and is informative of feature importance or the interplay between features (Methods). Cell-type-specific features, such as marker genes and cis-regulatory elements, can be discovered without clustering in two ways. With known cell labels, marker features can be identified as neighboring features of cells through biological queries (Methods). Without known labels, marker features can be identified through calculating the imbalance of edge probabilities between a feature and all cells using metrics such as the Gini index (Methods).

Importantly, graph construction is inherently flexible, enabling SIMBA to be applied to a wide variety of single-cell tasks. In the following sections, we demonstrate the application of SIMBA to several popular single-cell tasks, including scRNA-seq, scATAC-seq, multimodal analysis, batch correction and multi-omics integration (Fig. 1).

### Single-cell RNA-seq analysis with SIMBA

scRNA-seq is the most widely used analysis for profiling single cells. Figure 2a provides an illustrative overview of the SIMBA graph construction and the resulting low-dimensional embedding matrix of both cells and genes in scRNA-seq analysis. SIMBA discretizes the normalized gene expression matrix into multiple levels (five levels, by default). The input graph is then constructed by connecting cells and genes through weighted edges on the basis of gene expression levels. SIMBA then generates embeddings of these nodes through a graph embedding procedure (Fig. 2a and Methods). Depending on the task, we have the full flexibility to visualize either the whole SIMBA embeddings (embeddings of cells and all genes in Supplementary Fig. 1c) or the partial SIMBA embeddings (embeddings of cells in Fig. 2b, or embeddings of cells and variable genes in Fig. 2c, or embeddings of any entities of interest) using UMAP.

We applied SIMBA to a popular peripheral blood mononuclear cells (PBMCs) dataset from 10x Genomics (Supplementary Table 2). SIMBA embeddings of cells showed clear separation of eight cell types, including B cells, megakaryocytes, CD14 monocytes, FCGR3A monocytes, dendritic cells, NK cells, CD4 T cells and CD8 T cells (Fig. 2b). SIMBA

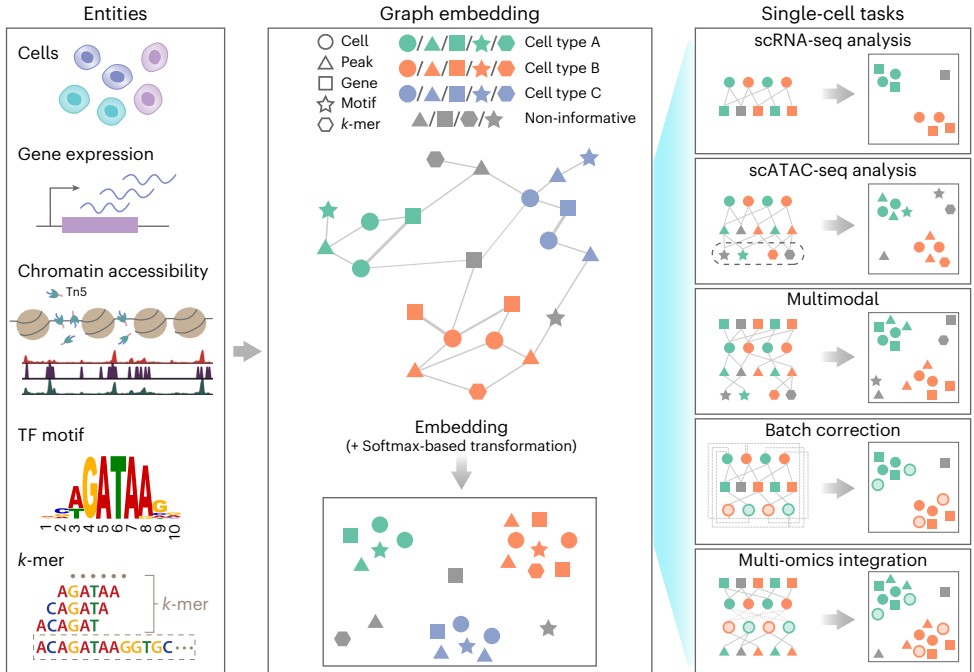

**Fig. 1 | SIMBA framework overview.** SIMBA co-embeds cells and various features measured during single-cell experiments into a shared latent space to accomplish both common tasks involved in single-cell data analysis and tasks that remain as open problems in single-cell genomics. Left, examples of possible biological entities that may be encoded by SIMBA, including cells, gene expression measurements, chromatin-accessible regions, TF motifs and k-mer sequences found in reads. Middle, SIMBA embedding plot with multiple types of entities into a low-dimensional space. All entities represented as shapes (cell, circle; peak, triangle; gene, square; TF motif, star; k-mer, hexagon) are colored by relevant cell type (green, orange and blue in this example). Non-informative features are colored dark gray. Within the graph, each entity is a node, and an edge indicates a relation between entities (for example, a gene is expressed in a cell, a chromatin region is accessible in a cell, or a TF motif or k-mer is present within an open chromatin region.). Once connected in a graph, these entities may be embedded into a shared low-dimensional space, with cell-type-specific entities embedded in the same neighborhood and non-informative features embedded elsewhere. Right, common single-cell analysis tasks that may be accomplished using SIMBA. Different opacity levels indicate cells of different experimental batches or single-cell modalities. Solid lines indicate experimentally measured edges. Dashed lines indicate computationally inferred edges.

embeddings of both cells and genes correctly recovered cellular heterogeneity and embedded informative genes close to relevant cell types (Fig. 2c). Previous marker genes used to annotate cells[2] were highlighted on the UMAP plot, showing that SIMBA not only accurately embedded major-cell-group-specific genes to the correct locations (for example, *IL7R*[15] was embedded into CD4 T cells, and *MS4A1* was embedded into B cells), but also was robust to rare-cell-group-specific genes (for example, *PPBP* was embedded into megakaryocytes), while non-informative housekeeping genes, such as *GAPDH* and *B2M*, were embedded in the middle of all cell groups (Fig. 2c and Supplementary Fig. 1c).

These highlighted genes can be further confirmed with 'barcode plots', which visualize the estimated probability of assigning a feature to a cell by SIMBA on the basis of the recovered edge confidence (Fig. 2d, Supplementary Fig. 1d and Methods). An imbalance in probability indicates the association of a gene to a subpopulation of cells (often corresponding to known cell types), whereas a uniform probability distribution indicates a non-cell-type-specific gene. For marker genes (*CST3* for monocytes and dendritic cells, *MS4A1* for B cells and *NGK7* for NK and CD8 T cells), we observed a clear excess in the probability of assigning each gene to their respective cell types. Conversely, for the housekeeping gene *GAPDH*, we observed a more uniform distribution with much lower probability of associating that gene with the top-ranked cells.

SIMBA also provides several quantitative metrics (termed 'SIMBA metrics'), including max value, Gini index, standard deviation (s.d.) and entropy, to assess cell-type specificity of various features without requiring the prior knowledge such as predefined cell types (Methods and Supplementary Figs. 1b and 3a). By inspecting the gene metric plot of max value versus Gini index (a higher value indicates higher cell-type specificity), we observed that the marker genes (for example, *CST3*, *NKG7*, *MS4A1*) fall in the upper right corner, as opposed to housekeeping genes (for example, *GAPDH*), which falls in the lower left corner (Fig. 2e). Cell-type specificity of the selected marker genes was further confirmed by visualizing their expression pattern on UMAP plots, along with SIMBA barcode plots (Fig. 2f and Supplementary Figs. 1d and 2) and quantitative validation (Supplementary Note 2 and Supplementary Fig. 5a). SIMBA metrics not only rank features on the basis of their cell-type specificity, but also filter out non-informative features to simplify the visualization of embeddings of cells and informative features, preventing the SIMBA space from being crowded with non-informative features.

We show that SIMBA does not require variable gene selection, an essential step in standard scRNA-seq pipelines such as Seurat or Scanpy. When tested with or without variable gene selection, SIMBA produced qualitatively similar embeddings (Fig. 2b and Supplementary Fig. 4e). However, we do observe that variable gene selection improves the efficiency of the training procedure. We also compared SIMBA with both clustering-dependent[2] and clustering-free[6,7] methods in marker gene detection (Supplementary Note 3 and Supplementary Figs. 4 and 6). The computational complexity of SIMBA software was also benchmarked against Scanpy and Seurat (Supplementary Note 4).

## Single-cell ATAC-seq analysis with SIMBA
scATAC-seq has been widely used to profile regions of open chromatin and identify functional *cis*-regulatory elements, such as enhancers

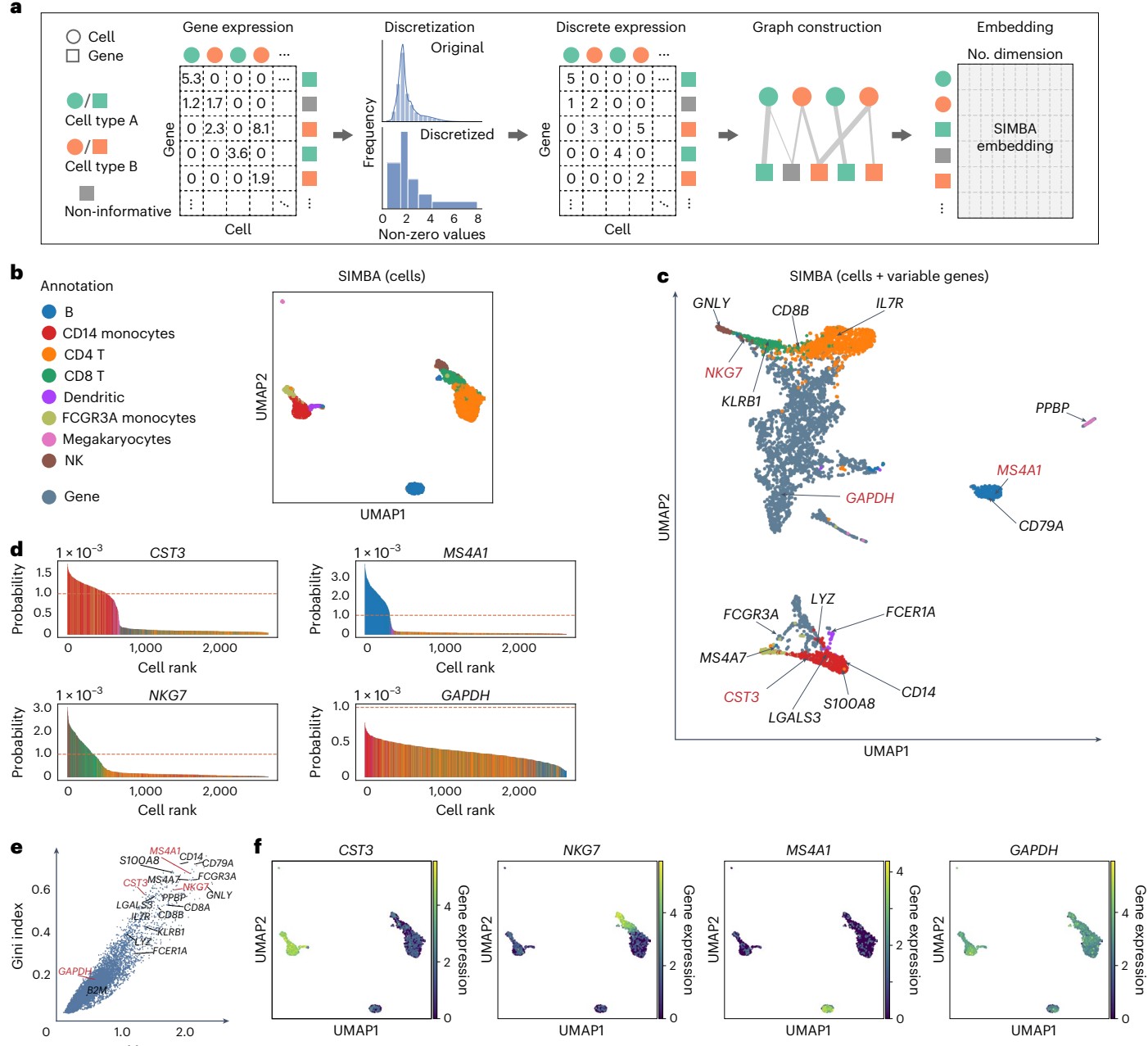

**Fig. 2 | Single-cell RNA-seq analysis of the 10x Genomics PBMCs dataset using SIMBA. a**, SIMBA graph construction and embedding in scRNA-seq analysis. Biological entities including cells and genes are represented as shapes and colored by relevant cell types (green and orange). Non-informative genes are colored dark gray. Gene expression measurements for each cell are organized into a cell-by-gene matrix. These normalized non-negative observed values undergo discretization into five gene expression levels. Cells and genes are then assembled into a graph with nodes representing cells and genes, and edges between them representing different gene expression levels. This graph may then be embedded into a lower-dimensional space resulting in a no. entities × no. dimension (by default, 50) SIMBA embedding matrix. **b**, UMAP visualization of SIMBA embeddings of cells colored by cell type. **c**, UMAP visualization of SIMBA embeddings of cells and variable genes. Cells are colored according to cell type,

as defined in **b**. Genes are colored slate blue. Cell-type-specific marker genes and housekeeping genes recovered by Scanpy are indicated with texts and arrows. Genes highlighted in red are shown in **d**, **e** and **f**. **d**, SIMBA barcode plots of genes *CST3*, *MS4A1*, *NKG7* and *GAPDH*. The x axis indicates the ordering of a cell as ranked by the probability for each cell to be associated with a given gene. The y axis describes the probability. The sum of probability over all cells is equal to 1. Each cell is one bar and colored according to cell type as defined in **b**. The orange dashed line indicates the probability score of $1 \times 10^{-3}$. **e**, SIMBA ranking of genes on the basis of the proposed metrics. All the genes are plotted according to the Gini index against max score. The same set of genes as in **c** are annotated. **f**, UMAP visualization of SIMBA embeddings of cells, colored by gene expression of (left to right): *CST3*, *NKG7*, *MS4A1* and *GAPDH*.

and active promoters. Cells are characterized by different types of features[16], such as accessible chromatin regions ('peaks' or 'bins') and *cis*-regulatory elements (DNA sequences) including TF motifs or *k*-mers. Unlike existing methods that can use only peaks or bins or

DNA sequences, SIMBA can leverage either single or multiple types of features to learn cell states because of its flexibility in graph construction. Also, as SIMBA encodes cell–feature or feature–feature relations into the graph on the basis of the simple binary presence of

a feature, it does not need additional normalization steps, such as term frequency-inverse document frequency (TF-IDF), which is required by most scATAC-seq analyses[16] (Fig. 3a). Through the embedding procedure, SIMBA generates embeddings of cells along with peaks and DNA sequences (Methods). Finally, either the partial SIMBA embeddings (embeddings of cells in Fig. 3b) or the whole SIMBA embeddings (embeddings of cells and all the features in Fig. 3c) can be visualized.

We applied SIMBA to a scATAC-seq dataset of 2,034 human hematopoietic cells with fluorescence-activated cell sorting (FACS)-characterized cell types[17] (Supplementary Table 2). For embeddings of cells alone, as shown in Fig. 3b, SIMBA accurately separated cells such that cells belonging to distinct cell types (defined on the basis of FACS labels) are visually distinguished. For embeddings of cells together with various types of features, as shown in Fig. 3c, SIMBA successfully embedded distinct features from both positional (peaks/bins) and sequence-content (TF motifs and k-mers) information together on the basis of their biological relations. Notably, on the basis of SIMBA metrics, these highlighted features embedded within each cell type all have high cell-type specificity scores (shown in the upper right part of SIMBA metric plots in Fig. 3d and Supplementary Fig. 3b).

Our analysis using SIMBA led to several key findings in human hematopoietic differentiation.

First, SIMBA identified key master regulators of hematopoiesis. As highlighted in Fig. 3c, we observed that motifs of previously reported TFs were embedded near their respective cell types in the UMAP plot. For example, the GATA1 and GATA3 motifs are proximal to megakaryocyte-erythroid progenitor (MEP) cells[18], the PAX5 and EBF1 motifs are near to common lymphoid progenitor (CLP) cells[19], and the CEBPB and CEBPD motifs are proximal to monocyte (mono) population[20].

Second, SIMBA revealed an unbiased set of DNA sequences, that is, k-mers, that represent important TF-binding motifs involved in hematopoiesis. We observed that these k-mers were embedded near their matching TF-binding motifs and relevant cell subpopulations (Fig. 3c,e, Supplementary Fig. 7b and Supplementary Note 5), indicating that SIMBA is capable of de novo motif discovery. For example, the DNA sequence GATAAG is embedded in MEPs; this sequence matches the binding motif of GATA1, the master regulator in erythropoiesis. We also calculated TF/k-mer activity scores[21] (high-variance TF motifs/k-mers) and visualized them on SIMBA embeddings of cells (Fig. 3f and Supplementary Fig. 7a,b). For example, the GATA1 TF motif and k-mer GATAAG, both of which were embedded in MEP cells by SIMBA, also showed high-level activity in MEP cells.

Third, SIMBA identified differentially accessible chromatin regions that may mediate cell-type-specific gene regulation (Supplementary Fig. 7c). For example, the two peaks near the genomic locus of the KLF1 gene, with coordinates chr19:12997999-12998154 (P1) and chr19:12998329-12998592 (P2), were embedded within MEP cells and were observed almost exclusively in MEP cells (Fig. 3e). Interestingly, P1, upstream of KLF1, contains the k-mer GATAAG, which matches the GATA1 binding motif, while the TF GATA1 is known to regulate the gene KLF1 and plays a pivotal role in erythroid cell and megakaryocyte development[22]. Therefore, by embedding these MEP-cell-related regulatory elements into the neighborhood of MEP cells, SIMBA demonstrates a novel means of studying the epigenetic landscape of cell differentiation.

Although SIMBA diverges from current scATAC-seq analysis methods by enabling the co-embedding of cells and features, we still qualitatively and quantitatively compared SIMBA embeddings of cells to state-of-the-art scATAC-seq analysis methods by their ability to distinguish cell types. Our analyses show that SIMBA, overall, outperforms current methods for scATAC-seq analysis, further demonstrating the wide utility of SIMBA (Supplementary Fig. 10 and Supplementary Note 5). We also show that there is negligible impact on the embeddings of cells generated by SIMBA upon the inclusion of sequences as additional features (Supplementary Figs. 9 and 27 and Supplementary Note 5).

## Single-cell multimodal analysis with SIMBA

Recently developed single-cell dual-omics technologies[23–26] can jointly profile transcriptome and chromatin accessibility within the same cells, providing a means to explore gene regulation principles. SIMBA is capable of learning cell heterogeneity and gene regulatory circuits from single-cell multi-omics data. Figure 4a depicts the graph construction and SIMBA embedding procedure. A gene expression matrix and chromatin accessibility (peaks), TF motif and k-mer match matrices are discretized and binarized, respectively, to construct a graph by creating edges between five entity (node) types, including cells, genes, peaks, TF motifs and k-mers. The graph embedding procedure generates SIMBA embeddings of cells and features. To avoid non-informative peaks dominating the space (Supplementary Fig. 11a,c), we leverage the flexibility of SIMBA embedding to visualize only the partial SIMBA embeddings to improve the visibility of cells and cell-type-specific features.

To demonstrate the versatility of SIMBA embeddings, we analyzed the cell populations undergoing hair follicle differentiation from mouse skin profiled with SHARE-seq[24]. First, we calculated SIMBA metrics (max values and Gini index scores) to assess the cell-type specificity of different types of features, including genes, TF motifs and peaks (Fig. 4b and Methods). As shown in Fig. 4b, we successfully recovered genes associated with hair follicles, such as Lef1 and Hoxc13. Similarly, TF motifs and peaks proximal to the genomic loci of these genes also score in the upper-right quadrant of the metric plots.

Next, we visualized and interrogated SIMBA embeddings of (1) cells; (2) cells and top-ranked genes on the basis of SIMBA metrics; and (3) cells, top-ranked genes and TF motifs on the basis of SIMBA metrics, and their neighboring peaks (Fig. 4c). SIMBA embeddings of cells revealed the three fate decisions from transit-amplifying cells (TACs), including inner root sheath (IRS), medulla and cuticle and cortex. SIMBA embeddings of cells and informative features uncovered important genes and regulatory factors along the hair-follicle differentiation trajectories. For example, marker genes Krt71, Krt31 and Foxq1 were embedded into their corresponding cell types: IRS, cuticle/cortex and medulla, respectively. Regulatory factors, such as Lef1 and Hoxc13, were embedded into the beginning and late stages of cuticle/cortex differentiation, respectively. Peaks near the Lef1 and Hoxc13 loci were also embedded into the nearby regions of these genes and motifs. The TF-motif distance could indicate a lag between TF expression and its binding activity. For example, pioneer factors can bind to inaccessible regions, assisting in opening them for other factors. In Fig. 4c, the Hoxc13 gene appears earlier than its motif, consistent with a previous study showing that Hoxc13 has the ability to bind inaccessible motifs[27]. The reported marker genes and TF motifs were also supported by UMAP and SIMBA barcode plots, with high probability towards the correct cell type labels (Supplementary Fig. 12a–d). We also performed scRNA-seq and scATAC-seq single-modality analyses within the SHARE-seq dataset and achieved consistent embedding results with multimodal analysis, demonstrating that the SIMBA embedding procedure is robust to the type and the number of features encoded in the input graph (Supplementary Fig. 11b,c).

Further, we demonstrated that the SIMBA co-embedding space of cells and features provides the potential to identify master regulators and infer their target regulatory genes. SIMBA successfully identified previously described master regulators, such as Lef1, Gata6, Nfatc1 and Hoxc13, as the top master regulators related to lineage commitment in mouse skin (Fig. 4d, Supplementary Table 3 and Methods). Moreover, SIMBA identified a previously unreported master regulator Relb, and a novel Relb+ cell subpopulation, within TAC-2 cells (Supplementary Fig. 28 and Supplementary Note 6). To infer the target genes of a given master regulator, we postulate that, in the shared SIMBA embedding space, (1) the target gene is close to both the TF motif and the TF gene; and (2) the accessible regions (peaks) near the target gene loci must be close to both the TF motif and the target TF gene. Resting on these assumptions of cis-regulatory dynamics

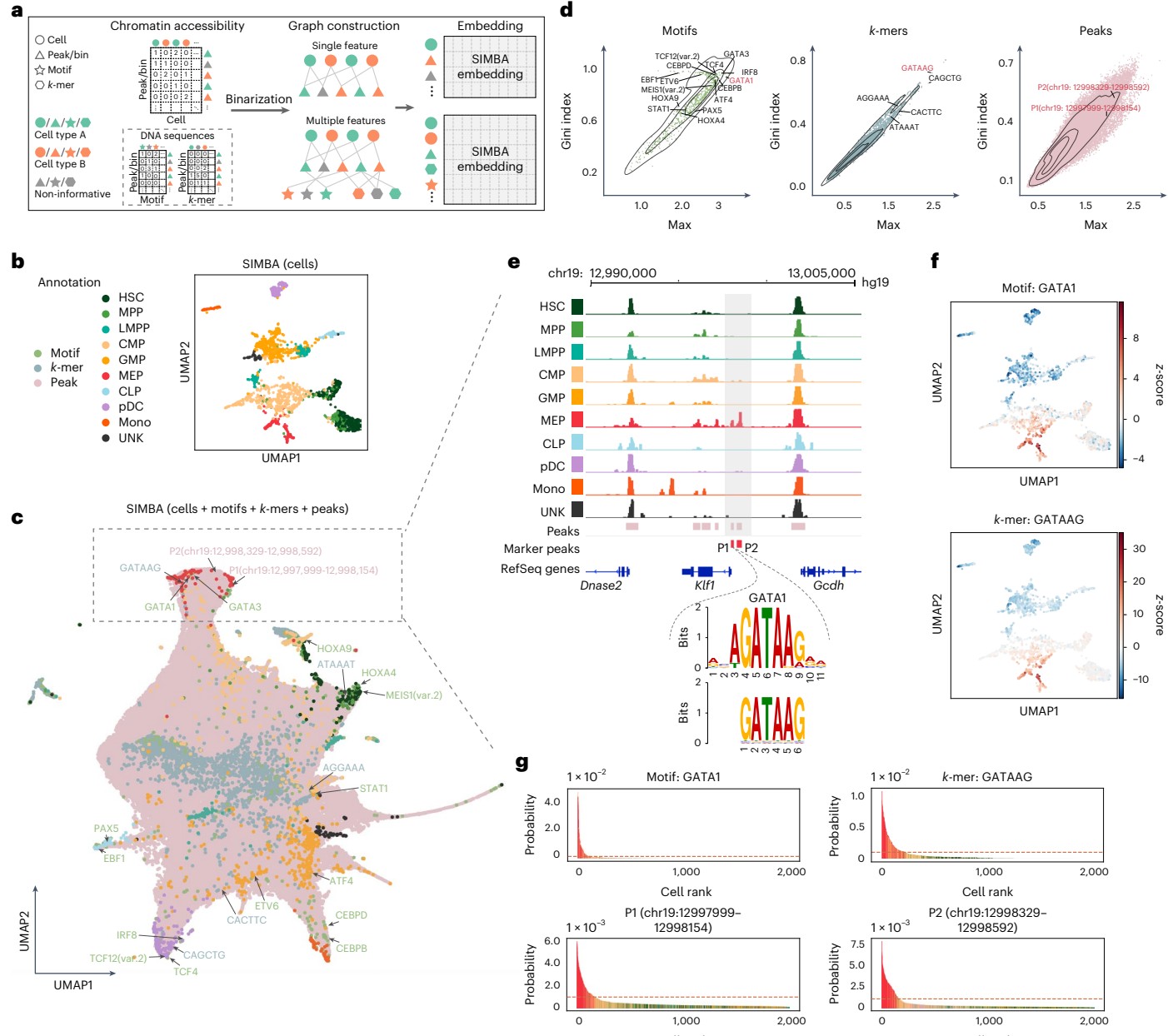

**Fig. 3 | Single-cell ATAC-seq analysis of the human hematopoiesis dataset using SIMBA. a**, SIMBA graph construction and embedding in scATAC-seq analysis. Biological entities including cells, peaks or bins, TF motifs and *k*-mers are represented as shapes and colored by relevant cell types (green and orange). Non-informative features are colored dark gray. Cells and chromatin-accessible features (peaks/bins) are organized into a cell × peak/bin matrix. When sequence information (TF motif or *k*-mer sequence) within these regions is available, they can be organized into two sub-matrices to associate a TF motif or *k*-mer sequence with each peak or bin. These constructed feature matrices are then binarized and assembled into a graph. When a single feature (chromatin accessibility) is used, the graph encodes cells and peaks/bins as nodes. When multiple features (both chromatin accessibility and DNA sequences) are used, this graph may then be extended with the addition of TF motifs and *k*-mer sequences as nodes. Finally, SIMBA embeddings of these entities are generated through a graph embedding procedure. **b**, UMAP visualization of SIMBA embeddings of cells colored by cell type. **c**, UMAP visualization of SIMBA embeddings of cells and features including TF motifs, *k*-mers and peaks. Cells are colored by cell type, while motifs, *k*-mers and peaks are colored green, blue and pink, respectively. Cell-type-specific features that are embedded near their corresponding cell types are indicated as the text labels (colored according to feature type) with arrows. **d**, SIMBA metric plots of TF motifs, *k*-mers and peaks. Cell-type-specific features annotated in **c** are highlighted. **e**, Genomic tracks of aligned scATAC-seq fragments, separated and colored by cell type. Two marker peaks P1 and P2 in red are shown beneath the alignment. Within the peak P1, *k*-mer GATAAG and its resembling GATA1 motif logo are highlighted. **f**, UMAP visualization of SIMBA embeddings of cells colored by TF activity scores of the GATA1 motif and *k*-mer GATAAG enrichment. **g**, SIMBA barcode plots of the GATA1 motif, the *k*-mer GATAAG and the two peaks P1 and P2. Cells are colored according to cell type labels described above. The dashed red line indicates the same cutoff used in all four plots.

(Fig. 4e and Methods), SIMBA inferred target genes of master regulators, such as Lef1 and Hoxc13 (Fig. 4f, Supplementary Fig. 12e and Supplementary Table 4). Notably, SIMBA recovered target genes reported in the original study[24], including genes *Lef1*, *Jag1*, *Hoxc13* and *Gtf2ird1*, regulated by the TF Lef1, and genes *Cybrd1*, *Hoxc13* and *St14*, regulated by the TF Hoxc13.

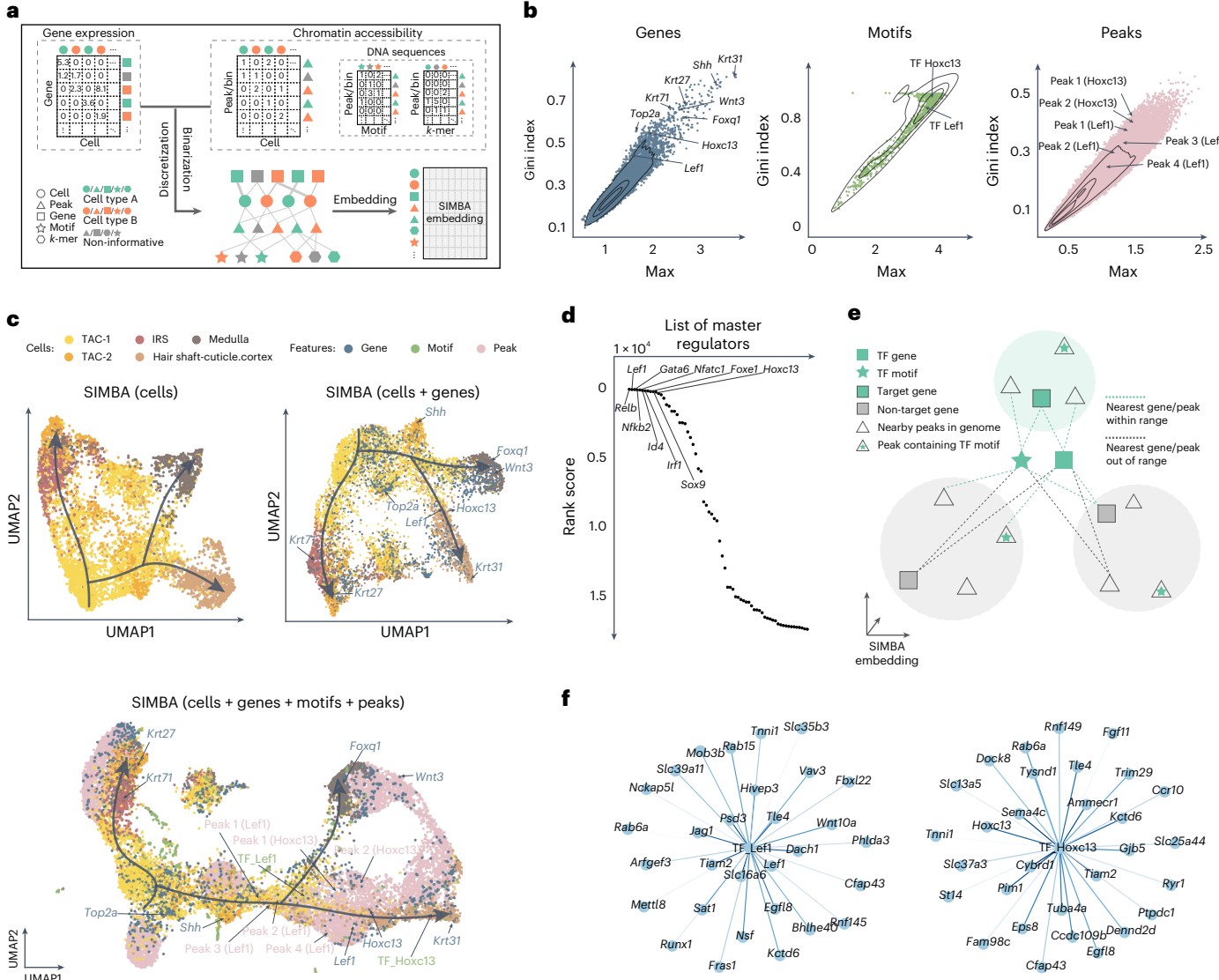

**Fig. 4 | Multimodal analysis of the SHARE-seq hair follicle dataset using SIMBA. a**, SIMBA graph construction and embedding in multimodal analysis. Overview of SIMBA's approach to multimodal (scRNA-seq + scATAC-seq) data analysis. **b**, SIMBA metric plots of genes, TF motifs and peaks. All these features are plotted according to the Gini index against max score. Cell-type-specific genes, TF motifs and peaks are highlighted. **c**, UMAP visualization of SIMBA embeddings of cells (top left), cells and genes (top right) and cells along with genes, TF motifs and peaks (bottom). **d**, Ranked master regulators identified by SIMBA. The rank score indicates the rank of a given TF gene among all genes (including non-TFs)

on the basis of the distance from the genes to the selected TF motif in the SIMBA embedding space. **e**, Schematic of SIMBA's strategy for identifying target genes given a master regulator in the high-dimensional SIMBA embedding space (by default, 50 dimensions). The master regulator is represented by a TF motif (a star filled in green) and a TF gene (a square without borders filled in green). A colored shade indicates a SIMBA embedding area of a gene, containing a gene (a square with borders) and peaks (triangles) near the genomic locus of the gene. Green shades indicate areas of target genes, and gray shades indicate areas of non-target genes. **f**, Top 30 target genes of TFs Lef1 and Hoxc13, as inferred by SIMBA.

In addition to SHARE-seq, we also applied SIMBA to another two dual-omics datasets (Supplementary Table 2), the mouse cerebral cortex dataset profiled by SNARE-seq[23] (Supplementary Figs. 13 and 14) and the multiome PBMCs dataset from 10x Genomics (Supplementary Figs. 15 and 16). By validating the embeddings of cells and features with given cell type labels, marker genes from the original study and differentially accessible chromatin regions, we further demonstrate the suitability of SIMBA for multimodal analysis.

### Single-cell batch correction with SIMBA
As single-cell data collection expands across multiple institutions, there is an increased demand for analysis methods that can account for technical covariates. Batch correction is essential to remove technical variation while preserving biological signals. However, existing

methods are dependent on clustering, and the detection of markers is prone to inconsistencies when combining genes identified from the uncorrected space of each batch. By contrast, SIMBA generates embeddings of both cells and genes, enabling batch-effect removal and marker gene detection in an integrated space without clustering.

SIMBA accomplishes batch correction by encoding multiple scRNA-seq datasets into a single graph (Fig. 5a). Cell nodes across batches are connected to the shared gene nodes through experimentally measured edges, as in scRNA-seq graph construction. Batch correction is further enhanced through computationally inferred edges drawn between similar cell nodes across datasets using a truncated randomized singular value decomposition (SVD)-based procedure. From the resulting graph, SIMBA generates batch-corrected embeddings of cells and genes, allowing for individual-cell-level marker detection

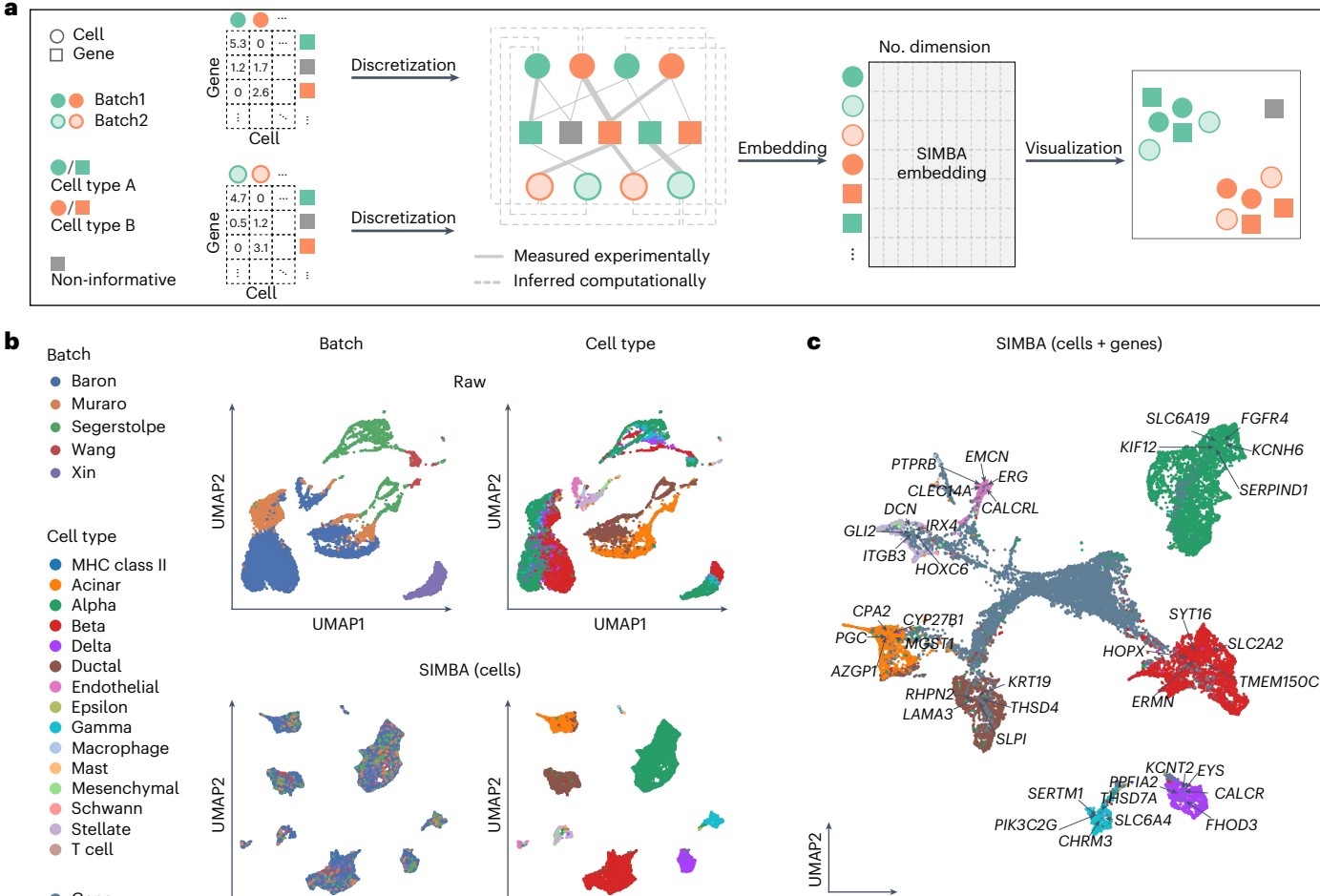

**Fig. 5 | Batch correction analysis of scRNA-seq data using SIMBA. a**, SIMBA graph construction and embedding in batch correction analysis. Overview of SIMBA's approach to batch correction across scRNA-seq datasets. Distinct shapes indicate the type of entity (cell or gene). Colors distinguish batches or cell types. **b**, UMAP visualization of the scRNA-seq human pancreas dataset, with five batches of different studies before and after batch correction. Cells are colored by scRNA-seq data source and cell type, respectively. Top, UMAP visualization before batch correction. Bottom, UMAP visualization after batch correction with SIMBA. **c**, UMAP visualization of SIMBA embeddings of cells and genes, with batch effect removed and known marker genes highlighted.

through biological queries of cells in the shared latent space (Methods). We visualized both SIMBA embeddings of cells (Fig. 5b) and the whole embeddings of cells and genes (Fig. 5c) in UMAP.

We applied SIMBA to two multi-batch scRNA-seq datasets: a mouse atlas dataset composed of two batches, and a human pancreas dataset spanning five batches used in a recent benchmark study[28] (Supplementary Table 2). The mouse atlas dataset contains two scRNA-seq datasets with shared cell types from different sequencing platforms. The human pancreas dataset contains five samples pooled from five sources using four sequencing techniques, in which not all cell types are shared across each sample. For both datasets, SIMBA successfully corrected batch effects, evenly mixing batches within annotated cell-type clusters, while maintaining the segregation of these clusters in the resulting embedding, indicating preservation of biological signal and elimination of confounding technical covariates (Fig. 5b and Supplementary Fig. 19b). It is important to note that the mouse atlas dataset was collected from nine organ systems, so there is some expected heterogeneity within cell-type labels. Conversely, the human pancreas datasets are curated from a single organ, and SIMBA sufficiently separated cell types into transcriptionally distinct, homogeneous cell clusters (Fig. 5b).

SIMBA not only removes batch effects during graph embedding, but also simultaneously identifies cell-type-specific marker genes

(Fig. 5c). Marker genes can be identifiable by performing biological queries for neighboring genes within cell types in the batch-corrected SIMBA space (Methods). In the case of unknown cell labels, marker genes can be identified by calculating SIMBA metrics. SIMBA correctly embeds known cell-type-specific marker genes proximal to the correct cell-type labels, while non-marker genes were non-proximal to specifically labeled cells (Supplementary Figs. 17 and 18). The resulting marker genes recapitulated the clustering-based differential expression (DE) analysis results for each dataset[29–34] (for example, *Cdh5*, *Tie1* and *Myct1* for endothelial cell, *C1qc* and *Fcgr1* for macrophage and *S100a8* and *Trem3* for neutrophil in the mouse atlas dataset; *KIF12* for alpha cell and *KRT19* for ductal cell in the human pancreas dataset) and are expressed specifically in the queried cell types (Supplementary Figs. 17 and 18).

Although SIMBA is a versatile graph embedding method, we evaluated SIMBA embeddings of cells for this task with methods that were specifically designed for batch correction. We considered three widely adopted batch correction methods that demonstrated top-tier performance based on a recent benchmark study[28]: Seurat3 (ref.[12]), LIGER[11] and Harmony[10]. Our results indicate that SIMBA achieved comparable batch correction performance, both qualitatively and quantitatively, while enabling simultaneous marker-gene detection by providing additional embeddings of genes (Supplementary Note 7 and Supplementary Fig. 19).

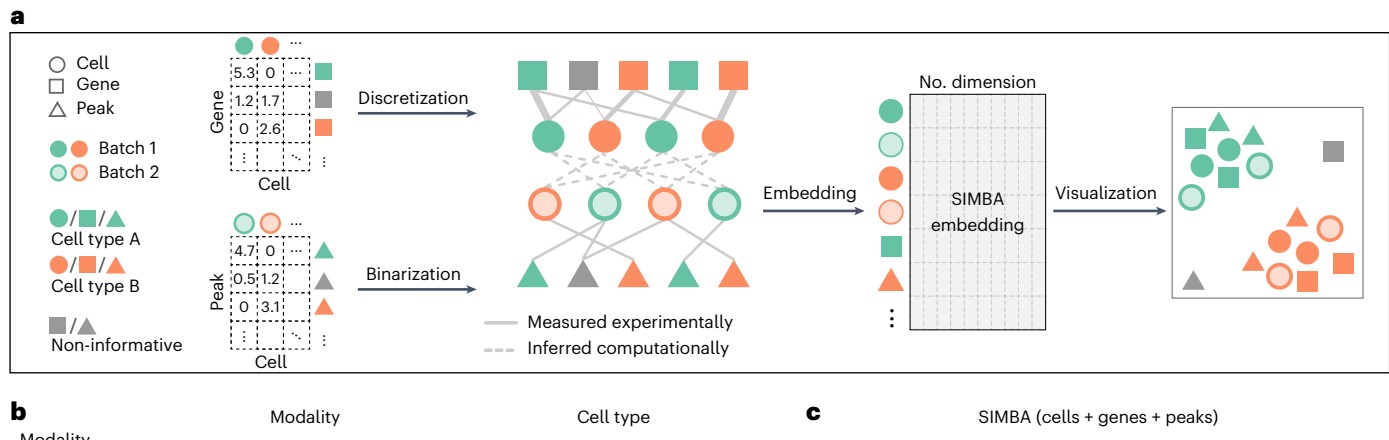

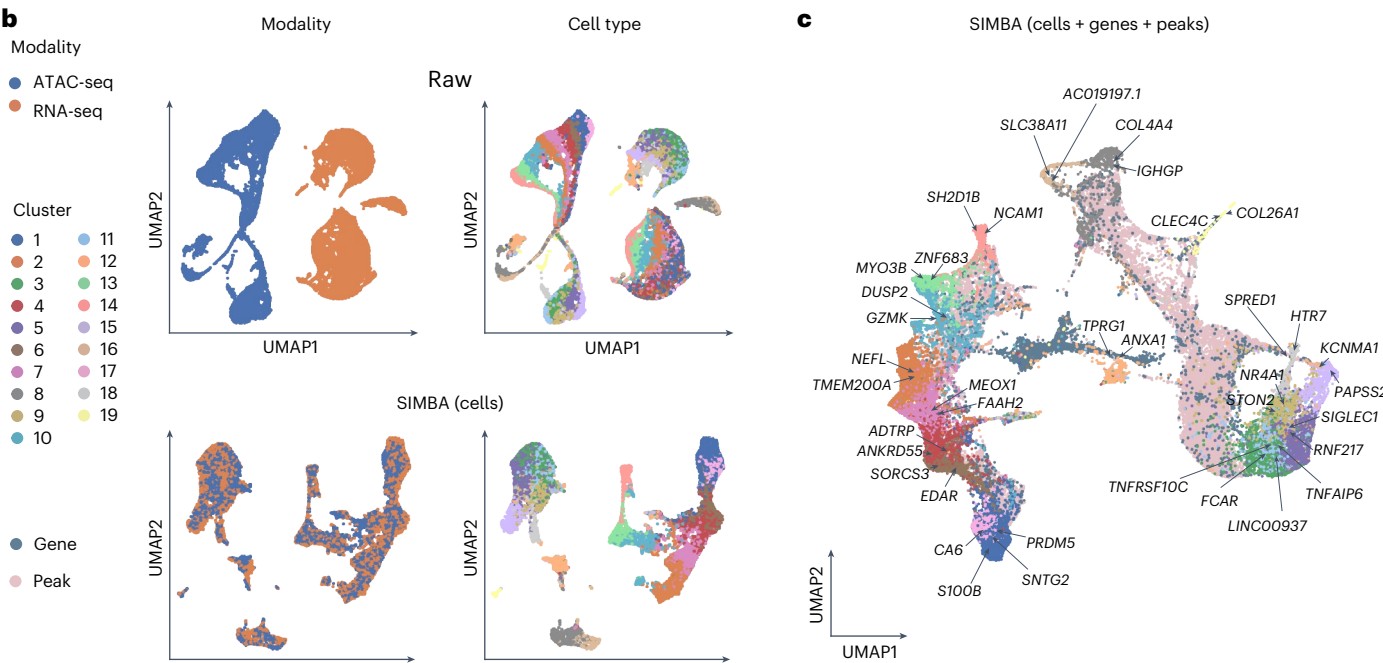

**Fig. 6 | Multi-omics integration of scRNA-seq and scATAC-seq data using SIMBA. a**, SIMBA graph construction and embedding in multi-omics integration. Overview of SIMBA's approach to data integration across scRNA-seq and scATAC-seq. Distinct shapes indicate the type of entity (cell, gene or peak). Colors distinguish batches or cell types. **b**, UMAP visualization of the integrated scRNA-seq and scATAC-seq data manually created from the 10x human PBMCs dataset before and after data integration. Cells are colored by single-cell modality and cell type, respectively. Top, UMAP visualization before integration. Bottom, UMAP visualization after integration with SIMBA. **c**, UMAP visualization of SIMBA embeddings of cells, genes and peaks, with two cell modalities integrated and known marker genes highlighted.

## Single-cell multi-omics integration with SIMBA

Single-cell assays can now measure a broad range of cellular modalities, requiring methods that leverage these features and integrate multi-omics data to study cell state comprehensively. Current multi-omics integration methods follow a workflow similar to batch correction. Unlike existing methods, SIMBA can explore multi-type features directly in the integrated SIMBA space and detect distinct marker features without clustering, enabling simultaneous multi-omics integration and clustering-free marker feature detection, specifically when applied to scRNA-seq and scATAC-seq datasets.

SIMBA builds independent graphs for scRNA-seq and scATAC-seq data, connects them through computationally inferred edges on the basis of shared gene expression modules and embeds the graph of cells, genes and peaks into a low-dimensional space to represent the integrated space of multiple modalities (Fig. 6a and Methods). This enables individual-cell-level marker detection of multi-type features by performing biological queries of cells in SIMBA space. The embeddings

of these multi-omics entities can be visualized either partially or in their entirety using UMAP.

To facilitate the evaluation of data integration performance, we created datasets with ground-truth labels by manually splitting the dual-omics datasets into two single-modality datasets (that is, scRNA-seq and scATAC-seq), in which we know the true matching between cells across the two modalities. We then applied SIMBA to the integration analysis of two case studies where scRNA-seq and scATAC-seq datasets are generated from the SHARE-seq mouse skin dataset and the 10x Genomics multiome human PBMCs dataset, respectively (Supplementary Table 2).

We first visualized SIMBA embeddings of cells and observed that SIMBA was able to preserve cellular heterogeneity while evenly mixing the two modalities (Fig. 6b and Supplementary Fig. 21b). We then visualized SIMBA embeddings of cells, genes and top-ranked peaks on the basis of SIMBA metrics and observed that, in addition to learning cellular heterogeneity, SIMBA simultaneously identified marker genes and peaks at single-cell resolution (Fig. 6c and Supplementary

Fig. 21). In the co-embedding space, we observed that the neighbor genes of cells (highlighted in UMAP plots) are each exclusively expressed in their corresponding cell types (Supplementary Figs. 21a–e and 22a–c,e). For example, in the SHARE-seq mouse skin dataset, *Foxq1* and *Shh* are located within medulla and TAC-2, respectively; in the 10x PBMCs dataset, *PAPSS2* and *KCNMA1*, which are the marker genes of blood monocytes, are embedded close to each other. Similarly, we observed that the neighbor peaks of cells show a clear cell-type-specific accessibility pattern that is robust to the cluster size of a given cell type (Supplementary Figs. 21f and 22d).

The joint embedding of cells and features produced by SIMBA is fundamentally distinguished from other multi-omics integration methods. However, we still sought to compare SIMBA embeddings of cells with two widely adopted single-cell multi-omics integration methods, Seurat3 and LIGER, on the basis of their ability to integrate single-cell modalities while persevering cellular heterogeneity (Supplementary Note 8). We observed that SIMBA achieved the overall best performance on the mouse skin SHARE-seq dataset and 10x PBMCs multiome dataset.

## Discussion

The rapid development of multi-omics assays has outpaced the corresponding computational frameworks required to gain integrative insights from such rich data. This disparity highlights a need for methods that break through previous limitations and that can easily be extended to future cell measurements. SIMBA satisfies the need as a comprehensive and extensible method for exploring cellular heterogeneity and regulatory mechanisms. SIMBA models cells and measured features as nodes encoded in a graph and employs a scalable graph embedding procedure to embed nodes of cells and features into a shared latent space. We demonstrate that direct graph representations of single-cell data capture not only the relations between cells and the quantified features of the experiment (for example, gene expression or chromatin accessibility), but also hierarchical relations between features. The SIMBA co-embedding space enables simultaneous learning of cellular heterogeneity and cell-type-specific multimodal features and complements the current gene regulatory network analyses. SIMBA also circumvents the ordinary reliance on cell clustering for feature discovery that may lead to artifactual discovery or false negative results.

SIMBA has been extensively benchmarked across single-cell modalities and tasks, obtaining performance metrics that are better than or comparable to those of current state-of-the-art methods developed for the respective task. These results suggest a wide applicability of SIMBA's graph-based framework, obviating the need to combine multiple analysis tools.

Neural network embeddings hold substantial promise for the analysis of biological data. Previous applications of embedding models include functional annotation of genes[35], modeling TF-binding preferences[13,36] and more recent single-cell RNA-seq analyses[37,38].

Despite its promising capabilities, SIMBA faces potential limitations and has areas for improvement. Integrating sample-level data, such as time points and perturbations, could prove challenging, because it requires additional layers of complexity to accurately represent these dimensions. Spatial data could enhance SIMBA's ability to analyze complex datasets, such as spatial transcriptomics[39], by incorporating spatial proximity into the graph. Additionally, the framework could be extended to analyze three-dimensional chromatin conformation by encoding DNA segment interactions to represent gene–regulatory region links[40]. Although adapting SIMBA to various experimental designs will be achievable, interpreting the output embedding may vary depending on the input graph and training process, necessitating domain-specific expertise.

Overall, SIMBA is versatile and can accommodate features of various domains, provided they can be encoded into a connected graph. We believe that SIMBA will simplify the burden of developing methods for new single-cell tasks and measurements, while laying a groundwork for the development of new non-cluster-centric analysis methods.

## Online content

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

## Methods

### Single-cell data preprocessing

**Single-cell RNA-seq.** Genes expressed in fewer than three cells were filtered. Raw counts were library-size-normalized and subsequently log-transformed. Optionally, variable gene selection[12] (a Python version is implemented in SIMBA that is inspired by Scanpy[2]) may be performed to remove non-informative genes and accelerate the training procedure. Notable differences in the resulting cell embeddings were not observed upon limiting feature input to those identified by variable gene selection, but SIMBA embeddings of non-variable genes will not be generated as they are not encoded in the graph.

**Single-cell ATAC-seq.** Peaks present in fewer than three cells were filtered. Optionally, we implemented a scalable truncated-SVD-based procedure to select variable peaks as a preliminary step to additionally filter non-informative peaks and accelerate the training procedure. First, the top $k$ principal components (PCs) were selected, with $k$ chosen on the basis of the elbow plot of variance ratio. Then, for each of the top $k$ PCs, peaks were automatically selected on the basis of the loadings using a knee point detection algorithm implemented by 'kneed'[41]. Finally, peaks selected for each PC were combined and denoted as 'variable peaks.' Similar to the observation made with scRNA-seq data, the optional step of variable peak selection has a negligible effect on the resulting cell embedding. Despite this minimal impact on the resulting embedding, this feature selection step has a notable practical advantage in reducing training procedure time.

k-mer and motif scanning was performed using the packages 'Biostrings' and 'motifmatchr' with JASPAR2020 (ref. [42]). Included in the implementation of SIMBA is a convenient R command line script 'scan_for_kmers_motifs.R,' which will convert a list of peaks (formatted in a bed file) to a sparse peaks-by-$k$-mers/motifs matrix, which is stored as an hdf5-formated file.

### Graph construction (five scenarios)

**Single-cell RNA-seq analysis.** When constructing a graph of cells and genes, an edge is added between a cell and a gene if a gene is expressed in a given cell. To distinguish the strength of each edge, a binning procedure is proposed to categorize gene expression values into different levels while preserving the original distribution. Different levels of gene expression are encoded by different types of relations. In detail, the distribution of non-zero values in the normalized gene expression matrix was first approximated using a $k$-means clustering-based procedure. First, the continuous non-zero values were binned into $n$ intervals (by default, $n = 5$). Bin widths were defined using one-dimensional $k$-means clustering, wherein the values in each bin are assigned to the same cluster center. The continuous matrix is then converted into a discrete matrix wherein $1, …, n$ are used to denote $n$ levels of gene expression. Zero values are retained in this matrix. Then, the graph was constructed by encoding two types of entities, cells and genes, as nodes and relations with $n$ different weights between them, that is, $n$ levels of gene expression, as edges. These $n$ relation weights range from 1.0 to 5.0, with a step size of $5 / n$ denoting gene expression levels (lowest: 1.0, highest: 5.0), such that edges corresponding to high expression levels affect embeddings more strongly than do those with intermediate or low expression levels. As expected, we observe that the discretized distribution approaches the original distribution as we increase the number of bins. However, increased resolution of expression imparts little effect on the resulting embeddings, and the implemented procedure performs well consistently, even with a coarse-grained binning (five bins) (Supplementary Fig. 24). This discretization is implemented in the SIMBA package using the function, 'si.tl.discretize().'

In addition to relation type weights, SIMBA also supports encoding gene expression values directly as edge weights when constructing a graph. Supplementary Fig. 25 shows that, using either original (middle) or discretized (right) gene expression levels as edge weights, this procedure generates similar embeddings to those generated by the binning procedure (left). This further suggests that our current discretization is effective in capturing biological information, and SIMBA is robust to the binning procedure. This support of edge weight is implemented in the SIMBA package using the function, 'si.tl.gen_graph(add_edge_weights=True).'

**Single-cell ATAC-seq analysis.** Peak-by-cell matrices were binarized: '1' indicates that there is at least one read within a peak, and '0' was assigned otherwise. The graph was constructed by encoding two types of entities, cells and peaks, as nodes and the relation between them, denoting the presence of a given peak in a cell, as edges. The single relation type was assigned with a weight of 1.0. When the DNA sequence features were available, they were encoded into the graph using $k$-mer and motif sequence entities as nodes. This was performed by first binarizing the peak-by-$k$-mer–motif matrix and then constructing an extension to the original peak–cell graph using the peaks, $k$-mers and motifs as nodes and the presence of these entities within peaks as edges between these additional nodes and the peak nodes. The relation between $k$-mers and peaks was assigned a weight of 0.02, whereas the relation between TF motifs was assigned a weight of 0.2. Of note, $k$-mers and motifs may be used independently of each other as node inputs to the graph, depending on the specific analysis task.

**Multimodal analysis.** The above outlined strategies for graph construction using scRNA-seq and scATAC-seq data were combined to construct a multi-omics graph.

**Batch correction.** A graph for each batch was constructed as described in 'Single-cell RNA-seq analysis.' Edges between cells of different batches were inferred through a procedure based on truncated randomized SVD to link disjoint graphs of different batches. More specifically, in the case of scRNA-seq data, consider two gene expression matrices $X1_{n_1 \times m}$ and $X2_{n_2 \times m}$, where $n_1$ and $n_2$ denote the number of cells and $m$ denotes the number of the shared features, that is, variable genes, between datasets. The matrix $X_{n_1 \times n_2}$ was then computed by multiplying $X1$ and transposed $X2$ (denoted by $X2^T$):

$$X = X1 \times X2^T$$

Truncated randomized SVD was subsequently performed on $X$:

$$X \approx U \times \Sigma \times V^T$$

where $U$ is an $n_1 \times d$ matrix, $\Sigma$ is a $d \times d$ matrix and $V$ is an $n_2 \times d$ matrix (by default, the number of components $d = 20$).

Both $U$ and $V$ were further $L2$ normalized. For each cell in $U$, we searched for $k$ nearest neighbors in $V$, and vice versa (by default, $k = 20$). Eventually, only the mutual nearest neighbors between $U$ and $V$ were retained as inferred edges between cells (represented as dashed lines in Fig. 5a). The procedure of inferring edges between cells of different batches is implemented in the function 'si.tl.infer_edges()' in the SIMBA package.

For multiple batches, SIMBA can flexibly infer edges between any pair of datasets. In practice, however, edges are inferred between the largest dataset(s) or the dataset(s) containing the most complete set of expected cell types and other datasets.

**Multi-omics integration.** scRNA-seq and scATAC-seq graphs were constructed following steps in 'Single-cell RNA-seq analysis' and 'Single-cell ATAC-seq analysis,' respectively. To infer the edges between cells of scRNA-seq and scATAC-seq, gene activity scores were first calculated for scATAC-seq data[3]. More specifically, for each gene, peaks within 100 kb upstream and downstream of the TSS were considered. Peaks overlapping the gene body region or within 5 kb upstream of

gene bodies were given the weight of 1.0. Otherwise, peaks were weighted on the basis of their distances to TSS using the exponential decay function: $e^{\frac{-distance}{5000}}$. Subsequently, the gene score of each gene was computed as a weighted sum of the considered peaks. These gene scores were then scaled to respective gene size. These steps were implemented by the function 'si.tl.gene_scores()' in SIMBA. For user convenience, the SIMBA package curates the gene annotations of several commonly used reference genomes, including hg19, hg38, mm9 and mm10. Once gene scores were obtained, the same procedure described in 'Batch correction' was performed to infer edges between cells profiled by scRNA-seq and scATAC-seq using the function 'si.tl.infer_edges()' in SIMBA.

The procedure of generating constructed graphs was implemented in the function 'si.tl.gen_graph()' in the SIMBA package.

## Graph embeddings with type constraints

Following the construction of a multi-relational graph between biological entities, we adapted graph embedding techniques from the knowledge graph and recommendation systems literature to construct unsupervised representations for these entities.

We provided as input a directed graph $G = (V, E)$, where $V$ is a set of entities (vertices) and $E$ is a set of edges, with a generic edge $e = (u, v)$ between a source entity $u$ and destination entity $v$. We further assumed that each entity has a distinct known type (for example, cell or peak).

Graph embedding methods learn a $D$-dimensional embedding vector for each $v \in V$ by optimizing a link prediction objective via stochastic gradient descent, with $D = 50$ used for our experiments. We denote the full embedding matrix as $\Theta \in R^{|V| \times D}$ and the embedding for an entity $v$ as $\theta_v$.

For an edge $e = (u, v)$, we denote $s_e = \theta_u \cdot \theta_v$ as the score for $e$, and optimize a multi-class log loss:

$$\mathcal{L}_e = -\log \frac{\exp(s_e)}{\sum_{e' \in \mathcal{N}} \exp(s_{e'})} w_e$$

where $\mathcal{N}$ is a set of 'negative sampled' candidate edges[43] generated by corrupting $e$, and $w_e$ is the edge weight, which is the relation weight by default but can vary by edge within each relation type. For example, edges between cells and genes can be encoded as a single relation with varying edge weights that encode normalized gene-expression level (see 'Single-cell RNA-seq analysis' in Methods). This log loss objective attempts to maximize the score for all $(u, v) \in E$ and minimize it for $(u, v) \notin E$.

Negative samples were constructed by replacing either the source or target entity in the target edge $e = (u, v)$ with a randomly sampled entity. However, in graphs like ours, where only edges between certain entity types are possible, previous work has shown that it is beneficial to optimize the loss only over candidate edges that satisfy the type constraints[44]. Thus, for example, for a cell–peak edge, we sampled only negative candidates between cell and peak entities. This modification is crucial in our setting, because most randomly selected edges will be of an invalid type (for example, peak–peak), forcing the embeddings to primarily be optimized for irrelevant tasks (for example, having a low dot product between every pair of peaks).

Furthermore, it has been frequently observed that, in graphs with a wide distribution of node degrees, it is advantageous to sample negatives proportional to some function of the node degree to produce more informative embeddings that don't merely capture the degree distribution[14,45]. For each graph edge in the dataset encountered in a training batch, we produced 100 negatives by corrupting the edge with a source or destination sampled uniformly from the nodes with the correct types for this relation and 100 by corrupting the edge with a source or destination node sampled with probability proportional to its degree[14].

As with many ML methods, graph embeddings are prone to overfitting in a low-data regime (that is, low ratio of edges to parameters).

We observed overfitting, measured as a gap between training and validation loss on the link prediction task, which we addressed with $L2$ regularization on the embeddings ($\theta$):

$$\mathcal{L}_{reg} = \mathcal{L} + \lambda \sum_{u \in N} \sum_{d=1}^{D} \theta_{ud}^2.$$

with $\lambda = wd \times wd_{interval}$. The weight decay parameter ($wd$) by default was calculated automatically as $\frac{C}{N_e}$, where $N_e$ is the training sample size (that is, the total number of edges) and C is a constant. The weight decay interval ($wd_{interval}$), we set it to 50 for all experiments.

We used the PyTorch-BigGraph framework, which provides efficient computation of multi-relation graph embeddings over multiple entity types and can scale to graphs with millions or billions of entities[14]. For 1.3 million cells, the PyTorch-BigGraph training itself takes only about 1.5 hours using 12 CPU cores without the requirement of GPU (https://simba-bio.readthedocs.io/en/latest/rna_10x_mouse_brain_1p3M.html).

The resulting graph embeddings have two desirable properties:

1. First-order similarity: for two entity types $T_1$ and $T_2$ with a relation between them, edges with high likelihood should have higher dot product; specifically, for any $u \in T_1$, the predicted probability distribution over edges to $T_2$ originating from $u$ is approximated as $\frac{e^{\mathbf{x_u \cdot x_v}}}{\sum_{v' \in T_2} e^{\mathbf{x_u \cdot x_{v'}}}}$.

2. Second-order similarity: within a single entity type, entities that have 'similar contexts,' that is, a similar distribution of edge probabilities, should have similar embeddings. Thus, the embeddings of each entity type provide a low-rank latent space that encodes the similarity of those entities' edge distributions.

## Evaluation of the model during training

During the PyTorch-BigGraph training procedure, a small percentage of edges was held out (by default, the evaluation fraction is set to 5%) to monitor overfitting and evaluate the final model. Five metrics were computed on the reserved set of edges, including mean reciprocal rank (MRR, the average of the reciprocal of the ranks of all positives), R1 (the fraction of positives that rank better than all their negatives, that is, have a rank of 1), R10 (the fraction of positives that rank in the top 10 among their negatives), R50 (the fraction of positives that rank in the top 50 among their negatives) and AUC (area under the curve). By default, we show MRR, along with training loss and validation loss, while other metrics are also available in the SIMBA package (Supplementary Fig. 1a). The learning curves for validation loss and these metrics can be used to determine when training has been completed. The relative values of training and validation loss, along with these evaluation metrics, can be used to identify issues with training (underfitting versus overfitting) and tune the hyperparameters weight decay, embedding dimension and number of training epochs appropriately. For example, in Supplementary Fig. 1a, training can be stopped once the validation loss plateaus. However, for most datasets, the default parameters do not need tuning (Supplementary Note 9).

## Softmax transformation

PyTorch-BigGraph training provides initial embeddings of all entities (nodes). However, entities of different types (for example, cells versus peaks, cells of different batches or modalities) have different edge distributions and thus may lie on different manifolds of the latent space. To make the embeddings of entities of different types comparable, we transformed the embeddings of features with the Softmax function by using the first-order similarity between cells (reference) and features (query). For batch correction or multi-omics integration, the Softmax transformation was also performed on the basis of the first-order similarity between cells of different batches or modalities.

Given the initial embeddings of cells (reference) $(\mathbf{v}_{\mathbf{c_1}}, \ldots, \mathbf{v}_{\mathbf{c_n}})$ and features $(\mathbf{v}_{\mathbf{f_1}}, \ldots, \mathbf{v}_{\mathbf{d_m}})$, the model-estimated probability $P$ of an edge $(c_i, f_j)$ obeys:

$$P\left(e_{c_i, f_j}\right) \propto \exp\left(\mathbf{v}_{\mathbf{c_i}} \cdot \mathbf{v}_{\mathbf{f_j}}\right)$$

Therefore, if a random edge was sampled from feature $f_j$ to a cell, the model would estimate the probability distribution $p$ over such edges as:

$$p_{c_i, f_j} = \frac{\exp(\mathbf{v}_{\mathbf{c_i}} \cdot \mathbf{v}_{\mathbf{f_j}})}{\sum_{k=1}^{n} \exp(\mathbf{v}_{\mathbf{c_k}} \cdot \mathbf{v}_{\mathbf{f_j}})}$$

That is, the Softmax weights between all cells $\{c_i\}$ and the feature $f_j$. We can then compute new embeddings $\hat{\mathbf{v}}$ for features as a linear combination of the cell embeddings weighted by the edge probabilities raised to some power:

$$\hat{\mathbf{v}}_{f_j} = \frac{\sum_{i=1}^{n} p_{c_i, f_j}^{T^{-1}} \mathbf{v}_{\mathbf{c_i}}}{\sum_{i=1}^{n} p_{c_i, f_j}^{T^{-1}}}$$

$T$ is a temperature hyperparameter that controls the sharpness of the weighting over cells. At $T = 1$, the cell embeddings are weighted by their estimated edge probabilities; at $T \to 0$, each feature embedding is assigned the cell embedding of its nearest neighbor; at $T \to \infty$, it becomes a discrete uniform distribution, and each query becomes the average of reference embeddings. We set $T = 0.5$ for all the analyses.

These steps are implemented in the function 'si.tl.embed()' in the SIMBA package.

## Metrics to assess cell-type specificity

SIMBA calculates a probability score (represented as a dot product) of assigning a feature to a cell, and therefore generates a probability distribution of all cells for each feature. On the basis of this or its derived probability distribution (as shown in SIMBA barcode plots), four metrics can assess the cell type specificity of each feature from different aspects. The max score[46] averaging the normalized probabilities of the top 50 cells serves as a metric of confidence towards cell-type assignment and aids in filtering noisy features (a higher value indicates higher cell-type specificity). The Gini index[46] is calculated from a Softmax-transformed probability distribution to evaluate the deviation from a perfectly uniform distribution and thus features that show an imbalanced distribution (that is, cell-type-specific) are assigned with higher Gini index values (a higher value indicates higher cell-type specificity). s.d. measures the amount of variation in the probability distribution, and a high value indicates a higher deviation within the cells (that is, cell-type-specific) (a higher value indicates higher cell-type specificity). Entropy measures the information content and captures to what extent the cells are spread out over a Softmax-transformed probability distribution; a lower entropy indicates the distribution is nearly concentrated on one subset of cells (that is, cell-type-specific) (a lower value indicates higher cell-type specificity). We observed that these four metrics generally give consistent results. For each SIMBA metric plot, by default, the Gini index is plotted against the max value. For feature $f_j$:

The max value is defined as the average normalized similarity of top $k$ cells (by default, $k = 50$). The similarity normalization function is defined as:

$$\text{norm}(x_i) = x_i - \log \frac{\sum_{j=1}^{n} \exp(x_j)}{n}$$

where $i = 1, \ldots, n$, $n$ is the number of cells and $x_i$ represents the dot product of $\hat{\mathbf{v}}_{\mathbf{f_j}}$ and the embedding of cell $i$.

The max value is computed as:

$$\max(f_j) = \frac{\sum_{i=1}^{k} \text{norm}(x_i)}{k}$$

The Gini index is computed as:

$$\text{gini}(f_j) = \frac{\sum_{i=1}^{n} (2i - n - 1) \times p_{c_i, f_j}}{n \sum_{i=1}^{n} p_{c_i, f_j}}$$

The s.d. is computed as:

$$\text{s.d.}(f_j) = \sqrt{\frac{1}{n-1} \sum_{i=1}^{n} (x_{c_i, f_j} - \mu)^2}$$

where $\mu = \frac{1}{n} \sum_{i=1}^{n} x_{c_i, f_j}$.

Entropy is computed as:

$$\text{entropy}(f_j) = -\sum_{i=1}^{n} p_{c_i, f_j} \log(p_{c_i, f_j})$$

The statistical significance of these metrics can be optionally calculated on the basis of the comparison with the metrics derived from 'null' feature nodes that have shuffled edges with the same node degree (that is, the number of edges linked to a node) distribution as the input graph. For example, to obtain the significance of genes' cell-type-specificity metrics in scRNA-seq analysis, we constructed a graph with the original gene and cell nodes together with the 'null' gene nodes that have shuffled edges to the original cell nodes. For indirectly linked DNA sequence features, such as TF motifs and $k$-mers, the shuffle was performed for the edges between the DNA sequence features and peaks. The degree-preserving edge shuffle is achieved by randomly permuting the node index among the nodes with the same degree, separately for source and destination nodes. The degree-preserving shuffle ensures the null metric distribution reflects the bias in edge scores from null nodes to cells solely owing to the property of each node while destroying the graph connectivity (that is, biological information in SIMBA's input graph).

For the significance calculation, SIMBA learns the embeddings of null nodes together with the cells and features from the original graph. During the training with the null nodes, the loss from the edges involving the null nodes will not affect the real nodes' embeddings, and this is enabled by using the 'fix' operator in simba_pbg. For example, the loss from the edge between a real cell and a null gene will not be propagated to the embedding of the real cell. The default number of null nodes is set to 20 times the number of genes for scRNA-seq data and 5 times the number of peaks for scATAC-seq and SHARE-seq data. $P$ values are calculated on the basis of the null distribution of the metric values from random graph shuffling. The false discovery rate is further calculated using the Benjamini–Hochberg method.

## Queries of entities in SIMBA space

The informative SIMBA embedding space serves as a database of entities including cells and features. To query the 'SIMBA database' for the neighboring entities of a given cell or feature, we first built a $k$-d tree of all entities based on their SIMBA embeddings. We then searched for the nearest neighbors in the tree using Euclidean distance. To do so, SIMBA query can perform either $k$ nearest neighbors (KNN) or nearest neighbor search within a specified radius. SIMBA also provides the option to limit the search to entities of certain types, which is useful when a certain type of entity substantially outnumbers others. For example, the $k$ nearest features of a given cell may be all peaks, while genes are the features of interest. In this case, SIMBA allows users to add 'filters' to ensure that nearest neighbor search is performed within

the specified types of entities. This procedure is implemented in the function 'st.tl.query()' and its visualization is implemented in the function 'st.pl.query()' in the SIMBA package.

## Identification of master regulators

To define a master regulator a priori, we postulated that both its TF motif and TF gene should be cell-type specific, given that active gene regulation involves both the expression of a TF and accessibility of its binding sites. Thus, the TF motif and TF gene should be embedded closely in the shared latent space. To identify master regulators, we took into consideration both the cell type specificity of each pair of a TF motif and a TF gene, and the distance between them. More specifically, for each TF motif, first its distances (Euclidean distance by default) to all the genes are calculated in the SIMBA embedding space. Then, the rank of this TF gene among all these genes is computed. In addition, we also assess the cell type specificity of the pair using SIMBA metrics (by default, max value and Gini index are used). The same procedure is performed for all TFs. Finally, we identify master regulators by filtering out TFs with low cell-type specificity and scoring them by TF gene rank. This procedure is implemented in the function 'st.tl.find_master_regulators()' in the SIMBA package.

## Identification of TF target genes

To infer the target genes of a given master regulator, we postulated that, in the shared SIMBA embedding space, (1) the target gene is close to both the TF motif and the TF gene, indicating that the expression of the target gene is highly correlated with the expression of the TF and the accessibility of the TF motif in a cell-type-specific way; and (2) the accessible regions (peaks) near the target gene loci must be close to both the TF motif and the target TF gene, indicating that the accessibility of the *cis*-regulatory elements near the target gene locus is highly correlated with the expression of the TF and the accessibility of the TF motif in a cell-type-specific way.

Given a master regulator, its target genes are identified by comparing the locations of the TF gene, TF motif and the peaks near the genomic loci of candidate target genes in the SIMBA co-embedding space (Fig. 4e). More specifically, we first searched for *k* nearest neighbor genes around the motif (TF motif) and the gene (TF gene) of this master regulator, respectively (*k* = 200, by default). The union of these neighbor genes is the initial set of candidate target genes. These genes are then filtered on the basis of the criterion that open regions (peaks) within 100 kb upstream and downstream of the TSS of a putative target gene must contain the TF motif.

Next, for each candidate target gene, we computed four types of distances in SIMBA embedding space: distances between the embeddings of (1) the candidate target gene and TF gene; (2) the candidate target gene and TF motif; (3) peaks near the genomic locus of the candidate target gene and TF motif; and (4) peaks near the genomic locus of the candidate target gene and the candidate gene. All the distances (Euclidean distances by default) are converted to ranks out of all genes or all peaks to make the distances comparable across different master regulators.

The final list of target genes is decided using the calculated ranks, using two criteria: (1) at least one of the nearest peaks to TF gene or TF motif is within a predetermined range (top 1,000, by default); and (2) the average rank of the candidate target gene is within a predetermined range (top 5,000, by default). This procedure is implemented in the function 'st.tl. find_target_genes ()' in SIMBA.

## Benchmarking scATAC-seq computational methods

To compare SIMBA with other scATAC-seq computational methods, including SnapATAC[4], Cusanovich[47] and cisTopic[48], we employed the previously developed benchmarking framework from Chen et al.[16] (Supplementary Table 2). This framework evaluates different methods on the basis of their ability to distinguish cell types. We applied three clustering algorithms: *k*-means clustering, hierarchical clustering and Louvain on the feature matrix derived from each method.

For datasets with ground truth (FACS-sorted labels or known tissue labels), including simulated bone marrow data, Buenrostro et al.[17] and sci-ATAC-seq subset, three metrics including adjusted Rand index (ARI), adjusted mutual information (AMI) and homogeneity are applied to evaluate the performance. ARI measures the similarity between two clusters, comparing all pairs of samples assigned to matching or different clusters in the predicted clustering solution versus the true cluster/cell type label. AMI describes an observed frequency of co-occurrence compared with an expected frequency of co-occurrence between two variables, informing the mutual dependence or strength of association of these two variables. Homogeneity measures whether a clustering algorithm preserves cluster assignments towards samples that belong to a single class. A higher metric value indicates a better clustering solution.

For the 10x PBMCs dataset with no ground truth, the residual average Gini Index (RAGI) proposed in the benchmarking study[16] is used as the clustering evaluation metric. RAGI measures the relative exclusivity of marker genes to their corresponding clusters in comparison to housekeeping genes, which should demonstrate low specificity to any given cluster. In brief, the mean Gini Index is computed for both marker genes and housekeeping genes. The difference between the means is computed to obtain the average residual specificity (that is, RAGI) of a clustering solution with respect to marker genes. A higher RAGI indicates a better separation of biologically distinct clusters.

## Benchmarking single-cell batch correction methods

The batch correction performance of SIMBA was compared to Seurat3 (ref. 12), LIGER[11] and Harmony[10] in two benchmark datasets: the mouse atlas dataset and the human pancreas dataset (Supplementary Table 2). For Seurat3, LIGER and Harmony, the batch correction was done with the same parameters used in a previous benchmark study[28].

To evaluate the batch integration performance, average Silhouette width (ASW), adjusted Rand index (ARI) and local inverse Simpson's index (LISI)[10] were calculated for the batches and cell types using the Euclidean distance, as described in a previous benchmark[28]. To make a fair evaluation, only the cell types that are present in all batches were considered. We used the same number of dimensions (50) for these methods, and all other parameters were set as in the benchmark (Supplementary Note 7).

## Benchmarking single cell multi-omics integration methods

Two pairs of scRNA-seq and scATAC-seq datasets manually split from the dual-omics SHARE-seq mouse skin dataset and 10x PBMCs dataset, respectively, were used for the modality integration task. For Seurat3 and LIGER, the parameters and preprocessing were done as described in their documentations. However, for the LIGER analysis of the SHARE-seq mouse skin dataset, the parameter 'lambda' was set to 30, and the 'ref_dataset' was set to scATAC-seq to get a better alignment. For the Raw results, the activity matrix of scATAC-seq was constructed using Seurat3, and the first 20 PCs of the scRNA-seq count matrix and the activity matrix were used for the comparison. The integration results generated by each method were evaluated with four metrics—anchoring distance, anchoring distance rank, silhouette index and cluster agreement (Supplementary Note 8).

## Reporting summary

Further information on research design is available in the Nature Portfolio Reporting Summary linked to this article.

## Data availability

All the datasets used in this study, including eight scRNA-seq datasets, four scATAC-seq datasets and three dual-omics datasets, are summarized in Supplementary Table 2 and are curated in the SIMBA package

(https://simba-bio.readthedocs.io/en/latest/API.html#datasets). They can be easily downloaded and imported directly to reproduce the analyses presented in this paper. We have also deposited all the datasets to Zenodo at https://doi.org/10.5281/zenodo.7697355.

In addition, we also provide the source of these published datasets. For scRNA-seq datasets, the 10x PBMCs dataset is available at https://support.10xgenomics.com/single-cell-gene-expression/datasets/1.1.0/pbmc3k; the two mouse atlas datasets are available from https://github.com/JinmiaoChenLab/Batch-effect-removal-benchmarking/tree/master/Data/dataset2; the five human pancreas datasets are available from https://github.com/JinmiaoChenLab/Batch-effect-removal-benchmarking/tree/master/Data/dataset4. The scATAC-seq datasets are available from https://github.com/pinellolab/scATAC-benchmarking. For dual-omics datasets, the SHARE-seq mouse skin dataset is available from GSE140203; the mouse cerebral cortex SNARE-seq dataset is available from GSE126074; the 10x PBMCs multiome dataset is available from https://support.10xgenomics.com/single-cell-multiome-atac-gex/datasets/1.0.0/pbmc_granulocyte_sorted_10k. Source data are provided with this paper.

## Code availability

We provide a comprehensive Python package 'simba' available at https://anaconda.org/bioconda/simba and https://github.com/pinellolab/simba. All the proposed procedures are implemented in the 'simba' package. 'simba' can be easily installed with conda 'conda install simba'. We also built a website (https://simba-bio.readthedocs.io), providing a detailed introduction of the 'simba' software and several SIMBA tutorials for different types of single-cell analyses presented in this paper. Scripts used for performance comparison are available at https://github.com/pinellolab/simba_comparison. The version of 'simba' used for the analyses presented in this paper has been deposited to Zenodo (https://doi.org/10.5281/zenodo.7697337).

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

## Acknowledgements

The authors thank L. Wu, Facebook, for the helpful discussions about StarSpace; S. Ma and J. Buenrostro, Broad Institute of MIT and Harvard, for sharing the SHARE-seq data and metadata; and J. Delano, Harvard Medical School, for testing the SIMBA package and providing feedback. We also acknowledge members of the Pinello lab, Massachusetts General Hospital/Harvard Medical School, for helpful comments and feedback. This project has been made possible in part by grant number 2019-202669 from the Chan Zuckerberg Foundation to L.P. L.P. is also partially supported by the National Human Genome Research Institute (NHGRI) Genomic Innovator Award (R35HG010717). M.E.V. is supported by the National Institutes of Health (NIH) under the Ruth L. Kirschstein National Research Service Award (1F31CA257625) from the National Cancer Institute (NCI).

## Author contributions

H.C. and L.P. conceived this project. H.C. developed SIMBA, wrote the SIMBA package and performed SIMBA analysis on all datasets. A.L. contributed to the adaptation of PyTorch-BigGraph to single-cell analysis. J.R. led the development of statistical procedure and the comparison analysis in marker gene detection. J.R. and H.C. performed the comparison analysis on batch correction and data integration. M.E.V. led the comparison analysis in computational complexity and the evaluation of input features. M.E.V. and H.C. performed the comparison analysis on scATAC-seq data. L.P. and A.L. provided guidance and supervised this project. All the authors wrote and approved the final manuscript.

## Competing interests

A.L. was employed by Facebook AI Research during the time he worked on this manuscript. The remaining authors declare no competing interests.

## Additional information

**Correspondence and requests for materials** should be addressed to Adam Lerer or Luca Pinello.

Adam Lerer

# Reporting Summary

## Statistics

For all statistical analyses, confirm that the following items are present in the figure legend, table legend, main text, or Methods section.

| n/a | Confirmed | |
|---|---|---|
| ☐ | ☒ | The exact sample size (*n*) for each experimental group/condition, given as a discrete number and unit of measurement |
| ☒ | ☐ | A statement on whether measurements were taken from distinct samples or whether the same sample was measured repeatedly |
| ☐ | ☒ | The statistical test(s) used AND whether they are one- or two-sided *Only common tests should be described solely by name; describe more complex techniques in the Methods section.* |
| ☐ | ☒ | A description of all covariates tested |
| ☐ | ☒ | A description of any assumptions or corrections, such as tests of normality and adjustment for multiple comparisons |
| ☐ | ☒ | A full description of the statistical parameters including central tendency (e.g. means) or other basic estimates (e.g. regression coefficient) AND variation (e.g. standard deviation) or associated estimates of uncertainty (e.g. confidence intervals) |
| ☐ | ☒ | For null hypothesis testing, the test statistic (e.g. *F*, *t*, *r*) with confidence intervals, effect sizes, degrees of freedom and *P* value noted *Give P values as exact values whenever suitable.* |
| ☒ | ☐ | For Bayesian analysis, information on the choice of priors and Markov chain Monte Carlo settings |
| ☒ | ☐ | For hierarchical and complex designs, identification of the appropriate level for tests and full reporting of outcomes |
| ☒ | ☐ | Estimates of effect sizes (e.g. Cohen's *d*, Pearson's *r*), indicating how they were calculated |

*Our web collection on statistics for biologists contains articles on many of the points above.*

## Software and code

Policy information about availability of computer code

| Data collection | No software was used for data collection. |
|---|---|

| Data analysis | We provide a comprehensive Python package 'simba' available at https://anaconda.org/bioconda/simba and https://github.com/pinellolab/simba. All the proposed procedures are implemented in the "simba" package. 'simba' can be easily installed with conda "conda install simba". We also built a website (https://simba-bio.readthedocs.io), providing a detailed introduction of the 'simba' software and several SIMBA tutorials for different types of single-cell analyses presented in this manuscript. Scripts used for performance comparison are available at https://github.com/pinellolab/simba_comparison. The version of 'simba' used for the analyses presented in this manuscript was deposited at https://doi.org/10.5281/zenodo.7697337.<br><br>Tools used in the data analysis in this manuscript:<br><br>python v3.7<br>simba v1.1<br>simba_pbg v1.1<br>Scanpy v1.7.1<br>singleCellHaystack v0.3.4<br>CellID v 1.2.1<br>Seurat v3.2.3<br>LIGER v 0.5.0<br>Harmony v0.1.0 |
|---|---|

For manuscripts utilizing custom algorithms or software that are central to the research but not yet described in published literature, software must be made available to editors and reviewers. We strongly encourage code deposition in a community repository (e.g. GitHub). See the Nature Portfolio guidelines for submitting code & software for further information.

# Data

Policy information about availability of data

All manuscripts must include a data availability statement. This statement should provide the following information, where applicable:
- Accession codes, unique identifiers, or web links for publicly available datasets
- A description of any restrictions on data availability
- For clinical datasets or third party data, please ensure that the statement adheres to our policy

All the datasets used in this study, including eight scRNA-seq datasets, four scATAC-seq datasets, and three dual-omics datasets are summarized in Supplementary Table 2 and are curated in the SIMBA package(https://simba-bio.readthedocs.io/en/latest/API.html#datasets). They can be easily downloaded and imported directly to reproduce the analyses presented in this manuscript. We have also deposited all the datasets to Zenodo at https://doi.org/10.5281/zenodo.7697355.

In addition, we also provide the source of these published datasets. For scRNA-seq datasets, the 10x PBMCs dataset is available at https://support.10xgenomics.com/single-cell-gene-expression/datasets/1.1.0/pbmc3k; the two mouse atlas datasets are available from https://github.com/JinmiaoChenLab/Batch-effect-removal-benchmarking/tree/master/Data/dataset2; the five human pancreas datasets are available from https://github.com/JinmiaoChenLab/Batch-effect-removal-benchmarking/tree/master/Data/dataset4. The scATAC-seq datasets are available from https://github.com/pinellolab/scATAC-benchmarking. For dual-omics datasets, the SHARE-seq mouse skin dataset is available from GSE140203; the mouse cerebral cortex SNARE-seq dataset is available from GSE126074; the 10 PBMCs multiome dataset is available from https://support.10xgenomics.com/single-cell-multiome-atac-gex/datasets/1.0.0/pbmc_granulocyte_sorted_10k.

# Human research participants

Policy information about studies involving human research participants and Sex and Gender in Research.

| Reporting on sex and gender | N/A |
|---|---|
| Population characteristics | N/A |
| Recruitment | N/A |
| Ethics oversight | N/A |

Note that full information on the approval of the study protocol must also be provided in the manuscript.

# Field-specific reporting

Please select the one below that is the best fit for your research. If you are not sure, read the appropriate sections before making your selection.

☒ Life sciences   ☐ Behavioural & social sciences   ☐ Ecological, evolutionary & environmental sciences

For a reference copy of the document with all sections, see nature.com/documents/nr-reporting-summary-flat.pdf

# Life sciences study design

All studies must disclose on these points even when the disclosure is negative.

| | |
|---|---|
| Sample size | A sample-size calculation was not conducted. Instead, the original authors who made their datasets publicly available determined the sample size. |
| Data exclusions | There was no exclusion of data. |
| Replication | We didn't have control over the experimental design because we utilized pre-existing public datasets. Thus, this is not applicable. |
| Randomization | We didn't have control over the experimental design because we utilized pre-existing public datasets. Thus, this is not applicable. |
| Blinding | We didn't have control over the experimental design because we utilized pre-existing public datasets. Thus, this is not applicable. |

# Reporting for specific materials, systems and methods

We require information from authors about some types of materials, experimental systems and methods used in many studies. Here, indicate whether each material, system or method listed is relevant to your study. If you are not sure if a list item applies to your research, read the appropriate section before selecting a response.

## Materials & experimental systems

| n/a | Involved in the study |
|---|---|
| ☒ | ☐ Antibodies |
| ☒ | ☐ Eukaryotic cell lines |
| ☒ | ☐ Palaeontology and archaeology |
| ☒ | ☐ Animals and other organisms |
| ☒ | ☐ Clinical data |
| ☒ | ☐ Dual use research of concern |

## Methods

| n/a | Involved in the study |
|---|---|
| ☒ | ☐ ChIP-seq |
| ☒ | ☐ Flow cytometry |
| ☒ | ☐ MRI-based neuroimaging |

