## [Peer Review File · Nature Methods]

Peer Review Information

Manuscript Title: SIMBA: SIngle-cell eMBedding Along with features

Corresponding author name(s): Luca Pinello

Editorial Notes:

Reviewer Comments & Decisions:

Decision Letter, initial version:
--

15th May 2022

Dear Dr Pinello,

Your Article entitled "SIMBA: SIngle-cell eMBedding Along with features" has now been seen by 2 reviewers, whose comments are attached. While they find your work of potential interest, they have raised serious concerns which in our view are sufficiently important that they preclude publication of the work in Nature Methods, at least in its present form.

As you will see, the reviewers raise important concerns about different aspects of the paper. Should substantial revisions and new analysis allow you to fully address these criticisms we would be willing to look at a revised manuscript (unless, of course, something similar has by then been accepted at Nature Methods or appeared elsewhere). This includes submission or publication of a portion of this work somewhere else. We hope you understand that until we have read the revised paper in its entirety we cannot promise that it will be sent back for peer-review.

If you are interested in revising this manuscript for submission to Nature Methods in the future, please contact me to discuss your appeal before making any revisions. Otherwise, we hope that you find the reviewers' comments helpful when preparing your paper for submission elsewhere.

Sincerely,

Lin Tang, PhD
Senior Editor

Nature Methods

Reviewers' Comments:

Reviewer #1:

Remarks to the Author:

Remark to authors

The manuscript by Chen et al proposed a single-cell analysis for multimodal datasets based on graph embedding approaches. The authors demonstrate the applicability of their approach using multiple examples from the analysis of single-cell RNA-seq data, multi-omic analysis, data integration, and batch correction for uni and multiple modalities. The idea of encoding features (genes, peaks, k-mers, etc) along with cells is an interesting and promising approach that others have recently proposed. The paper's scope and claims are broad and cover a range of applications/methods within specialized subfields of single-cell computational methods (software development, data integration, multimodal data analysis). While this shows the approach's potential, it also necessitates the method to be compared with the highly optimized method for each task, which might be challenging. The usability and speed are also crucial for such a general framework which are not compared currently benchmarked. Overall I like the idea of finding multimodal features in a cluster-free way which is exciting and could have been the main focus of the paper; in the current shape, it is not clear why users would prefer SIMBA, over popular frameworks such as Seurat or scanpy, or other specialized software.

Major comments:

1- Others have proposed using knowledge graphs and/or embedding features in multimodal data integration (Cao and Gao 2022) and RNA-seq (Bilous et al. 2021; Zhao et al. 2021). It would be great if the authors put their work in the context of existing work and outline the main novelty of SIMBA.

2- As mentioned in the remark, I tested the method on the PBMC (~2.7k cells) example, and the pipeline ran slightly slower than scanpy analysis. However, for bigger datasets (~15k Pancreas), the pipeline was much slower on my laptop compared to scanpy and Seurat. While this is understandable partly due to the back-bone algorithm training of the embedding, most people perform analysis in CPU-based devices/servers. Thus, extensive benchmarking against popular frameworks in computational complexity and speed would be necessary.

3- The authors propose using the GINI-max combination to select essential features for each cell type. At the same time, cluster-based methods use DE testing and report statistical significance (e.g. P-value or Bayes factor) to account for false-positive discovery. How would SIMBA account for that? How does it compare to statistical methods?

4- The discretization of RNA expression of n-bins is not justified. This binning might remove the biological nuance when cell-type differences are subtle. Authors should assess how changing the current binning strategy might affect results. Examples of exciting scenarios could be genetic, disease perturbation, or highly similar subpopulations.

5- I really liked to link to TF/kmers for multiome datasets this could be pretty useful. However, I am not fully convinced why gene/TFs or as stated by the authors "1) the target gene is close to both the TF motif and the TF gene; 2) the accessible regions (peaks) near the target gene loci must be close to both the TF motif and the target TF gene." To the best of my understanding, there is explicit mechanistic or causal modeling linking those different modalities.

6- The relation between k-mers and peaks was assigned a weight of 0.02 while the relation between TF motifs was assigned a weight of 0.2. Of note, k-mers and motifs may be used? How and why these numbers are selected?

7- How robust is the training to different parameters? How much hyper-opt is needed for each dataset?

8- There are many conclusions and validation based on 2d UMAPs, which can be misleading. I suggest authors include more quantitative evidence to support their claims.

Minor:

1- The current distribution of the code on bioconda seems to be not functional. I tried to install conda, which failed on both intel and M1 laptops.

2- The Supplementary figures are pretty busy and with many UMAP plots that are hard to read.

3- There are some typos and missing words :

"Second, SIMBA analysis identified an unbiased set of DNA sequences, i.e., k-mers, that e important ..."

References:

Bilous, Mariia, Loc Tran, Chiara Cianciaruso, Santiago J. Carmona, Mikael J. Pittet, and David Gfeller. 2021. "Super-Cells Untangle Large and Complex Single-Cell Transcriptome Networks." bioRxiv. <https://doi.org/10.1101/2021.06.07.447430>.

Cao, Zhi-Jie, and Ge Gao. 2022. "Multi-Omics Single-Cell Data Integration and Regulatory Inference with Graph-Linked Embedding." *Nature Biotechnology*, May, 1–9.

Zhao, Yifan, Huiyu Cai, Zuobai Zhang, Jian Tang, and Yue Li. 2021. "Learning Interpretable Cellular and Gene Signature Embeddings from Single-Cell Transcriptomic Data." *Nature Communications* 12 (1): 1–15.

Reviewer #2:

Remarks to the Author:

This paper describes SIMBA, which embeds single cells and their features in a shared space and claims that the method can carry out many different types of single-cell analysis tasks, including working with multi-omic datasets. The method detects marker genes, corrects for batch effects, integrates multi-omics datasets, and generates biological insights such as identifying master regulators. The

paper shows an impressive array of applications of the software, some of which is quite convincing and some of which requires more work or better explanations of what was done. In general, the paper does quite a poor job of citing related work.

=====
=====

Introduction/overview

The paper makes clear that SIMBA is not algorithmically novel, but it does a poor job of providing details about its lineage. Prior work is only mentioned briefly in the discussion. For example, the multi-feature graph embedding idea has been proposed by StarSpace (Wu et al. 2017), which is not cited.

Line 73-74: In some cases there may not be any relationship between features. If each feature is associated with a genomic coordinate, then these coordinates may naturally link features between the two domains. But for non-genomic data (e.g., protein abundances or single-cell imaging data), such linkages may not exist.

Line 74-75: I don't really understand the critique in these lines. I guess it's related to the point above -- is it really the case that all the batch correction methods cited in this paragraph require being able to find feature correspondences?

Line 77-78: If the claim in this sentence is true, then it should be backed up with specific citations. "Most batch correction methods" is too vague.

The term "multi-entity" is not clear. All graphs contain multiple entities (multiple nodes and multiple edges). I think the point here is that the graph contains multiple node types. Please clarify this point.

Line 92: "derived from social networking technologies" requires a cite. Which specific algorithm or algorithms is it based on?

Line 99: What are the semantics of a cell-feature edge? In general, it's not clear what edge types are allowed and which are not (i.e., can every type of node have an edge to every other type)? Later (line 205), we are told that "cells and genes are connected by edges that embody the relation between them," without being told what are the semantics of that relationship. It's only in line 217 that we learn that the embedding "correctly embedded informative genes close to relevant cell types." Here, I am guessing that "informative" means genes that are differentially expressed in a given cell type. But I'm still left wondering what this really means. What happens, for example, to genes that are expressed in two or three cell types? The paragraph goes on to explain that non-cell-type specific genes get embedded into the middle, but I'm still left wondering about the interpretability of this kind of embedding in general. It's easy to think of scenarios where, say, three cell types A, B and C are arrayed in a line, and a gene that is expressed in all three ends up getting embedded in the middle cell type (B), giving the impression that it is specific to B.

Line 139-142: Give citations for the "recent graph embedding techniques."

In general, it would help to clarify the source of the algorithms in SIMBA. It seems, for example, that many of the ideas here come from the node2vec paper. You are up front about the fact that SIMBA is not novel, in the sense that it's borrowing ideas from other fields like social network analysis, but the

citation chain should be clearer.

Line 180: Suddenly we are told that each edge has an associated probability. This was a surprise to me. Is this supposed to be a probabilistic framework? If so, what is the semantics of the edge probability? Later, in line 228, we learn that we can get the "estimated probability of assigning a feature to a cell by SIMBA based on the recovered edge confidence," which implies perhaps that the edge probabilities have a particular semantics. How this semantics is enforced is not clear to me.

=====

Fig. 2

For the cell-type specific gene marker identification, the authors should carry out a more systematic evaluation on cell type marker gene selection. Currently SIMBA is only compared against scanpy. It would be more convincing to compare with methods that are developed for this particular task, such as Cell-ID and Vandenbon et al. 2020.

What is the red horizontal line in Fig. 2d?

Line 289: I am not convinced by this argument, since it is not stated whether IL7R and FCER1A are good cell type markers.

=====

Fig. 3

The idea (line 309) of embedding peaks and DNA sequences was previously put forth by the BindSpace paper (Yuan Nature Methods 2019). In general, the methods section should clarify which steps are taken from previous methods and which are novel to SIMBA.

Line 311: The fact that SIMBA uses Boolean features rather than TF-IDF features is not obviously a good (or bad) thing. The TF-IDF transformation is not computationally expensive. A priori, it's certainly plausible that switching to a Boolean representation could cause a significant loss of information. Also, please include a cite for scATAC-seq pipelines that use TF-IDF.

Line 315: I like that, in this case, the semantics of the edge types are clearly specified. :)

Line 330: This sentence, I think, requires some evidence that the blobs in the picture actually correspond to distinct cell types. It is certainly true that SIMBA separates the cells into distinct clusters, but the claim that these correspond to distinct cell types requires further analysis.

The analysis on lines 371-404 does a good job of showing the potential utility of the SIMBA co-embedding with this particular dataset.

I am not fully convinced by the examples in Figure 3. How are the peaks selected? They don't seem to be highly prioritized based on the cell-type specific calculation. In addition, the peaks used as examples here are not very close to GATA1, and in fact, they seem to belong to two different cell

subpopulations. Do peaks and k-mers located close to GATA1 in Fig. 3C make sense instead?

In Figure 3d, please change the scatter plot to a density plot or use transparent dots. Also, what is the cutoff for cell type specific features and how is it determined?

=====

Fig. 4

Line 456-462 & Fig. 4b: Lef1's gene/motif/peaks don't seem cell type specific. In addition, it is unclear how Lef1 is linked to the hair follicle as claimed.

Fig. 4c: How are the arrows drawn? Why are Lef1 peaks, gene and motif somewhat far away from each other? How are Lef1 peaks selected among all peaks?

Fig. 4d: The "rank score" needs to be further explained.

Fig. 4e: This panel is hard to understand. The figure legend suggests it is intended to illustrate how to find target genes of a given master regulator, but I don't see where the master regulator is. Besides, what are the green and gray shades? The SIMBA embedding axis on the bottom left corner indicates three dimensions, which is unnecessary since the plot is visualized in 2D.

=====

Fig. 6

Fig. 6c: To better visualize the multi-feature distributions on the embedding space, it would be helpful to make a separate UMAP for each node category (gene, cell, peaks) with the same coordinates.

=====

Methods

Line 745-746: Regarding the scRNA normalization, it seems that some genes will naturally have higher expression and some will not. Will this create a bias in the learned embeddings and downstream analysis?

Line 893-897: Does the edge weight between two nodes matter during optimization? The "degree" control used in the negative edge set seems potentially related, but it is not clear how degree is calculated, and how negatives are selected based on two nodes and different edge weights.

Line 904: The loss function is optimized over all types of edges, and it is not shown whether the different types of edges are optimized well. There should be an evaluation for how well the learned embeddings recapitulate known edges in each edge category on a hold out edge set.

line 942: It seems that the overall loss function includes some of the hyperparameters, so I don't understand how the hyperparameters can be selected based on the loss term. More generally, the model contains multiple hyperparameters. Please list them and show the grid of hyperparameters

tuned.

line 1006: The max and gini scores seem somewhat arbitrary; please add citations or intuitions. Furthermore, for the cell-type specificity measurement, the gini index doesn't make sense to me in this context, i.e. the score is affected by i , yet there is no description of how the index i is determined.

=====
=====

Other major comments:

The paper does not make a very convincing, systematic case that the co-embedding algorithm of genes, peaks, motifs and k-mers makes sense. Have you validated the gene-peak distance based on gene-peak proximity?

Please add some biological explanation on what it means if a TF is located far away from its corresponding motif.

The authors claim that SIMBA does clustering-free analysis (line 26 and in a lot of subsequent texts), which sounds interesting and reasonable. However, almost all of the example studies depend on previous cluster information. Can SIMBA find something surprising compared to previous methods -- e.g. a specific subset of cells driven by some unknown markers, or cell-specific regulation within a cell type?

The potential advantage of using graph embeddings on multiple features, besides interpretation, seems to be improving embedding accuracy. Does the performance (cell clustering, marker gene detection, batch correction) improve with more types of features taken into account?

Their code works fine in anaconda.

=====
=====

Minor comments:

What value of k is used for the k-mers?

Fig 1: bottom right: multi-omics integration figure, the transparent nodes are confusing.

line 907: Why normalize the sum of node embeddings by N_e (total number of edges)?

Although we cannot publish your paper, it may be appropriate for another journal in the Nature Portfolio

[REDACTED]

Note that any decision to opt in to In Review at the original journal is not sent to the receiving journal on transfer. You can opt in to [In Review](https://www.nature.com/nature-research/for-authors/in-review) at receiving journals that support this service by choosing to modify your manuscript on transfer. In Review is available for primary research manuscript types only.

Author Rebuttal to Initial comments

Reviewer #1

The manuscript by Chen et al proposed a single-cell analysis for multimodal datasets based on graph embedding approaches. The authors demonstrate the applicability of their approach using multiple examples from the analysis of single-cell RNA-seq data, multi-omic analysis, data integration, and batch correction for uni and multiple modalities.

The idea of encoding features (genes, peaks, k-mers, etc) along with cells is an interesting and promising approach that others have recently proposed. The paper's scope and claims are broad and cover a wide range of applications/methods within specialized subfields of single-cell computational methods (software development, data integration, multimodal data analysis). While this shows the approach's potential, it also necessitates the method to be compared with the highly optimized method for each task, which might be challenging. The usability and speed are also crucial for such a general framework which are not currently benchmarked. Overall I like the idea of finding multimodal features in a cluster-free way which is exciting and could have been the main focus of the paper; in the current shape, it is not clear why users would prefer SIMBA, over popular frameworks such as Seurat or scanpy, or other specialized software.

We appreciate the reviewer's excitement for our work and acknowledgement of our approach's potential.

To the best of our knowledge, SIMBA is the first method that proposes the concept of co-

embedding single- cells and multi-type features. Although a recent method ‘Cell-ID’¹ initially explored the simultaneous representation of cells and genes, it is non-graph-based and is limited to a single feature (i.e., genes). Without explicitly proposing the concept ‘co-embedding’, Cell-ID primarily focuses on gene signature extraction from scRNA-seq data. In contrast, SIMBA encodes both cells and various types of features into a large-scale graph and generates the co-embeddings of cells and multiple-type features, including not only genes, but also peaks and DNA sequences. This allows us to discover multimodal features involved in regulatory circuits in a clustering-free way, which none of existing methods can achieve. In addition, SIMBA is a more general graph- based co-embedding method, as opposed to a specific gene signature extraction method, and the same co- embedding procedure has been demonstrated to tackle a wide range of single-cell tasks.

A detailed comparison of SIMBA with other methods is presented in response to comment 1) and quantitative metrics to show the advantages of our approach over popular frameworks will be better highlighted in the manuscript.

Major comments:

1) *Others have proposed using knowledge graphs and/or embedding features in multimodal data integration (Cao and Gao 2022) and RNA-seq (Bilous et al. 2021; Zhao et al. 2021). It would be great if at authors put their work in the context of existing work and outline the main novelty of SIMBA.*

We have carefully reviewed the three manuscripts²⁻⁴ suggested by the reviewer and created a summary table to highlight the differences between SIMBA and them.

Comparison of
SIMBA with
existing methods

	Core algorithm	Graph nodes	Node featurization	Requires PCA or LSI as preprocessing steps	Output	scRNA-seq	scATAC-seq	Batch correction (single modality)	Multimodal (paired multi-omics)	Multi-omics integration (unpaired multi-omics)	Software Quality		
											Easy installation (software available in public packaging systems)	Dedicated Documentation website	Comprehensive and self-contained (no dependencies on heavy packages such as scanpy or seurat for preprocessing and plotting)
SIMBA	Multi-relation graph embeddings with type constraints	Cell and omics features (including genes, open regions, DNA sequences, etc.)	No	No	Co-embeddings of both cells and features (comparable)	✓	✓	✓	✓	✓	✓	✓	✓
GLUE [1]	Variational auto-encoders	Omics features (including genes, open regions, methylated sites)	No	Yes	Embeddings of cells and embeddings of features (not comparable)	X	X	X	X	✓	✓	✓	X
Metacell [2]	k-nearest neighbors	Cells	Yes	Yes	Embeddings of cells	✓	X	X	X	X	X	X	✓
scETM [3] * *not a graph-based method	Variational auto-encoders	NA	NA	No	Embeddings of cells and embeddings of features (not comparable)	✓	X	✓	X	X	✓	X	X

In summary, compared to existing methods, SIMBA provides the most comprehensive software with the following key innovations and advantages:

- 1) It doesn't require node featurization and ad-hoc transformations;
- 2) It can embed both cells and their features into the same space;
- 3) The SIMBA embeddings can be used to tackle a variety of biological tasks with comparable or better performance than dedicated methods, as demonstrated in the manuscript.

We are planning to revise the introduction section to better acknowledge previous efforts in this area and to highlight the novelties and advantages of SIMBA over competing methods.

2) As mentioned in the remark, I tested the method on the PBMC (~2.7k cells) example, and the pipeline ran slightly slower than scanpy analysis. However, for bigger datasets (~15k Pancreas), the pipeline was much slower on my laptop compared to scanpy and Seurat. While this is understandable partly due to the back-bone algorithm training of the embedding, most people perform analysis in CPU-based devices/servers. Thus, extensive benchmarking against popular frameworks in computational complexity and speed would be necessary.

This is a great suggestion, and we are planning to perform the proposed benchmark comparing complexity, running time and memory usage of SIMBA against Seurat and SCANPY on three scRNA-seq datasets: PBMC dataset (~2.7k cells, 10x Genomics), Pancreas dataset⁵ (~15k cells) and a large brain dataset (~1.3M cells, 10x Genomics).

3- The authors propose using the GINI-max combination to select essential features for each cell type. At the same time, cluster-based methods use DE testing and report statistical significance (e.g. P-value or Bayes factor) to account for false-positive discovery. How would SIMBA account for that? How does it compare to statistical methods?

This is an excellent suggestion. The current implementation of SIMBA provides ranked features similar to the recent, non-cluster-based method CellID¹. However, we agree with the

reviewer's comment that this is an important point. To address this, we will provide in the revised version a statistical procedure to assess the significance of the recovered cell type specificity and account for potential false positives.

4) The discretization of RNA expression of n -bins is not justified. This binning might remove the biological nuance when cell-type differences are subtle. Authors should assess how changing the current binning strategy might affect results. Examples of exciting scenarios could be genetic, disease perturbation, or highly similar subpopulations.

We are planning to expand the method section to explain the rationale of the proposed binning strategy and report new analyses to show how our binning procedure is robust. Importantly, we have implemented a new procedure that doesn't require the discretization of RNA expression and we will show in the revised version of the manuscript how this procedure can be alternatively applied.

5) *I really liked to link to TF/kmers for multiome datasets this could be pretty useful. However, I am not fully convinced why gene/TFs or as stated by the authors "1) the target gene is close to both the TF motif and the TF gene; 2) the accessible regions (peaks) near the target gene loci must be close to both the TF motif and the target TF gene." To the best of my understanding, there is explicit mechanistic or causal modeling linking those different modalities.*

We thank the reviewer for recognizing the usefulness of SIMBA in multimodal analysis. We agree this part was not sufficiently explained. We will expand on the text with examples to better describe our procedure and the mechanistic assumptions behind it.

6) *The relation between k-mers and peaks was assigned a weight of 0.02 while the relation between TF motifs was assigned a weight of 0.2. Of note, k-mers and motifs may be used? How and why these numbers are selected?*

We reasoned that it is important to consider for our optimization function the informativeness of each type of features and their representation in the graph (i.e., the number of nodes and edges). These weights may be modified depending on how informative a certain type of features is. For example, TF motifs are more informative and less abundant than unfiltered k-mers in explaining TF binding and therefore are assigned with a higher weight. In practice, we picked optimal weights (ranging from 0 to 10) with a grid search on 3 datasets and found that they performed consistently well across 12 other datasets. Therefore, we used the same relation weights in all the analyses shown in the manuscript.

7) *How robust is the training to different parameters? How much hyper-opt is needed for each dataset?*

The only parameter that needs to be tuned in our experiments is the “weight decay”. However, this may be empirically optimized based on the training sample size (i.e., the total number of edges) of the dataset, and therefore we provide an automatic procedure to determine the optimal weight decay parameter. All the experiments across 5 tasks in 15 datasets were performed with this procedure, and therefore we believe this is a robust procedure. In addition, we also provided several training metrics (they can be plotted using the function

“*si.pl.pbg_metrics()*”) to monitor the training process so the users can further fine-tune the auto-generated weight decay parameter based on these training metrics if needed. We will expand the method section of the manuscript to better clarify this.

8) *There are many conclusions and validation based on 2d UMAPs, which can be misleading. I suggest authors include more quantitative evidence to support their claims.*

We are planning to revise the manuscript through additional quantitative metrics to evaluate the quality of the SIMBA embeddings.

For scATAC-seq analysis, batch-correction, and multi-omics integration, we have already evaluated the SIMBA embeddings both qualitatively and quantitatively. We have also compared SIMBA against state-of-the-art methods using published benchmark frameworks for each task. To improve the clarity of these findings in the manuscript, we will better highlight these proposed quantitative metrics and validation procedures.

For scRNA-seq analysis, we are planning to collect gene signatures of cell types from published studies and evaluate the ability of SIMBA to recover these cell-type-specific marker genes. Marker gene recovery will be performed by calculating the ratios of correctly identified genes to the total number of collected marker genes at varying distance thresholds.

For multimodal analysis, we will quantify gene expression features as described above for the scRNA-seq analysis. For chromatin accessibility and TF-motif recovery, we will validate SIMBA embeddings by calculating the percentage of cell-type-specific peaks (or TF-motifs) that are in close proximity to the embeddings of each cell type based on external annotations.

Minor:

9) *The current distribution of the code on bioconda seems to be not functional. I tried to install conda, which failed on both intel and M1 laptops.*

We thank the reviewer for testing our software and appreciate that they are considering this during the review process. Unfortunately, we are unable to reproduce this error. In comment 2, we note that the reviewer describes being able to run SIMBA (see: Comment 2). Additionally, another reviewer has explicitly stated that “Their code works fine in anaconda.”

To ensure usability, we have implemented a continuous integration workflow through the SIMBA GitHub repository. This workflow automatically tests each release of SIMBA for both Linux and MacOS :

<https://github.com/pinellolab/simba/actions/workflows/CI.yml>, and all tests pass.

However, as the latest Intel and M1 chips roll out, we will keep updating and testing our software to make sure it is functional. We would appreciate if the reviewer could (anonymously) file a task on our repo with details of their issues so we can make the software

more widely accessible.

10) *The Supplementary figures are pretty busy and with many UMAP plots that are hard to read.*

We thank the reviewer for this suggestion and accordingly, we will split those busy supplementary figures into multiple figures.

11) *There are some typos and missing words:*

"Second, SIMBA analysis identified an unbiased set of DNA sequences, i.e., k-mers, that e important ..."

Unfortunately, it appears that the reviewer was reading an old version of our manuscript, which had been revised before submission to *Nature Methods*.

This sentence only appeared in our very first bioRxiv version, which was posted on October 18, 2021. Many changes have been introduced ever since. In our submitted version, this sentence has already been changed to “Second, SIMBA revealed an unbiased set of DNA sequences, i.e., k-mers, that represent important TF binding motifs involved in hematopoiesis.” without typos (Line 378-379). Nonetheless, we thank the reviewer for their attention to detail.

Reviewer #2

This paper describes SIMBA, which embeds single cells and their features in a shared space and claims that the method can carry out many different types of single-cell analysis tasks, including working with multi-omic datasets. The method detects marker genes, corrects for batch effects, integrates multi-omics datasets, and generates biological insights such as identifying master regulators. The paper shows an impressive array of applications of the software, some of which is quite convincing and some of which requires more work or better explanations of what was done. In general, the paper does quite a poor job of citing related work.

We are glad that the reviewer found the application of SIMBA to various tasks and numerous datasets impressive. We are also grateful for the suggestions that will help improve the presentation of our work. We will revise the manuscript by adding more relevant citations to better acknowledge past efforts and further highlight the novelty of our method.

=====
Introduction/overview

1) The paper makes clear that SIMBA is not algorithmically novel, but it does a poor job of providing details about its lineage. Prior work is only mentioned briefly in the discussion. For example, the multi-feature graph embedding idea has been proposed by StarSpace (Wu et al. 2017), which is not cited.

We are planning to cite additional prior works relevant to the lineage of graph embedding algorithms.

2) Line 73-74: In some cases there may not be any relationship between features. If each feature is associated with a genomic coordinate, then these coordinates may naturally link features between the two domains. But for non-genomic data (e.g., protein abundances or single-cell imaging data), such linkages may not exist.

This is an excellent point, and we will revise the manuscript to better explain our procedure. In SIMBA each feature doesn't need to be associated with a genomic coordinate. For example, in the current implementation for features corresponding to gene expression we only use their numerical expression values, and the genomic coordinates of the corresponding transcript are not necessary.

In general, SIMBA will work if features of different domains can be encoded into a single connected graph either directly or indirectly. Thus, even if a direct edge between two features is not observed in the input graph, these features can still be indirectly linked through intermediate edges with common entities. For examples, for the relations peak-gene and peak-protein abundance (genomic) – or gene-spatial location (non-genomic) , these features can still be indirectly linked through cells.

3) Line 74-75: I don't really understand the critique in these lines. I guess it's related to the point above – is it really the case that all the batch correction methods cited in this paragraph require being able to find feature correspondences?

We are planning to revise the text in the introduction section to clarify that finding feature correspondences is not required for all the batch correction/multi-omics integration methods.

4) Line 77-78: *If the claim in this sentence is true, then it should be backed up with specific citations. “Most batch correction methods” is too vague.*

We are planning to replace this sentence by explicitly listing the top-performing methods with citations based on a recent batch-correction benchmark study ⁵.

5) *The term “multi-entity” is not clear. All graphs contain multiple entities (multiple nodes and multiple edges). I think the point here is that the graph contains multiple node types. Please clarify this point.*

Yes, ‘multi-entity’ refers to multiple node types. We will revise the manuscript to better clarify the term ‘multi- entity’. Briefly, the concept of multi-entity proposed in PyTorch-BigGraph⁶ (PBG) is often referred in the field of graph representation learning as “heterogeneous graph” where nodes and edges can be heterogenous and/or correspond to different semantic. In our case nodes correspond to different domains (cells, gene, peaks) and the edges correspond to different relations i.e., a cell is expressing a gene, or a cell has a given chromatin accessibility peak.

6) Line 92: *“derived from social networking technologies” requires a cite. Which specific algorithm or algorithms is it based on?*

We will add references to relevant algorithms^{6,7}.

7) Line 99: *What are the semantics of a cell-feature edge? In general, it’s not clear what edge types are allowed and which are not (i.e., can every type of node have an edge to every other type)? Later (line 205), we are told that “cells and genes are connected by edges that embody the relation between them,” without being told what are the semantics of that relationship.*

It’s only in line 217 that we learn that the embedding “correctly embedded informative genes close to relevant cell types.” Here, I am guessing that “informative” means genes that are differentially expressed in a given cell type. But I’m still left wondering what this really means.

What happens, for example, to genes that are expressed in two or three cell types? The paragraph goes on to explain that non-cell-type specific genes get embedded into the middle, but I'm still left wondering about the interpretability of this kind of embedding in general. It's easy to think of scenarios where, say, three cell types A, B and C are arrayed in a line, and a gene that is expressed in all three ends up getting embedded in the middle cell type (B), giving the impression that it is specific to B.

Our plan to address this point is to revise the description of the procedures used in SIMBA and explicitly explain the semantics of a cell-feature edge and the interpretation of the 2D embedding as follows:

In SIMBA, the semantics of each cell-feature edge correspond to a single-cell measurement (e.g., the expression value of a gene or a chromatin-accessible peak observed in a cell). However, edges are also allowed between different features to capture and model the underlying regulatory mechanisms. For example, edges between

peaks and TF-motifs (or k-mers) capture the notion that a given TF may bind to a regulatory peak. In general, given a set of nodes corresponding to the different entities modeled in the knowledge graph, not all cell-feature or feature-feature edges are allowed. Figure 1 summarizes this potential relation represented by an edge and the semantics for the analyses presented in this manuscript, namely: (cell-gene): a cell expresses a given gene, (cell- peak): a cell has an accessible chromatin region, (peak-TF motif): a peak sequence contains a significant binding site for a given TF, (peak-kmer): a peak sequence contains a given k-mer sequence. No other edges are allowed (e.g., peak-peak, or gene-gene).

Regarding the interpretation of the embeddings and of informative genes, the 2D UMAP projection of cells and feature is helpful to qualitatively visualize their relations, however additional information is provided to the users to interpret cell type specificity of prioritized features in addition to their location. In fact, when external annotations are available users can visually inspect our proposed *barcode plots* for each feature as a diagnostic of cell type specificity, as explained below.

There are three main steps involved in generating the 2D co-embedding space: 1) the raw embeddings of cells and features (by default, 50 dimensions) are generated after PyTorch-BigGraph (PBG) training, in which embeddings of cells and features are not comparable; 2) the raw embeddings are converted into SIMBA co- embeddings of cells and features (by default, 50 dimensions) through Softmax transformation, in which embeddings of cells and features are comparable; 3) SIMBA co-embeddings of cells and features are visualized using UMAP (by default, 2 dimensions) (see methods).

The obtained 2D visualization can present to users a visual interpretation of the cell type specificity of features. This is particularly effective if features are highly cell type specific (i.e., relevant only for a single cell type).

However, some rare artifacts may emerge if all the features are embedded in the 2D UMAP as for the scenario proposed by the reviewer.

However, this scenario in practice may rarely happen in the 50-dimensional SIMBA embeddings as opposed to a 2-dimensional plane. Nonetheless, using the SIMBA barcode plot (e.g., Figure 2d) we can directly plot and investigate the probability of assigning a feature to each cell in this more expressive space. In this scenario (shown in the figure below), all three cell types will show high probability for this gene and will be evenly mixed in the barcode plot

(top panel), as opposed to the case of being cell type B -specific, in which only cell type B will show high probability (bottom panel). The 2D embedding serves as a qualitatively and useful way to visualize both cells and features but it also has its limitations just as other general dimensionality reduction methods. Therefore, we suggest users inspect SIMBA barcode plots to further validate and more quantitatively assess cell type specificity based on the neighboring features of cells.

8) Line 139-142: Give citations for the “recent graph embedding techniques.”

We will add relevant references⁶.

9) In general, it would help to clarify the source of the algorithms in SIMBA. It seems, for example, that many of the ideas here come from the node2vec paper. You are up front about the fact that SIMBA is not novel, in the sense that it’s borrowing ideas from other fields like social network analysis, but the citation chain should be clearer.

We are planning to revise the manuscript by adding more citations of related works and further clarifying the novelty of SIMBA.

In brief, SIMBA leverages the PyTorch-BigGraph (PBG) framework. PBG has been compared against node2vec-like general graph embedding methods in its own manuscript and its superior performance has been demonstrated. Our manuscript is focused on adapting PBG to single-cell analysis. As mentioned in comment 1), due to the unique challenges of single-cell data, in SIMBA, we have introduced multiple crucial and algorithmically innovative changes and procedures, including Softmax transformation, weight decay for controlling overfitting, etc. These procedures in SIMBA play a crucial role in modelling single-cell data and generating comparable embeddings (co-embeddings) of cells and features, which general

graph embedding methods alone cannot achieve.

10) Line 180: Suddenly we are told that each edge has an associated probability. This was a surprise to me. Is this supposed to be a probabilistic framework? If so, what is the semantics of the edge probability? Later, in line 228, we learn that we can get the “estimated probability of assigning a feature to a cell by SIMBA based on the recovered edge confidence,” which implies perhaps that the edge probabilities have a particular semantics. How this semantics is enforced is not clear to me.

We will further clarify the semantics of edge likelihoods where it is first mentioned.

In brief, the graph embedding algorithm can be thought of as producing a model (whose parameters are the graph embeddings themselves) that approximates the likelihood of an edge (u, v) existing in the graph. These likelihoods are what we refer to when we talk about “estimated probabilities”. We will clarify this in the text.

Fig. 2

12) For the cell-type specific gene marker identification, the authors should carry out a more systematic evaluation on cell type marker gene selection. Currently SIMBA is only compared against scanpy. It would be more convincing to compare with methods that are developed for this particular task, such as Cell-ID and Vandenbon et al. 2020.

We are planning to systematically compare SIMBA with Cell-ID¹ and Vandenbon et al. 2020⁸ in detecting marker genes as suggested by the reviewer.

13) What is the red horizontal line in Fig. 2d?

As the four genes have different scales of probability scores, we use the value of $1e-3$ as a line of reference to visually compare them. We will further clarify this in the legend.

14) Line 289: I am not convinced by this argument, since it is not stated whether IL7R and FCER1A are good cell type markers.

We will add relevant references in support of these two genes as cell type markers.

Fig. 3

14) The idea (line 309) of embedding peaks and DNA sequences was previously put forth by the BindSpace paper (Yuan Nature Methods 2019). In general, the methods section should

clarify which steps are taken from previous methods and which are novel to SIMBA.

We will extend the manuscript to better highlight the novelty of SIMBA against previously developed methods such as BindSpace.

Bindspace was built on the TagSpace model from Starspace⁷, in which each DNA sequence is treated as a sentence, each k-mer is treated as a word, and TFs are treated as hashtags (there are no cells involved). It is a non-graph-based model and is limited to tasks that can be converted into this strict setting. It was only used to analyze HT-SELEX data and has not been proven to be able to perform any single-cell tasks.

In contrast, SIMBA is based on a graph-based model that allows for multi-type entities and multi-type relations, making it flexible to handle any tasks that can be encoded as a general graph. Therefore, SIMBA can be used to

co-embed both cells and features, while Bindspace can only embed sequence features. By leveraging the efficient and scalable Pytorch-BigGraph framework⁶, and introducing multiple crucial and innovative procedures in this manuscript to create comparable cell-feature embedding together with the ranking of features based on the recovered edge likelihood. SIMBA has been demonstrated to perform various single-cell tasks and achieved the best or comparable performance against current state-of-the-art methods.

15) Line 311: The fact that SIMBA uses Boolean features rather than TF-IDF features is not obviously a good (or bad) thing. The TF-IDF transformation is not computationally expensive. A priori, it's certainly plausible that switching to a Boolean representation could cause a significant loss of information. Also, please include a cite for scATAC-seq pipelines that use TF-IDF.

We appreciate this insightful and analytical comment and find this discussion extremely productive towards continued improvement of methods development in this area. We will add citations for scATAC-seq pipelines that use TF-IDF.

We previously published a study benchmarking scATAC-seq computational methods (Chen et al Nature Communications 2018), in which we discussed various featurization procedures proposed for scATAC-seq data. We are aware that TF-IDF is an effective and non-expensive preprocessing step. However, here we wanted to show that without TF-IDF, SIMBA is still able to achieve better performance than other methods. This control experiment is required to discern the relative contribution of TF-IDF and SIMBA to the overall performance enhancement. We would like to point out that the best performance, which was achieved through straightforward binarization by SIMBA highlights the advantages of the graph embedding procedure used by SIMBA.

16) Line 315: I like that, in this case, the semantics of the edge types are clearly specified. 😊

We are glad that the reviewer liked the way edge types were explained.

17) Line 330: This sentence, I think, requires some evidence that the blobs in the picture

actually correspond to distinct cell types. It is certainly true that SIMBA separates the cells into distinct clusters, but the claim that these correspond to distinct cell types requires further analysis.

We will provide additional details on the interpretation of the SIMBA embedding results and quantitative metrics to show how well our proposed embedding can separate cell types.

18) The analysis on lines 371-404 does a good job of showing the potential utility of the SIMBA co-embedding with this particular dataset.

We are glad that the reviewer liked this part.

19) I am not fully convinced by the examples in Figure 3. How are the peaks selected? They d'n't seem to be highly prioritized based on the cell-type specific calculation. In addition, the peaks used as examples here are not very close to GATA1, and in fact, they seem to belong to two different cell subpopulations. Do peaks and k-mers located close to GATA1 in Fig. 3C make sense instead?

We will revise the manuscript to further clarify Figure 3.

In brief, the goal of Figure 3 is to assess embeddings of cells and features using prior biological knowledge, as opposed to the discovery of novel features. Therefore, the peaks were selected based on the genomic locus of the KLF1 gene. Instances of MEP-specific regulatory elements and gene regulation is supported by previous studies described in the manuscript (Line 391-404). We therefore expect MEP-specific regulatory elements and their corresponding genes to be embedded nearby MEP cells. We observed that these features are all embedded in the neighborhood of MEP cells (and far from other cell types) supporting the quality of the SIMBA embeddings. However, to show the effectiveness of our ranking procedure we will also investigate in the revised manuscript the cell type specificity and biological relevance of the peaks and k-mers closest to GATA1 as suggested.

20) In Figure 3d, please change the scatter plot to a density plot or use transparent dots. Also, what is the cutoff for cell type specific features and how is it determined?

We thank the reviewer of the great suggestion. We will overlay the density on the scatter plots (using a contour plot). The current implementation of SIMBA provides ranks of features instead of specifying cutoffs. But we are planning to provide in the revised version a statistical procedure to decide the statistical significance of each feature so that a cutoff can be specified. Please see also our response to Reviewer #1, comment 3).

Fig. 4

21) Line 456-462 & Fig. 4b: Lef1's gene/motif/peaks don't seem cell type specific. In addition, it is unclear how Lef1 is linked to the hair follicle as claimed.

9+.

Lef1 was one of the important genes described in the SHARE-seq paper. Therefore, we tried to recapitulate its biological role in skin development based on our embedding. Notably, Lef1 is automatically recovered as a key developmental gene with our master-regulator ranking (see Fig 4d). As shown in Supp Figure 7a, although Lef1 shows a Hair Shaft-cuticle cortex-specific pattern, it is also partially expressed in several other cell types.

Therefore, it is not surprising that Lef1 doesn't have the highest cell-type specificity score.

22) Fig. 4c: How are the arrows drawn? Why are Lef1 peaks, gene and motif somewhat far away from each other? How are Lef1 peaks selected among all peaks?

The arrows represent likely differentiation trajectories and were manually drawn for illustrative purposes based on previous SHARE-seq literature.

Despite the embeddings of the Lef1 peaks, genes, and motifs are not the closest to each other they are all embedded into the neighborhood of cells transitioning from TAC2 to Hair shaft.

In addition, given the expression of Lef1 can be regulated by the binding of up-stream factors and its expression sustained along a developmental path, we highlighted in the embedding the location of the peaks nearby the genomic locus of Lef1 (Peak1(Lef1), Peak2(Lef1), Peak3(Lef1). Interestingly, these peaks are embedded before the embedding corresponding to the Lef1 expression. This is consistent with what was described in the original study⁹, suggesting a delay between the chromatin accessibility of peaks within a gene locus and its expression.

As mentioned above, Lef1 peaks were selected based on the genomic locus of Lef1.

23) Fig. 4d: The “rank score” needs to be further explained.

We will add more details in both Fig.4d legend and the manuscript to explain the “rank score”.

24) Fig. 4e: This panel is hard to understand. The figure legend suggests it is intended to illustrate how to find target genes of a given master regulator, but I don't see where the master regulator is. Besides, what are the green and gray shades? The SIMBA embedding axis on the bottom left corner indicates three dimensions, which is unnecessary since the plot is visualized in 2D.

We will modify the panel to indicate the location of the master regulator more clearly. We will also modify the figure legend to explain the green and gray shades.

By default, SIMBA generates a 50-dimension cell x feature embedding. We use the three-dimensional axis to indicate that the identification of target genes is performed in the original SIMBA space, as opposed to the 2D UMAP space, which is solely for the purpose of visualization. However, we agree the three-dimensional axis may be confusing, and therefore we will remove it and explain in the legend.

Fig. 6

25) Fig. 6c: To better visualize the multi-feature distributions on the embedding space, it would be helpful to make a separate UMAP for each node category (gene, cell, peaks) with the same coordinates.

We will add new supplementary figures to show separate UMAP for each type of nodes with the same coordinates.

Methods

26) Line 745-746: Regarding the scRNA normalization, it seems that some genes will naturally have higher expression and some will not. Will this create a bias in the learned embeddings and downstream analysis?

We used in SIMBA a widely used normalization procedure (library-size normalization and log-transformation). Also, given that we explicitly consider different bins for genes expression, the resulting embedding should incorporate information for genes expressed at different levels (see lines 775-777). In addition, users can prioritize genes that have higher expression (or low expression) if necessary, by modulating the relation weights for the different bins.

27) Line 893-897: Does the edge weight between two nodes matter during optimization? The degree control used in the negative edge set seems potentially related, but it is not clear how degree is calculated, and how negatives are selected based on two nodes and different edge weights.

The loss is the sum of terms for each (true) edge in the graph, weighted by its edge weight. Each of these terms itself consists of the positive components corresponding to the true edge, and the negative component corresponding to corrupted edges. Therefore, a higher weight on an edge will lead to more accurately assigning higher score to this edge than its corruptions (at the expense of accuracy in lower-weighted edges). We will revise this sentence.

28) Line 904: The loss function is optimized over all types of edges, and it is not shown whether the different types of edges are optimized well. There should be an evaluation for how well the learned embeddings recapitulate known edges in each edge category on a hold out edge set.

The evaluation procedure has been already implemented in the current SIMBA package. The final model is evaluated on a held-out set of edges (by default, 5% of training edges). The evaluation metrics can be visualized with the function “simba.pl.pbg_metrics()”, which provides the training and validation losses individually for each edge type. We will provide this information for our experiments in the revised manuscript.

29) line 942: It seems that the overall loss function includes some of the hyperparameters, so I don't understand how the hyperparameters can be selected based on the loss term. More generally, the model contains multiple hyperparameters. Please list them and show the grid

of hyperparameters tuned.

There is only one parameter, ‘weight decay’ that requires dataset-specific tuning. The weight decay parameter is associated with sample size (the number of edges) and tuned using an automated procedure (see methods, line 906-909). We used the same default setting for all the other hyperparameters across all the experiments. See also response to Reviewer #1 comments 6) and 7).

30) line 1006: The max and gini scores seem somewhat arbitrary; please add citations or intuitions. Furthermore, for the cell-type specificity measurement, the gini index doesn't make sense to me in this context, i.e. the score is affected by i , yet there is no description of how the index i is determined.

We thank the reviewer for this comment and in response, we will more clearly justify the choice of the max and the Gini index for evaluating informativeness and cite the related works that have employed them for similar tasks.

Briefly, our model calculates a probability score of assigning a feature to a cell (as shown in SIMBA barcode plots). Based on the probability distribution of each feature across all cells, the Gini index are used to evaluate the deviation from a perfectly uniform distribution. Thus, features that show an imbalanced distribution (cell- type-specific) are assigned with a higher Gini index. The max score averaging the probabilities of top 50 cells is used to evaluate the highest probability of cell-type-specificity and filter out potentially noisy features.

Based on the equation of the Gini index described in line 1006, the Gini index is not affected by i , which represents cell i . Rather, the Gini score of each feature is determined by the total number of cells, n , which is fixed for each dataset.

Other major comments:

31) The paper does not make a very convincing, systematic case that the co-embedding algorithm of genes, peaks, motifs and k-mers makes sense. Have you validated the gene-peak distance based on gene-peak proximity?

We will expand on the investigation of SIMBA co-embedding results and further validate how gene-peak genomic proximity is reflected in the SIMBA co-embedding space.

32) Please add some biological explanation on what it means if a TF is located far away from its corresponding motif.

We will further improve the biological interpretation of the distance between a TF and its corresponding TF motif in the SIMBA space.

Briefly this may suggest a lag between the expression of a TF and its binding activity in the accessible regions. We know that pioneer factors can bind initially to inaccessible regions and help in recruiting other factors to open these regions. For example, in Figure 4c, the

Hoxc13 gene appears earlier than the Hoxc13 motif in SIMBA co-embedding space. This is in agreement with a previous independent study showing Hoxc13 ability to bind inaccessible motifs ¹⁰.

33) The authors claim that SIMBA does clustering-free analysis (line 26 and in a lot of subsequent texts), which sounds interesting and reasonable. However, almost all of the example studies depend on previous cluster information. Can SIMBA find something surprising compared to previous methods -- e.g. a specific subset of cells driven by some unknown markers, or cell-specific regulation within a cell type?

Currently we have validated SIMBA embeddings of cells and features by referring to previous methods, including both computational methods (e.g., comparison with other computational methods, chromVAR for

confirming TFs and k-mers, etc.) and experimental studies (e.g., the study of human hematopoietic regulatory landscape, the multimodal SHARE-seq study, FACS-sorting labels, etc.).

In the revised manuscript we will further investigate this point by assessing the ability of SIMBA to discover marker features based on their proximity to a specific subset of cells. We will then compare our findings with previous literature to assess the recovered markers.

34) The potential advantage of using graph embeddings on multiple features, besides interpretation, seems to be improving embedding accuracy. Does the performance (cell clustering, marker gene detection, batch correction) improve with more types of features taken into account?

In principle, including multiple features could help with the characterization of cellular heterogeneity and marker detection. We have performed SIMBA analyses using different sets of features for both scATAC-seq data (Supp Figure 4) and multimodal analysis (Supp Figure 6). Qualitatively, we did not observe significant differences. But we are planning to further perform a quantitative evaluation for these analyses and add the discussion of how involving more features may impact on the graph embedding procedure.

35) Their code works fine in anaconda.

We thank the reviewer for testing the SIMBA software and acknowledging its functionality.

=====

Minor comments:

36) What value of k is used for the k-mers?

We used k=6 for k-mers. This value is commonly used for scATAC-seq analysis.

37) Fig 1: bottom right: multi-omics integration figure, the transparent nodes are confusing.

We will further clarify the nodes with different levels of transparency in the legend.

Briefly, in the task of multi-omics, cells are profiled with different techniques. In order to distinguish them, we use opaque nodes to denote cells profiled using one technique and the transparent nodes with opaque edges to denote cells profiled using another technique.

38) line 907: Why normalize the sum of node embeddings by N_e (total number of edges)?

We do not normalize the sum of node embeddings by N_e . Instead, N_e is used to calculate the weight decay parameter (wd). wd is introduced to improve the training performance and to mitigate potential overfitting problems. This parameter is associated with sample size, i.e., the larger the sample size is, the smaller wd we should use. In SIMBA, wd is automatically calculated by taking N_e into account.

References

1. Cortal, A., Martignetti, L., Six, E. & Rausell, A. Gene signature extraction and cell identity recognition at the single-cell level with Cell-ID. *Nat Biotechnol* (2021).
2. Cao, Z.J. & Gao, G. Multi-omics single-cell data integration and regulatory inference with graph-linked embedding. *Nat Biotechnol* (2022).
3. Bilous, M. et al. Metacells untangle large and complex single-cell transcriptome networks. *bioRxiv 2021.06.07.447430* (2022).
4. Zhao, Y., Cai, H., Zhang, Z., Tang, J. & Li, Y. Learning interpretable cellular and gene signature embeddings from single-cell transcriptomic data. *Nat Commun* **12**, 5261 (2021).
5. Tran, H.T.N. et al. A benchmark of batch-effect correction methods for single-cell RNA sequencing data. *Genome Biol* **21**, 12 (2020).

6. Lerer, A. et al. Pytorch-biggraph: A large-scale graph embedding system. *arXiv preprint arXiv:1903.12287* (2019).
7. Wu, L.Y. et al. in Thirty-Second AAAI Conference on Artificial Intelligence (2018).
8. Vandenbon, A. & Diez, D. A clustering-independent method for finding differentially expressed genes in single-cell transcriptome data. *Nat Commun* **11**, 4318 (2020).
9. Ma, S. et al. Chromatin Potential Identified by Shared Single-Cell Profiling of RNA and Chromatin. *Cell* (2020).
10. Bulajić, M. et al. Differential abilities to engage inaccessible chromatin diversify vertebrate Hox binding patterns. *Development* **147**, dev194761 (2020).

Decision Letter, first revision:

8th Aug 2022

Dear Dr Pinello,

Thank you for your letter asking us to reconsider our decision on your Article, "SIMBA: SIngle-cell eMBedding Along with features". We very much appreciate your patience when the editorial team evaluate and discuss this appeal. After careful consideration we have decided that we are willing to consider a revised version of your manuscript that fully addresses all the concerns raised by our reviewers.

- * include a point-by-point response to our referees and to any editorial suggestions
- * please underline/highlight any additions to the text or areas with other significant changes to facilitate review of the revised manuscript
- * address the points listed described below to conform to our open science requirements
- * ensure it complies with our general format requirements as set out in our guide to authors at www.nature.com/naturemethods
- * resubmit all the necessary files electronically by using the link below to access your home page

[REDACTED]

We hope to receive your revised paper within eight weeks. We are very aware of the difficulties caused by the COVID pandemic to the community. If you cannot send it within this time, please let us know. In this event, we will still be happy to reconsider your paper at a later date so long as nothing similar has been accepted for publication at Nature Methods or published elsewhere.

OPEN SCIENCE REQUIREMENTS

REPORTING SUMMARY AND EDITORIAL POLICY CHECKLISTS

When revising your manuscript, please submit reporting summary and editorial policy checklists.

DATA AVAILABILITY

Please include a "Data availability" subsection in the Online Methods. This section should inform readers about the availability of the data used to support the conclusions of your study, including accession codes to public repositories, references to source data that may be published alongside the paper, unique identifiers such as URLs to data repository entries, or data set DOIs, and any other statement about data availability. At a minimum, you should include the following statement: "The data that support the findings of this study are available from the corresponding author upon request", describing which data is available upon request and mentioning any restrictions on availability. If DOIs are provided, please include these in the Reference list (authors, title, publisher (repository name), identifier, year). For more guidance on how to write this section please see:

<http://www.nature.com/authors/policies/data/data-availability-statements-data-citations.pdf>

CODE AVAILABILITY

Please include a "Code Availability" subsection in the Online Methods which details how your custom code is made available. Only in rare cases (where code is not central to the main conclusions of the paper) is the statement "available upon request" allowed (and reasons should be specified).

MATERIALS AVAILABILITY

ORCID

We look forward to hearing from you soon. Thank you for submitting your paper to Nature Methods.

Sincerely,

Lin Tang, PhD
Senior Editor
Nature Methods

Author Rebuttal, first revision:

Response to Reviewers

We really appreciate the thoughtful and helpful comments and suggestions provided by the reviewers and we are grateful for their time and effort in reviewing our work. Their constructive feedback has been invaluable in helping us to improve our manuscript and method.

We revised the manuscript in response to their comments and we believe we addressed their concerns. Briefly, we made significant changes to the main text and supplementary materials, including the addition of two new procedures in SIMBA: modeling of continuous edge weight and statistical significance estimation of recovered edges. We also added 6 new supplementary notes, 13 new supporting figures, and 1 new table, and performed additional benchmarking analyses on additional datasets to compare SIMBA with other methods. We hope the reviewers will appreciate the effort we put into addressing their comments and are willing to make further revisions if necessary.

Reviewer #1

The manuscript by Chen et al proposed a single-cell analysis for multimodal datasets based on graph embedding approaches. The authors demonstrate the applicability of their approach using multiple examples from the analysis of single-cell RNA-seq data, multi-omic analysis, data integration, and batch correction for uni and multiple modalities.

The idea of encoding features (genes, peaks, k-mers, etc) along with cells is an interesting and promising approach that others have recently proposed. The paper's scope and claims are broad and cover a wide range of applications/methods within specialized subfields of single-cell computational methods (software development, data integration, multimodal data analysis). While this shows the approach's potential, it also necessitates the method to be compared with the highly optimized method for each task, which might be challenging. The usability and speed are also crucial for such a general framework which are not

compared currently benchmarked. overall I like the idea of finding multimodal features in a cluster-free way which is exciting and could have been main the focus of the paper; in the current shape, it is not clear why users would prefer SIMBA, over popular frameworks such as Seurat or scanpy, or other specialized software.

We appreciate the reviewer's excitement for our work and acknowledgment of our approach's potential.

To the best of our knowledge, SIMBA is the first method that proposes the concept of co-embedding single- cells and multiple types of features. 'Cell-ID'¹ is a recently-proposed method, which explored the simultaneous representation of cells and genes, but it is limited to a single feature (i.e., genes) and not based on graph embedding. Without explicitly proposing the concept of co-embedding, Cell-ID primarily focuses on gene signature extraction from scRNA-seq data. In contrast, SIMBA encodes both cells and various feature types (including genes, chromatin accessibility, DNA sequences), simultaneously into a large-scale graph, from which the resulting co-embedding is generated. This approach allows us to discover multi-modal batteries of features spanning different molecular levels of regulatory circuits without clustering; currently, no method can achieve this. In addition, we demonstrate that SIMBA, as a more general graph-based approach can accomplish other common single-cell analysis tasks "for free" including batch correction and multi-omics integration.

We have updated our comparisons of SIMBA with other methods in the manuscript. In addition, quantitative metrics to show the advantages of our approach over popular frameworks are now better presented in the manuscript. We have also added extensive benchmarking of computational complexity to address a related, later comment (#2). Taken together, we feel these additions strongly enhance the manuscript and inform readers why they might choose SIMBA over (or, in addition to) other popular frameworks like Scanpy and Seurat.

Major comments:

1) *Others have proposed using knowledge graphs and/or embedding features in multimodal data integration (Cao and Gao 2022) and RNA-seq (Bilous et al. 2021; Zhao et al. 2021). It would be great if at authors put their work in the context of existing work and outline the main novelty of SIMBA.*

We have carefully reviewed the three manuscripts²⁻⁴ suggested by the reviewer and have created a succinct summary table to highlight the differences between these works and SIMBA.

Supplementary Table 1: Comparison of SIMBA with existing methods

Method	Core algorithm	Graph nodes	Node featurization	Requires PCA or LSI as preprocessing steps	Output	scRNA-seq	scATAC-seq	Batch correction (single modality)	Multimodal (paired multi-omics)	Multi-omics integration (unpaired multi-omics)	Software Quality		
											Easy installation (software available in public packaging systems)	Dedicated Documentation website	Comprehensive and self-contained (no dependencies on heavy packages such as Scanpy or Seurat for preprocessing and plotting)
SIMBA	Multi-relation graph embeddings with type constraints	Cell and omics features (including genes, open regions, DNA sequences, etc.)	No	No	Co-embeddings of both cells and features (comparable)	✓	✓	✓	✓	✓	✓	✓	✓
GLUE	Variational auto-encoders	Omics features (including genes, open regions, methylated sites)	No	Yes	Embeddings of cells and embeddings of features (not comparable)	✗	✗	✗	✗	✓	✓	✓	✗
Metacell	k-nearest neighbors	Cells	Yes	Yes	Embeddings of cells	✓	✗	✗	✗	✗	✗	✗	✓
scETM* <small>*not a graph-based method</small>	Variational auto-encoders	NA	NA	No	Embeddings of cells and embeddings of features (not comparable)	✓	✗	✓	✗	✗	✓	✗	✗

Drawing from the information presented in the table, we highlight a few key advantages when comparing SIMBA to existing methods:

- 1) SIMBA does not require node featurization nor ad-hoc transformations.
- 2) The co-embedding of cells and features in a common space generated by SIMBA enables an intuitive interface for inferring biologically meaningful information (e.g., marker genes, regulators, cell circuits).
- 3) While SIMBA is not a dedicated method for specific single-cell analysis tasks such as batch correction or multi-omics integration, it is able to perform comparably well or better than existing state-of-the-art methods that are dedicated to respective tasks. We demonstrate this quantitatively in the manuscript (Supplementary Figs. 10, 19, and 23).
- 4) SIMBA is implemented as a high-quality, comprehensive, and well-documented Python package built on the Anndata structure, and compatible with, Scanpy – the most widely-used Python-based single-cell analysis framework, making it readily accessible to a wide audience.

To address this point in the manuscript, we have expanded the introduction section to acknowledge previous efforts in this area better as well as highlight the novelties and advantages of SIMBA over competing methods.

*“Although there are recent proposals for knowledge-graph-based single-cell embedding methods²⁻⁴, the core algorithm and graph construction procedure underlying SIMBA as well as the resulting embedding and downstream applications offer unique advantages. GLUE², which performs integration on unpaired single-cell multi-omic data, embeds cells into a low-dimension manifold using a variational autoencoder (VAE). This procedure is guided by a knowledge-based graph of omic features as nodes. It outputs cell embedding and feature graph, however the cell and feature embeddings are not comparable. GLUE is unapplicable to use-cases outside of unpaired multi-modal analyses, including single-modality analyses, making it a tool with a very specific use-case. Metacell³ is a scRNA-seq analysis method that uses a k-nearest neighbor (kNN) graph with node featurization to merge highly similar cells into ‘metacells’. It is applicable to only scRNA-seq data and the resulting embedding contains only cells. scETM⁴ is a generative topic model for integrative analysis of scRNA-seq data. It learns embeddings of cells and genes through VAE. Similar to GLUE, its resulting embeddings are not comparable. Similar to Metacell, it can only be applied to scRNA-seq data. Compared to existing methods and summarized in **Table 1**, several key innovations and advantages of SIMBA are most notably highlighted by (1) SIMBA does not require node featurization nor ad-hoc transformations. (2) The co-embedding of cells and features in a common space generated by SIMBA enables an intuitive interface for inferring biologically meaningful information (e.g., marker genes, regulators, cell circuits). (3) While SIMBA is not a dedicated method for specific single-cell analysis tasks, it is able to perform better or comparably well than existing state-of-the-art methods that are dedicated to respective tasks. Finally, (4) SIMBA is implemented as a high-quality, comprehensive, and well-documented Python package built on the Anndata structure and compatible with, Scanpy – the most widely-used Python-based single-cell analysis framework, making it readily accessible to a wide audience.”*

2) As mentioned in the remark, I tested the method on the PBMC (~2.7k cells) example, and the pipeline ran slightly slower than scanpy analysis. However, for bigger datasets (~15k Pancreas), the pipeline was much slower on my laptop compared to scanpy and Seurat. While this is understandable partly due to the back-bone algorithm training of the embedding, most people perform analysis in CPU-based devices/servers. Thus, extensive

benchmarking against popular frameworks in computational complexity and speed would be necessary.

We appreciate the reviewer's thoughtful proposal of benchmarking the computational requirements of SIMBA against other popular frameworks, such as Scanpy and Seurat. To address this suggestion, we performed the proposed benchmark, measuring the running time and memory requirements of SIMBA, Scanpy, and Seurat on three datasets: a ~2,700 cell PBMC dataset (10X Genomics), a ~15,000 cell pancreas dataset (Baron et al., 2016), and a large ~1.3 million cell mouse neuron dataset (10X Genomics). Additionally, we down-sampled the

1.3 million mouse neuron dataset into samples of sizes: 500, 1,000, 5,000, 10,000, 25,000, 50,000, 100,000, 250,000, 500,000, and 1,000,000 cells to resolve a per-cell performance comparison of each method.

Briefly, for all datasets, Scanpy had lower run-time requirements than SIMBA. Run-times required by Seurat were slightly less than or similar to Scanpy. This is especially notable for the progressive down-sampling experiment wherein as the dataset size is increased beyond 25,000 cells, Seurat's run-time requirement ceases to increase and even reduces. Seurat is parameterizable such that trade-offs in memory and run-time enable appropriate scaling. Using the default parameters, the dataset was chunked such that the memory requirement continued to expand while the required run-time experienced a decrease with an increase in dataset size, up to 500,000 cells where Seurat was no longer able to run within the memory constraints of the machine (500 Gb).

Progressive down-sampling of the large 1.3M mouse neuron dataset illustrates that Scanpy consistently required less memory than Seurat though the two methods scale at nearly the same rate upon dataset expansion (**Supplementary Fig. 26d**). In contrast, SIMBA has a much higher baseline memory requirement for even very small ($N=500$ cells) datasets though experiences a sub-linear trend in the maximum memory required upon dataset expansion. After 100,000 cells, SIMBA requires less memory to operate than Seurat and begins to approach the memory requirement trendline of Scanpy. The large maximum memory requirement by SIMBA for small datasets is largely attributable to the training procedure, which learns the embeddings of both cells and genes, as opposed to other tools that learn only cellular states. For a typical dataset, such as the 10x PBMCs (3,000 cells) dataset (**Supplementary Fig. 26a**) or the Pancreas dataset⁵ (**Supplementary Fig. 26b**), Seurat required nearly as much memory as SIMBA, while Scanpy required less than either. In any case, each method is reasonable to use on a machine with $< 4\text{Gb}$ RAM (less than is currently standard for a common laptop computer).

We summarize these results and the benchmarking methodology in an addition to the manuscript as Supplementary Note 3 and Supplementary Fig. 26. Additionally, we have added a succinct description of these results by modifying the paragraph originally on lines 294-298:

“We show that SIMBA does not require variable gene selection, an essential step in standard scRNA-seq pipelines such as Seurat or Scanpy. When tested with or without variable gene selection, SIMBA produced qualitatively similar embeddings (Fig, 2b and

Supplementary Fig. 4e). However, we do observe that variable gene selection improves the efficiency of the training procedure. To better understand the computational complexity of SIMBA as compared to Scanpy and Seurat, we measured the memory as well as the running time required for each method on three different datasets of varying size, including the 10x 1.3M mouse neuron dataset, progressively down-sampled from the full dataset to 500 cells. For all datasets, Scanpy required less time to run than SIMBA and Seurat less time than Scanpy (**Supplementary Note 3**). The training step of SIMBA required more memory than any step in either the Scanpy or Seurat workflow though all three methods required similar total allotments of memory.”

3) *The authors propose using the GINI-max combination to select essential features for each cell type. At the same time, cluster-based methods use DE testing and report statistical significance (e.g. P-value or Bayes factor) to account for false-positive discovery. How would SIMBA account for that? How does it compare to statistical methods?*

The submitted implementation of SIMBA provided ranked features similar to the recent, non-cluster-based method CellID¹. However, we agree with the reviewer's comment. This is an excellent suggestion. Providing the statistical significance of an identified marker feature is an important aspect to account for potential false- positive discoveries and plan experimental validations. Motivated by this concept of an improved statistical framework for feature discovery, we have invested significant effort to revise the manuscript wherein we derive a new procedure to calculate the significance and false discovery rates of the cell state specificity metrics we have previously proposed (Gini, max, entropy, and standard deviation). In the following paragraphs, we provide a short summary of this statistical comparison process. We have also made additions to the Results section as well as the Methods section, which are shown after this short summary.

To assess the statistical significance of a proposed metric, we may consider random permutations of the input graph while preserving the graph statistics (node degree). The resulting null distribution of the metric values is used for calculating statistical significance of the metric values for the original features.

This procedure is implemented by adding to the original graph the “null” feature nodes that have shuffled edges to the cell nodes while preserving the node degree distribution. The updated training procedure learns embeddings of the null feature nodes while preventing the null feature nodes from altering the original embeddings. We describe the full details in the Methods section,

To demonstrate the value of this procedure, we have reanalyzed the 10X PBMC dataset to evaluate the significance of the recovered maker genes and to propose a reasonable threshold. Using an FDR threshold of 0.1 (the red dotted line in the plots below), the known maker genes we have previously highlighted are significant, while housekeeping genes are not, as expected.

We now provide a new tutorial notebook to demonstrate how to use this new implemented function at

[https://github.com/huidongchen/simba_tutorials/blob/main/v1.1sig/rna_10xpmbc_all_genes_with_significan ce.ipynb](https://github.com/huidongchen/simba_tutorials/blob/main/v1.1sig/rna_10xpmbc_all_genes_with_significan_ce.ipynb)

Additions to the **Results** section:

“The statistical significance of the cell type specificity metrics used by SIMBA can be calculated using a statistical procedure that considers random permutations of the input graph while preserving the node degree distribution. (Methods).”

Additions to the **Methods** section:

“The statistical significance of these metrics can be optionally calculated based on the comparison with the metrics derived from “null” feature nodes that have shuffled edges with the same node degree (i.e., the number of edges linked to a node) distribution as the input graph. For example, to obtain the significance of genes’ cell type specificity metrics in scRNA-seq analysis, we construct a graph with the original gene and cell nodes together with the “null” gene nodes that have shuffled edges to the original cell nodes. For indirectly linked DNA sequence features such as TF motifs and k-mers, the shuffle is performed for the edges between the DNA sequence features and peaks. The degree-preserving edge shuffle is achieved by randomly permuting the node index among the nodes with the same degree, separately for source and destination nodes. The degree-preserving shuffle ensures the null metric distribution reflects the bias in edge scores from null nodes to cells solely due to the property of each node while destroying the graph connectivity (i.e., biological information in SIMBA’s input graph).

For the significance calculation, SIMBA learns the embeddings of null nodes together with the cells and features from the original graph. During the training with the null nodes, the loss from the edges involving the null nodes will not affect the real nodes’ embeddings and this is enabled by using the ‘fix’ operator in simba_pbg. For example, the loss from the edge between a real cell and a null gene will not be propagated to the embedding of the real cell. The default number of null nodes is set to 20 times the number of genes for scRNA-seq data and 5 times the number of peaks for scATAC-seq and SHARE-seq data.”

4) *The discretization of RNA expression of n-bins is not justified. This binning might remove*

the biological nuance when cell-type differences are subtle. Authors should assess how changing the current binning strategy might affect results. Examples of exciting scenarios could be genetic, disease perturbation, or highly similar subpopulations.

When constructing a graph of cells and genes, an edge is added between a cell and a gene, if a gene is expressed in a given cell. To distinguish the strength of a cell-gene edge, we used different types of relations to indicate different levels of gene expression. To this end, we proposed a binning procedure that approximates the distribution of non-zero gene expression values. This binning procedure differentiates gene expression levels while preserving the original distribution. Increasing the resolution of gene expression through increasingly granular discretization approaches the original counts distribution (Supplementary Fig. 24). Interestingly however, we observe that increased resolution of expression imparts little effect on the resulting embeddings and the implemented procedure performs consistently well even with a coarse-grained binning (5 bins).

We considered that, despite the above-described results, as the reviewer suggests, there may still be cases wherein binning gene expression values may obscure biological nuance. To fully account for this possibility, we have implemented a new procedure that does not require feature discretization and instead accepts continuous

distributions of features to be encoded as edge weights between nodes. We have tested this new procedure on the 10x PBMCs dataset by creating a graph encoding gene expression values directly as edge weights.

Supplementary Fig. 25 shows that this procedure generates similar embeddings using either original (middle) or discretized (right) gene expression as edge weights compared to the binning procedure (left). This further suggests that our current discretization is effective in capturing biological information and SIMBA is robust to the binning procedure.

The supports of edge weights to model continuous features without discretization has been implemented in SIMBA v1.2 and a new tutorial notebook is provided at https://github.com/huidongchen/simba_tutorials/blob/main/v1.2/rna_10xpmbc_edgeweigts.ipynb.

To document this new procedure and to better justify the original discretization procedure, as suggested by the reviewer, we have expanded the Methods section as follow:

“When constructing a graph of cells and genes, an edge is added between a cell and a gene if a gene is expressed in a given cell. To distinguish the strength of each edge, a binning procedure is proposed to categorize gene expression values into different levels while preserving the original distribution. Different levels of gene expression are encoded by different types of relations.”

“As expected, we observe that the discretized distribution approaches the original distribution as we increase the number of bins. However, increased resolution of expression imparts little effect on the resulting embeddings and the implemented procedure performs consistently well even with a coarse-grained binning (5 bins) (Supplementary Fig. 24)”

“In addition to relation type weights, SIMBA also supports encoding gene expression values directly as edge weights when constructing a graph. Supplementary Fig. 25 shows that this procedure generates similar embeddings using either original (middle) or discretized (right) gene expression as edge weights compared to the binning procedure (left). This further suggests that our current discretization is effective in capturing biological information and SIMBA is robust to the binning procedure. This support of edge weight is implemented in the SIMBA package using the function, “si.tl.gen_graph(add_edge_weights=True)”.

5) I really liked to link to TF/kmers for multiome datasets this could be pretty useful. However, I am not fully convinced why gene/TFs or as stated by the authors "1) the target gene is close to both the TF motif and the TF gene; 2) the accessible regions (peaks) near the target gene loci must be close to both the TF motif and the target TF gene." To the best of my understanding, there is explicit mechanistic or causal modeling linking those different modalities.

We thank the reviewer for recognizing the usefulness of SIMBA in multimodal analysis. We reasoned that the distances between features (TFs, peaks, genes) can be used to investigate their shared gene regulatory context, putative causal relationship, and cell-type-specificity. The condition in 1) prioritizes potential target genes that are co-expressed with the TF and TF binding site is also accessible specifically in the same cellular context. The condition in 2) requires the putative cis-regulatory elements of the potential target gene to correlate with the TF gene expression and motif accessibility specific to the shared cell state. This is motivated by the mechanistic observation that links the TF regulatory activities on the expression of a gene to its binding to specific DNA sequences in proximal accessible regions (e.g., promoters or enhancers). When both assumptions are met, this procedure can recover TF that potentially modulates the expression of a target gene through putative regulatory regions containing the TF motif.

We have expanded the description to further clarify the rationale for this procedure in the Methods section.

“To infer the target genes of a given master regulator, we postulate that in the shared SIMBA embedding space, 1) the target gene is close to both the TF motif and the TF gene, indicating that the expression of the target gene is highly correlated with the expression of the TF and the accessibility of the TF motif in a cell-type-specific way; 2) the accessible regions (peaks) near the target gene loci must be close to both the TF motif and the target TF gene, indicating that the accessibility of the cis-regulatory elements near the target gene locus is highly correlated with the expression of the TF and the accessibility of the TF motif in a cell-type-specific way.”

6) The relation between k-mers and peaks was assigned a weight of 0.02 while the relation between TF motifs was assigned a weight of 0.2. Of note, k-mers and motifs may be used? How and why these numbers are selected?

We reasoned that it is important to consider for our optimization function the informativeness of each type of features and their representation in the graph (i.e., the number of nodes and edges). These weights may be modified depending on how informative a certain type of features is. For example, TF motifs are more informative and less abundant than unfiltered k-mers in explaining TF binding and therefore are assigned with a higher weight. In practice, we selected weights with a grid search (ranging from 0 to 10) on 3 datasets and found that they performed consistently well across 12 other datasets. Therefore, we used the same relation weights in all the analyses shown in the manuscript.

7) *How robust is the training to different parameters? How much hyper-opt is needed for each dataset?*

The only parameter that needs to be tuned in our experiments is the “weight decay”. However, this can be empirically optimized based on the training sample size (i.e., the total number of edges) of the dataset.

Therefore, we provide an automatic procedure to determine the optimal weight decay parameter. This procedure performed consistently well in all the experiments across 5 tasks in 15 datasets, and therefore we believe this is a robust procedure. In addition, we also provide several training metrics (they can be plotted using the function “*si.pl.pbg_metrics()*”) to monitor the training process so the users can further fine-tune the auto-generated weight decay parameter based on these training metrics if needed.

We have incorporated these details into Supplementary Note 9.

*“We have performed a grid search to select the optimal hyperparameters and use the same hyperparameters for all the experiments across 15 datasets. The only parameter that needs to be tuned during the training is the “weight decay”. However, this can be empirically optimized based on the training sample size (i.e., the total number of edges) of the dataset, and therefore we provide an automatic procedure to determine the optimal weight decay parameter. “*si.tl.pbg_train(auto_wd=True)*”. All the experiments across 5 tasks in 15 datasets were performed with this procedure.*

*In addition, we also provide several training metrics (they can be plotted using the function “*si.pl.pbg_metrics()*”) to monitor the training process so the users can further fine-tune the auto-generated weight decay parameter based on these training metrics if needed.”*

8) *There are many conclusions and validation based on 2d UMAPs, which can be misleading. I suggest authors include more quantitative evidence to support their claims.*

We agree that, in general, conclusions should not be drawn from low-dimensional embeddings, especially UMAP projections. Since part of the novelty proposed by SIMBA is inherently displayed as a UMAP (e.g., co-embedding features with cells), we have taken steps to ensure that qualitative observations are corroborated by quantitative evidence. This is made possible by using datasets that are externally annotated with orthogonal information such as FACS labels or expert-annotated tissue labels. These labels of cell state then act as a ground truth and allow us to make conclusions regarding the quality of the embeddings generated by SIMBA.

Qualitative and quantitative evaluation of SIMBA-generated embeddings for scATAC-seq, batch-correction, and multi-omics integration have been performed using externally annotated cell state labels. We have also compared SIMBA against state-of-the-art methods using published benchmark frameworks for each task. (Supplementary Figs. 10, 19, 23)

To supplement and strengthen the existing quantitative evaluations of SIMBA, we have provided a new procedure to validate the cell-type specific marker features identified by SIMBA. This procedure identifies marker features using a Gini-index of expression values across cell type labels. We reasoned that, by definition, a good marker feature is specifically expressed in one or a few cell types and therefore has relatively high inequality, (i.e., higher Gini index) across cell types.

We have applied this new procedure to three datasets: 10x PBMCs scRNA-seq, human hematopoiesis scATAC-seq, and mouse skin SHARE-seq multiome. These datasets provide well-annotated cell types that can be used to assess quantitatively the quality of the recovered marker and it is worth noticing that SIMBA never used this information to learn the embeddings or detect the markers. In fact, SIMBA does not need cell type or cluster labels to detect marker features.

For each dataset we first pick the top marker features identified by SIMBA (top 500 marker genes for scRNA-seq; top 5,000 marker peaks for scATAC-seq). For each marker feature, we then calculated its average expression or accessibility level for each cell type (top marker features are shown in the dot plots as shown in Supplementary Fig. 5). We then calculate the Gini index scores for these marker features and compared them with the “background” features (for gene expression, “background” genes consist of one group of 500 randomly selected genes and one group of house-keeping genes; for chromatin accessibility, “background” peaks consist of 5,000 randomly selected peaks). As shown in Supplementary Fig. 5, the Gini index scores of the marker features identified by SIMBA are significantly higher than “background” features (p -value $< 10^{-4}$, Mann-Whitney-Wilcoxon test) for all four cases.

In sum, this new quantitative procedure and analyses are independent of the UMAP visualizations and further validate the quality of the SIMBA embeddings and marker features identified by SIMBA.

Per the reviewer’s suggestion, we have incorporated the additional evaluation into the **Results** section and Supplementary Note 1:

Results:

“The cell type specificity of the selected marker genes was further confirmed by visualizing their expression pattern on UMAP plots (Fig. 2f and Supplementary Fig. 1d), accompanied by SIMBA barcode plots (Supplementary Fig. 1d) and quantitative validation (Supplementary Note 1, Supplementary Fig. 5a).”

Supplementary Note 1:

“To quantitatively evaluate embeddings generated by SIMBA, we provide a Gini-index-based quantitative procedure to validate the marker features identified by SIMBA. We reasoned that a good marker feature is specifically expressed in one or a few cell types and therefore has relatively high inequality, i.e., a higher Gini index, across cell types. We have applied this procedure to 3 datasets: the 10x PBMCs scRNA-seq dataset, the human hematopoiesis scATAC-seq dataset, and the mouse skin SHARE-seq multiome dataset.

These datasets provide well-annotated cell types that can be used to assess quantitatively the quality of the recovered marker and it is worth noticing that SIMBA never used this information to learn the embeddings or detect the markers. In fact, SIMBA does not need cell type or cluster labels to detect marker features.

For each dataset we first pick the top marker features identified by SIMBA (top 500 marker genes for scRNA-seq; top 5,000 marker peaks for scATAC-seq). For each marker feature, we then calculate its average expression or accessibility level for each cell type (top marker features are shown in the dot plots as shown in Supplementary Fig. 5). We then calculate the Gini index scores for these marker features and compared them with the “background” features (for gene expression, “background” genes consist of one group of 500 randomly selected genes and one group of house-keeping genes; for chromatin accessibility, “background” peaks consist of 5,000 randomly selected peaks).

As shown in **Supplementary Fig. 5**, the Gini index scores of the marker features identified by SIMBA are significantly higher than “background” features (p-value < 10⁻⁴, Mann-Whitney-Wilcoxon test) for all four cases.”

Minor:

9) The current distribution of the code on bioconda seems to be not functional. I tried to

install conda, which failed on both intel and M1 laptops.

We thank the reviewer for testing our software and appreciate that they are considering this during the review process. Unfortunately, we are unable to reproduce this error. In comment 2, we note that the reviewer describes being able to run SIMBA (see: Reviewer 1, Comment 2). Additionally, another reviewer has explicitly stated that “Their code works fine in anaconda.”.

To ensure usability, we have implemented a continuous integration workflow through the SIMBA GitHub repository. This workflow automatically tests each release of SIMBA for both Linux and MacOS:

<https://github.com/pinelloab/simba/actions/workflows/CI.yml>, and all tests passed.

We do additionally understand that new GPUs (M1 and M2) have recently been introduced to Apple computers, which changes how MacOS interacts with previously stable software distributions. To accommodate these developments, along with ever-changing chips, we will keep updating and testing our software to make sure it is functional. If this issue persists, we would appreciate if the reviewer could (anonymously) open an issue on our repo with details of their issues so we can make the software more widely accessible.

10) *The Supplementary figures are pretty busy and with many UMAP plots that are hard to read.*

We thank the reviewer for this suggestion. Accordingly, we have split several previously busy supplementary figures - including original Supplementary Figures 1, 8, and 9 – into multiple figures (Supplementary Figs.

1,2,13,14,15,16). In addition, we would be happy to revise and reformat additional figures as needed.

11) *There are some typos and missing words:*

"Second, SIMBA analysis identified an unbiased set of DNA sequences, i.e., k-mers, that e important ..."

Unfortunately, it appears that the reviewer was probably reading an old version of our manuscript, which had been revised before submission to *Nature Methods*.

This sentence only appeared in our very first bioRxiv version, which was posted on October 18, 2021. Significant changes have been introduced ever since. In our submitted version, this sentence has been changed to “Second, SIMBA revealed an unbiased set of DNA sequences, i.e., k-mers, that represent important TF binding motifs involved in hematopoiesis.” without typos (Line 378-379). Nonetheless, we thank the reviewer for their attention to detail.

Reviewer #2

This paper describes SIMBA, which embeds single cells and their features in a shared space and claims that the method can carry out many different types of single-cell analysis tasks, including working with multi-omic datasets. The method detects marker genes, corrects for batch effects, integrates multi-omics datasets, and generates biological insights such as identifying master regulators. The paper shows an impressive array of applications of the software, some of which is quite convincing and some of which requires more work or better explanations of what was done. In general, the paper does quite a poor job of citing related work.

We are glad that the reviewer found the application of SIMBA to various tasks and numerous datasets impressive. We are also grateful for the suggestions that improved the presentation of our work. We have revised the manuscript significantly to better acknowledge past efforts and further highlight the novelty of our method.

=====

Introduction/overview

1) The paper makes clear that SIMBA is not algorithmically novel, but it does a poor job of providing details about its lineage. Prior work is only mentioned briefly in the discussion. For example, the multi-feature graph embedding idea has been proposed by StarSpace (Wu et al. 2017), which is not cited.

We thank the reviewer for the suggestion. Although SIMBA leverages the PBG framework inspired by StarSpace, several novel crucial procedures, including Softmax transformation, weight decay for controlling overfitting, and entity-type constraints, have to be introduced to tackle the unique challenges in single-cell data. More importantly, SIMBA solves various single-cell problems through a unified procedure, i.e., co-embedding cells and features, in which cells and features are treated as equal entities. This is fundamentally different from existing methods, in which cells and features are considered separately and differently. Therefore, we believe SIMBA is algorithmically novel. We have further expanded the

section Overview of SIMBA. We hope the new edits we have made helps clarify the algorithmic novelty of SIMBA.

“Unlike existing methods that consider cells and features differently and primarily focus on learning cell states, SIMBA treats both cells and features as equal nodes in the same graph and thus solves various single-cell tasks through a unified procedure. Importantly, SIMBA introduces several crucial procedures including Softmax transformation, weight decay for controlling overfitting, and entity-type constraints to generate comparable embeddings (co-embeddings) of cells and features and to address the unique challenges in single-cell data.”

As suggested by the reviewer, we have also expanded the introduction section to better acknowledge previous efforts in this area and to highlight the novelties and advantages of SIMBA over competing methods. See also our response to Reviewer 1 comment 1).

We have also added the references to StarSpace in both the Introduction and Discussion sections.

2) Line 73-74: In some cases there may not be any relationship between features. If each feature is associated with a genomic coordinate, then these coordinates may naturally link features between the two domains. But for non-genomic data (e.g., protein abundances or single-cell imaging data), such linkages may not exist.

This is an excellent point. Fortunately, SIMBA does not require each feature to be associated with a genomic coordinate. For example, in the current implementation for features corresponding to gene expression we only use their numerical expression values, and the genomic coordinates of the corresponding transcript are not necessary.

In general, SIMBA will work if features of different domains can be encoded into a single connected graph either directly or indirectly. Thus, even if a direct edge between two features is not observed in the input graph, these features can still be indirectly linked through intermediate edges with common entities. For examples, for the relations peak-gene and peak-protein abundance (genomic) – or gene-spatial location (non-genomic), these features can still be indirectly linked through cells.

We have revised the **Overview of SIMBA** and **Discussion** sections to clarify

this point. Overview of SIMBA:

“In SIMBA, each cell-feature edge corresponds to a single-cell measurement (e.g., the expression value of a

gene or a chromatin-accessible peak observed in a cell). For example, if a gene is expressed in a cell, an edge is created between the gene and cell. The weight of this edge is determined by the gene expression level.

Similarly, an edge is added between a cell and a chromatin region if the region is open in this cell. Edges are also allowed between different features to capture and model the underlying regulatory mechanisms. For example, an edge between a chromatin region and a TF-motif (or k-mer) captures the notion that a TF may bind to a regulatory region containing a specific DNA sequence. Figure 1 summarizes potential relations represented by edges and the semantics for the analyses presented in this study, namely: (cell-gene): a cell expresses a given gene, (cell-peak): a cell has an accessible chromatin region, (peak-TF motif): a peak sequence contains a significant binding site for a given TF, (peak-kmer): a peak sequence contains a given k-mer sequence.”

Discussion:

“Overall, SIMBA will work as long as features of different domains can be encoded into a single connected graph either directly or indirectly. Thus, even if a direct edge between two

features is not observed in the input graph, these features can still be indirectly linked through intermediate edges with common entities. For examples, for the relations peak-gene and peak-protein abundance (genomic) – or gene-spatial location (non- genomic), these features can still be indirectly linked through cells.”

3) Line 74-75: I don't really understand the critique in these lines. I guess it's related to the point above – is it really the case that all the batch correction methods cited in this paragraph require being able to find feature correspondences?

Although finding feature correspondences is not required for all the batch correction/multi-omics integration analyses, the ability to leverage additional layers of information when such information is available still provides more potential for the information-rich multi-modal analyses. In fact, as demonstrated in a previous study ², incorporating the relations between features could improve single-cell analysis results. Therefore, we believe methods that are able to leverage additional available information in their models are more helpful and should be considered more advanced.

4) Line 77-78: *If the claim in this sentence is true, then it should be backed up with specific citations. “Most batch correction methods” is too vague.*

We have adjusted the sentence to add specific citations to the top-performing methods based on a recent batch- correction benchmark study ⁶.

“... current state-of-the-art batch correction/multi-omics integration methods⁷⁻⁹ ...”

5) *The term “multi-entity” is not clear. All graphs contain multiple entities (multiple nodes and multiple edges). I think the point here is that the graph contains multiple node types. Please clarify this point.*

We thank the reviewer for bringing this concern to our attention. Yes, ‘multi-entity’ here refers to multiple node types. The concept in PyTorch-BigGraph¹⁰ (PBG) is referred to as “multi-relation graph” where nodes and edges can correspond to different semantics. In our case nodes correspond to different domains (cells, gene, peaks) and the edges correspond to different relations i.e., a cell is expressing a gene, or a cell has a given chromatin accessibility peak.

We have revised the sentence to better clarify the term ‘multi-entity’ as follows:

“Unlike existing methods that require featurization of cells, SIMBA directly encodes the cell-feature or feature- feature relations into a large multi-relation (i.e., multiple node and edge type) graph.”

6) Line 92: *“derived from social networking technologies” requires a cite. Which specific algorithm or algorithms is it based on?*

We have added the references to relevant algorithms.

7) Line 99: *What are the semantics of a cell-feature edge? In general, it’s not clear what edge types are allowed and which are not (i.e., can every type of node have an edge to every other type)? Later (line 205), we are told that “cells and genes are connected by edges that embody the relation between them,” without being told what are the semantics of that*

relationship.

In SIMBA, the semantics of each cell-feature edge correspond to a single-cell measurement (e.g., the expression value of a gene or a chromatin-accessible peak observed in a cell). However, edges are also allowed between different features to capture and model the underlying regulatory mechanisms. For example, edges between peaks and TF-motifs (or k-mers) capture the notion that a given TF may bind to a regulatory peak. In general, although it is possible to encode any cell-feature or feature-feature edges into the graph, in this study we focus on exploring the cell-feature edges (or relations) that can be directly measured experimentally and the feature-feature edges that can be linked in a simple and straightforward way, as summarized in Figure 1, namely: (cell-gene): a cell expresses a given gene, (cell-peak): a cell has an accessible chromatin region, (peak-TF motif): a peak sequence contains a significant binding site for a given TF, (peak-kmer): a peak sequence contains a given k-mer sequence. Other types of edges (e.g., peak-peak, or gene-gene) are not explored in this manuscript.

We have expanded on the description of cell-feature edges in SIMBA to clarify the semantics of the relations. See also our response to Comment 2.

“In SIMBA, each cell-feature edge corresponds to a single-cell measurement (e.g., the expression value of a gene or a chromatin-accessible peak observed in a cell). For example, if a gene is expressed in a cell, an edge is created between the gene and cell. The weight of this edge is determined by the gene expression level.

Similarly, an edge is added between a cell and a chromatin region if the region is open in this cell. Edges are also allowed between different features to capture and model the underlying regulatory mechanisms. For example, an edge between a chromatin region and a TF-motif (or k-mer) captures the notion that a TF may bind to a regulatory region containing a specific DNA sequence. Figure 1 summarizes potential relations represented by edges and the semantics for the analyses presented in this study, namely: (cell-gene): a cell expresses a given gene, (cell-peak): a cell has an accessible chromatin region, (peak-TF motif): a peak sequence contains a significant binding site for a given TF, (peak-kmer): a peak sequence contains a given k-mer sequence.”

It’s only in line 217 that we learn that the embedding “correctly embedded informative genes close to relevant cell types.” Here, I am guessing that “informative” means genes that are differentially expressed in a given cell type. But I’m still left wondering what this really means.

Yes, “informative” means features being differentially expressed (or open/enriched for peaks and TF-motifs) in certain cell types. Given that these features are helpful in distinguishing cell types, they are considered “informative” as opposed to non-informative features such as house-keeping genes, which are consistently expressed in all cell types. We have revised the sentence to clarify this important point:

“...but also allows for the discovery of the defining features for each individual cell without relying on a clustering procedure, separating cell-type-specific (informative) features from the non-cell-type-specific (non- informative) features.”

What happens, for example, to genes that are expressed in two or three cell types? The paragraph goes on to explain that non-cell-type specific genes get embedded into the middle, but I'm still left wondering about the interpretability of this kind of embedding in general. It's easy to think of scenarios where, say, three cell types A, B and C are arrayed in a line, and a gene that is expressed in all three ends up getting embedded in the middle cell type (B), giving the impression that it is specific to B.

We thank the reviewer for sharing this concern.

Regarding the interpretation of the embeddings and of informative genes, the 2D UMAP projection of cells and feature is helpful to qualitatively visualize their relations. However, additional information can be provided to the users to interpret cell type specificity of prioritized features in addition to their location. In fact, when external annotations are available users can visually inspect our proposed *barcode plots* for each feature as a diagnostic of cell type specificity, as explained below.

There are three main steps involved in generating the 2D co-embedding space: 1) the raw embeddings of cells and features (by default, 50 dimensions) are generated after PyTorch-BigGraph (PBG) training, in which

embeddings of cells and features are not comparable; 2) the raw embeddings are converted into SIMBA co- embeddings of cells and features (by default, 50 dimensions) through Softmax transformation, in which embeddings of cells and features are comparable; 3) SIMBA co- embeddings of cells and features are visualized using UMAP (by default, 2 dimensions) (see methods).

The obtained 2D visualization can present to users a visual interpretation of the cell type specificity of features. This is particularly effective if features are highly cell type specific (i.e., relevant only for a single cell type).

However, some rare artifacts may emerge if all the features are embedded in the 2D UMAP as for the scenario proposed by the reviewer.

However, this scenario in practice may rarely happen in the 50-dimensional SIMBA embeddings as opposed to a 2-dimensional plane. Nonetheless, using the SIMBA barcode plot (e.g., Fig. 2d) we can directly plot and investigate the probability of assigning a feature to each cell in this more expressive space. In this scenario (shown in the figure below), all three cell types will show high probability for this gene and will be evenly mixed in the barcode plot (top panel), as opposed to the case of being cell type B -specific, in which only cell type B will show high probability (bottom panel). The 2D embedding serves as a qualitatively and useful way to visualize both cells and features but it also has its limitations just as other general dimensionality reduction methods. Therefore, we suggest users inspect SIMBA barcode plots to further validate and more quantitatively assess cell type specificity.

8) Line 139-142: Give citations for the “recent graph embedding techniques.”

We have added the missing citations.

9) In general, it would help to clarify the source of the algorithms in SIMBA. It seems, for example, that many of the ideas here come from the node2vec paper. You are up front about the fact that SIMBA is not novel, in the sense that it's borrowing ideas from other fields like social network analysis, but the citation chain should be clearer.

This is an important suggestion for clarifying the innovations proposed by SIMBA. Our method leverages the PyTorch-BigGraph (PBG) framework. In its own manuscript¹⁰ PBG was compared against node2vec-like related work and its superior performance has been demonstrated. Our manuscript is focused on adapting PBG to single-cell analysis and solving various single-cell tasks in an innovative and unified way. As mentioned in

comment 1), due to the unique challenges in modeling single-cell data, in SIMBA, we have introduced multiple crucial and algorithmically innovative changes and procedures, including Softmax transformation, weight decay to control overfitting, entity-type constraints, and novel visualizations and analyses. These procedures introduced in SIMBA play a crucial role in modelling single-cell data and generating comparable embeddings (co-embeddings) of cells and features, which general graph embedding methods alone cannot achieve.

We have incorporated these details to further highlight the algorithmic novelty of SIMBA. See also our response to Comment 1.

“Unlike existing methods that consider cells and features differently and primarily focus on learning cell states, SIMBA treats both cells and features as equal nodes in the same graph and thus solves various single-cell tasks through a unified procedure. Importantly, SIMBA introduces several crucial procedures including Softmax transformation, weight decay for controlling overfitting, and entity-type constraints to generate comparable embeddings (co-embeddings) of cells and features and to address the unique challenges in single-cell data.”

10) Line 180: Suddenly we are told that each edge has an associated probability. This was a surprise to me. Is this supposed to be a probabilistic framework? If so, what is the semantics of the edge probability? Later, in line 228, we learn that we can get the “estimated probability of assigning a feature to a cell by SIMBA based on the recovered edge confidence,” which implies perhaps that the edge probabilities have a particular semantics. How this semantics is enforced is not clear to me.

We thank the reviewer for pointing this out. The graph embedding algorithm can be thought of as producing a model (whose parameters are the graph embeddings themselves) that approximates the likelihood of an edge (u, v) existing in the graph. These likelihoods are what we refer to when we talk about “estimated probabilities”.

We have clarified the semantics of edge probability where it is first mentioned.

“In fact, the relationship between cells and features can be explored directly through their proximity in the SIMBA embedding as the distance between embedded nodes reflects their

edge probability, which is the likelihood of an edge existing in the graph and is informative of the potential importance of a feature to a cell and the interplay between features”.

Fig. 2

12) For the cell-type specific gene marker identification, the authors should carry out a more systematic evaluation on cell type marker gene selection. Currently SIMBA is only compared against scanpy. It would be more convincing to compare with methods that are developed for this particular task, such as Cell-ID and Vandenbon et al. 2020.

This is a great suggestion. SIMBA is a clustering-free method that doesn't require cell type definition to identify marker genes, as the two methods the reviewer suggested.

Cell-ID¹ and singleCellHaystack¹¹ (Vandenbon et al. 2020) identify marker genes at different levels. Cell-ID identifies marker genes for each cell, while singleCellHaystack identifies marker genes at the dataset level. As a result, we have compared the corresponding functionalities of SIMBA with these two tools separately below.

Cell-ID identifies cell-specific marker gene signatures without a significance measure (by default, it selects the top 200 genes), then uses this list of genes to predict for each cell in the input dataset the most likely cell type based on a reference dataset. The identification of cell-specific marker genes corresponds to SIMBA's functionality of finding marker genes for each cell based on the proximity of their embeddings. To fairly compare the accuracy of the cell-specific markers identified by Cell-ID and SIMBA, we used the same procedure proposed by Cell-ID to predict cell types using the marker genes identified by each method.

To this end, we compared the cell type prediction accuracy of the two methods on the Baron et al. human pancreas scRNA-seq dataset which has been used also in the CellID's tutorial (<https://rauselllab.github.io/CellID//vignettes/vign.html>). To predict cell types, we compared the marker genes identified by Cell-ID or SIMBA against the reference marker genes of all human cell types provided by the Panglao database¹². This process is similar to the one described in the Cell-ID tutorial, but it takes into account the different numbers of marker genes per cell identified by each method. We note that although the cell type labels from the original study are defined using clustering solutions derived by scRNA-seq, the prediction accuracy of predicting pancreas-specific cell types against all human reference cell types is a valid evaluation. In this comparison, SIMBA consistently shows better cell type prediction accuracy than CellID, especially when a small number of marker genes per cell is used

(Supplementary Fig. 6a-b).

singleCellHaystack identifies marker genes that have significantly different expression levels within a specific region of the cell embedding compared to the rest of the embedding. This functionality corresponds to the marker feature discovery using SIMBA metrics for cell type specificity.

Using the 10x PMBC scRNA-seq dataset, we've compared the cell type specificity of the marker gene sets identified by singleCellHayStack and SIMBA with different significance

thresholds. As both methods do not use the cell type information to define the marker genes, we can use the cell type labels of the original study as the ground truth to evaluate the marker features' cell type specificity. Similar to the procedure proposed in Reviewer 1 Comment 8, here the cell type specificity of the marker genes is calculated by the Gini index of the average log-normalized gene expression values per cell type. SIMBA consistently identified marker genes with better cell type specificity compared to singleCellHaystack (Supplementary Fig. 6c).

: $1.00e-04 < p \leq 1.00e-03$, *: $p \leq 1.00e-04$, ns: not significant, Mann-

Whitney-Wilcoxon test We have added the comparison results in the main text, and

Supplementary Note 2.

“We also compared SIMBA with two recent clustering-free marker gene detection methods, Cell-ID¹ and singleCellHaystack¹¹. Both Cell-ID and SingleCellHaystack are applicable to only scRNA-seq data. Cell-ID identifies cell-specific marker genes to annotate cell types whereas singleCellHaystack identifies dataset-level marker genes. SIMBA consistently showed better cell type prediction accuracy than Cell-ID and higher cell type specificity than SingleCellHaystack (Supplementary Fig. 6, Supplementary Note 2).”

The scripts to reproduce the above analyses are available at https://github.com/pinellolab/simba_comparison.

13) What is the red horizontal line in Fig. 2d?

We thank the reviewer for pointing this out. As the four genes have different scales of probability scores, we use the value of $1e-3$ as a line of reference to visually compare them. We have incorporated this detail into the legend.

“The orange dashed line indicates the probability score of $1e-3$.”

14) Line 289: I am not convinced by this argument, since it is not stated whether IL7R and FCER1A are good cell type markers.

We thank the reviewer for the suggestion. In the manuscript we mentioned that we use the same set of marker genes as proposed by the authors of Scanpy, including *IL7R* and *FCERIA* to annotate cell types (Line 218-219 in the original manuscript). In addition, we have added relevant references in support of these two genes as cell type markers.

Fig. 3

14) The idea (line 309) of embedding peaks and DNA sequences was previously put forth by the BindSpace paper (Yuan Nature Methods 2019). In general, the methods section should clarify which steps are taken from previous methods and which are novel to SIMBA.

Bindspace was built on the TagSpace model (sequence embedding) from Starspace¹³, wherein each sampled DNA sequence is treated as a sentence, each k-mer is treated as a word, and TFs are treated as hashtags (there are no cells involved in this context). It is a non-graph-based model and is limited to tasks that can be converted into this strict sentence-word-tag setting. Moreover, Bindspace was only used to analyze high-throughput sequencing SELEX (HT-SELEX) data and has not been proven to be able to perform any single-cell tasks.

In contrast, SIMBA is built on a graph-based model (graph embedding) that allows for multi-type entities and multi-type relations, making it flexible to handle any tasks that can be encoded as a graph. This enables cells to be linked to both genes and accessible chromatin regions, and an accessible chromatin region to be linked to TF motifs/k-mers within this region. Therefore, SIMBA can be used to co-embed both cells and various types of features, while Bindspace can only embed sequence features. Also, by introducing multiple innovative procedures in this manuscript, SIMBA is able to generate comparable co-embeddings of cells and features and an unified procedure to perform various single-cell tasks with better or comparable performance against current state-of-the-art methods. In addition, SIMBA leverages the Pytorch-BigGraph framework¹⁰ and it is significantly more efficient and scalable than the original StarSpace framework¹³, which is internally used by BindSpace.

Last but not least, Bindspace is available as a set of Bash and R scripts while SIMBA is implemented as a comprehensive and practical Python package and a dedicated documentation website is built to demonstrate its usability.

Therefore, we believe that SIMBA is fundamentally differently from BindSpace both algorithm-wise and task-wise. Regardless, we have acknowledged BindSpace in applying sequence embedding to modeling transcription factor binding preferences in the Discussion section:

“Neural network embeddings hold significant promise for the analysis of biological data. Previous applications of embedding models include functional annotation of genes¹⁴, modeling transcription factor binding preferences^{13, 15} and more recent single-cell RNA-seq analyses^{16, 17}.”

15) Line 311: The fact that SIMBA uses Boolean features rather than TF-IDF features is not obviously a good (or bad) thing. The TF-IDF transformation is not computationally expensive. A priori, it's certainly plausible that switching to a Boolean representation could cause a significant loss of information. Also, please include a cite for scATAC-seq pipelines that use TF-IDF.

We agree with the reviewer that the TF-IDF transformation is effective and computationally efficient. In a previous study¹⁸, we benchmarked different computational methods for scATAC-seq data and compared various featurization procedures, including TF-IDF and binarization. Although TF-IDF could be used in SIMBA, in place of binarization, we wanted to show that without using advanced transformation procedures such as TF-IDF SIMBA is still able to achieve superior performance to other methods. We would like to emphasize that SIMBA's ability to achieve the best performance with a simple binarization highlights the advantages of the graph embedding procedure used by SIMBA.

As requested, we added a reference to scATAC-seq pipelines that use TF-IDF.

16) Line 315: *I like that, in this case, the semantics of the edge types are clearly specified.* ☑

We are glad that the reviewer liked the way edge types were explained.

17) Line 330: *This sentence, I think, requires some evidence that the blobs in the picture actually correspond to distinct cell types. It is certainly true that SIMBA separates the cells into distinct clusters, but the claim that these correspond to distinct cell types requires further analysis.*

As mentioned in the manuscript, cell types were defined using fluorescence-activated cell sorting (FACS) in the original study and therefore in Fig. 3b, SIMBA embeddings of cells are colored by FACS labels instead of cluster labels. So given the fact that these cell types are well distinguished, we believe it is fair to state that SIMBA successfully separated cells belonging to distinct cell types. While this is mentioned in the manuscript, we have revised the sentence for increased clarity, as follows:

*“For the SIMBA embeddings of cells alone, as shown in **Fig. 3b**, SIMBA accurately separated cells such that cells belonging to distinct cell types (defined based on FACS labels) are visually distinguished.”*

18) *The analysis on lines 371-404 does a good job of showing the potential utility of the SIMBA co-embedding with this particular dataset.*

We are glad that the reviewer liked this part.

19) I am not fully convinced by the examples in Figure 3. How are the peaks selected? They d'n't seem to be highly prioritized based on the cell-type specific calculation. In addition, the peaks used as examples here are not very close to GATA1, and in fact, they seem to belong to two different cell subpopulations. Do peaks and k- mers located close to GATA1 in Fig. 3C make sense instead?

The goal of Fig. 3 is to assess embeddings of cells and features using prior biological knowledge, as opposed to the discovery of novel features. Therefore, the peaks were selected based on the genomic locus of the *KLF1* gene. Instances of MEP-specific regulatory elements and gene regulation are supported by previous studies described in the original manuscript (Line 391-404). We therefore expect MEP-specific regulatory elements and their corresponding genes to be embedded nearby MEP cells. We observed that these features are all embedded in the neighborhood of MEP cells (and far from other cell types) supporting the quality of the SIMBA embeddings.

As the reviewer suggested, we also further investigated the peaks and k-mers located close to the GATA1 TF in SIMBA embedding space. The selected neighboring peaks and k-mers are highlighted in **Supplementary Fig. 8a**. We show in **Supplementary Fig. 8b** that these peaks are preferentially open in the MEP cells and in a subset of the CMP cells that are either mixed with or adjacent to the main MEP cell subpopulation. We also show in **Supplementary Fig. 8c** that the normalized read counts of the selected k-mers are higher in MEPs than the other cell types. Furthermore, **Supplementary Fig. 8d** shows that the five k-mers closest to GATA1 all resemble the GATA1 TF motif. We believe these additional analyses demonstrate the effectiveness of the SIMBA embedding procedure.

We have revised the sentences to further clarify this point, and have incorporated additional details and analyses into the result section describing the scATAC-seq analysis.

“For example, the two peaks near the genomic locus of the KLF1 gene, with coordinates chr19:12997999- 12998154 (P1) and chr19:12998329-12998592 (P2) were embedded within MEP cells and almost exclusively observed in MEP cells on KLF1 genome track (Fig. 3e).”

*“We also investigated the peaks and k-mers that are embedded close to GATA1. The selected neighboring peaks and k-mers are highlighted in **Supplementary Fig. 8a**. We observe that these peaks are open in all MEP cells and part of CMP cells, which are either mixed with or adjacent to MEP cells (**Supplementary Fig. 8b**). We also observe that these kmers are highly enriched in MEP cells (**Supplementary Fig. 8c**) and resemble the GATA1 TF motif (**Supplementary Fig. 8d**).”*

20) In Figure 3d, please change the scatter plot to a density plot or use transparent dots. Also, what is the cutoff for cell type specific features and how is it determined?

We thank the reviewer of the suggestion. We have overlaid contour lines (representing density) on the scatter plots.

Instead of using a hard cutoff to determine marker features, SIMBA provides cell-type specificity ranks for each feature, using a procedure that is based on the distance between features and cells in the SIMBA embedding space. All known marker features are ranked highly by the SIMBA metric plots (Fig. 3d).

In addition, in the revised manuscript we have introduced a statistical procedure to assess the statistical significance of each feature so that a statistical test cutoff can be specified. Please see also our response to Reviewer 1, Comment 3.

Fig. 4

21) Line 456-462 & Fig. 4b: Lef1's gene/motif/peaks don't seem cell type specific. In addition, it is unclear how Lef1 is linked to the hair follicle as claimed.

Lef1 was one of the important genes described in the original SHARE-seq paper (Ma et al., 2020). Therefore, we tried to recapitulate its biological role in skin development based on our embedding. Notably, *Lef1* is automatically recovered as a key developmental gene based on our master-regulator ranking procedure (see Fig. 4d). As shown in Supplementary Fig. 12a, although *Lef1* shows a Hair Shaft-cuticle cortex-specific pattern, it is also partially expressed in several other cell types. Therefore, it is not surprising that *Lef1* doesn't have the highest cell-type specificity score. This demonstrates SIMBA's ability to discover master regulator TFs, corroborated through queries of both gene expression and motif accessibility within the cell type of interest in the SIMBA embedding space.

22) Fig. 4c: How are the arrows drawn? Why are Lef1 peaks, gene and motif somewhat far away from each other? How are Lef1 peaks selected among all peaks?

The arrows represent likely differentiation trajectories and were manually drawn for illustrative purposes based on previous SHARE-seq literature.

Despite the distance between the peaks, genes, and motifs of *Lef1*, they are all embedded in the neighborhood of cells transitioning from TAC2 to Hair shaft.

In addition, we highlighted the peaks (Peak1(Lef1), Peak2(Lef1), Peak3(Lef1)) near the genomic locus of *Lef1* in the SIMBA embedding. Interestingly, we observe that these peaks are embedded before the *Lef1* gene along the hair shaft-cuticle cortex developmental trajectory. Given that the gene *Lef1* is regulated by the binding of up-stream transcription factors and it is expressed throughout the differentiation into cuticle/cortex, this is consistent with what was described in the original study ¹⁹, suggesting a delay between the chromatin accessibility of peaks within a gene locus and its expression.

As mentioned in Line 476-477 in the original manuscript, Lef1 peaks were selected based on the genomic proximity to the Lef1 locus.

23) Fig. 4d: The “rank score” needs to be further explained.

We thank the reviewer for this suggestion. The rank score is explained under “Identification of master regulators” in the Method section. To clarify this concept, we have also incorporated the details into the legend of Fig.4d.

“(d) Ranked master regulators identified by SIMBA. The rank score indicates the rank of a given TF gene among all genes (including non-TFs) based on the distance from the genes to the selected TF motif in the SIMBA embedding space.”

24) Fig. 4e: This panel is hard to understand. The figure legend suggests it is intended to illustrate how to find target genes of a given master regulator, but I don’t see where the master regulator is. Besides, what are the green and gray shades? The SIMBA embedding axis on the bottom left corner indicates three dimensions, which is unnecessary since the plot is visualized in 2D.

We thank the reviewer for this suggestion. We have expanded the legend of Fig.4e to indicate the location of the master regulator and explain the green and gray shades.

By default, SIMBA generates 50-dimension embeddings of cells and features. Three axes are commonly used to represent high-dimensional space and therefore we used the three-dimensional axis to indicate that the identification of target genes is performed in the original high-dimensional SIMBA space, as opposed to the 2D UMAP space, which is solely for the purpose of visualization. We have made the edits in the legend to clarify this point.

“(e) Schematic description of SIMBA’s strategy for identifying target genes given a master regulator in the high-dimensional SIMBA embedding space (by default, 50 dimensions). The master regulator is represented by a TF motif (a star filled in green) and a TF gene (a square without borders filled in green). A colored shade indicates a SIMBA embedding area of a gene, containing a gene (a square with borders) and peaks (a triangle) near the genomic locus of the gene. Green shades indicate areas of target genes, and gray shades indicate areas of non-target genes.”

Fig. 6

25) Fig. 6c: To better visualize the multi-feature distributions on the embedding space, it would be helpful to make a separate UMAP for each node category (gene, cell, peaks) with the same coordinates.

We have added a new supplementary figure (Supplementary Fig. 20) to show separate UMAP plots for each type of nodes with the same coordinates.

Methods

26) Line 745-746: Regarding the scRNA normalization, it seems that some genes will naturally have higher expression and some will not. Will this create a bias in the learned embeddings and downstream analysis?

For scRNA-seq analysis, in SIMBA we use a standard normalization procedure - i.e., library-size normalization and log-transformation and therefore the normalization should not create a bias. The different gene expression levels after normalization are preserved during the embedding procedure through either (1) our binning strategy or (2) the newly-introduced continuous edge weights (see our response to Reviewer 1, Comment 4). So, we believe our procedure naturally captures the relative expression of different genes (and intensity of other features) in the learned embedding. Thus, we do not expect the introduction of a bias due to gene normalization or the embedding procedure.

27) Line 893-897: Does the edge weight between two nodes matter during optimization? The degree control used in the negative edge set seems potentially related, but it is not clear how degree is calculated, and how negatives are selected based on two nodes and different edge weights.

The loss is the sum of terms for each (true) edge in the graph, weighted by its edge weight. Each of these terms is minimized when the score for the edge is higher than that of a set of corruptions of the edge. Therefore, a higher weight on an edge will lead to more accurately assigning higher score to this edge than its corruptions (at the expense of accuracy in lower-weighted edges). So, edge weights do not affect the selection of the negative edges but are used to scale the loss calculated from the true and sampled negative edges. The degree of nodes of the input graph is used for the node-degree-preserving sampling of 100 negative edges, as described in the original Methods section (line 890-897).

To better clarify this point we have revised the **Methods** section as follow:

*“For an edge $e = (u, v)$, we denote $s_e = \theta_u * \theta_v$ as the score for e , and optimize a multi-class log loss*

$$\mathcal{L} = -\log \frac{\exp(s_e)}{\sum_{e' \in \mathcal{N}} \exp(s'_{e'})} w_e$$

*Where \mathcal{N} is a set of “negative sampled” candidate edges²⁰ generated by corrupting e and w_e is the edge weight, which is the relation weight by default but can vary by edge within each relation type. For example, edges between cells and genes can be encoded as a single relation with varying edge weights encoding normalized gene expression level (see **Graph construction i. in Methods**).”*

28) Line 904: The loss function is optimized over all types of edges, and it is not shown whether the different types of edges are optimized well. There should be an evaluation for how well the learned embeddings recapitulate known edges in each edge category on a hold out edge set.

We thank the reviewer for this suggestion. We agree that reporting loss over each edge category during training will provide more detailed evaluation for the learnt embeddings. Unfortunately, this will require significant changes to the framework of PyTorch-BigGraph, which is maintained by the Facebook AI Research (FAIR) group, and therefore it is difficult to address in a reasonable timeframe. But we are open to incorporating the edge-type-specific evaluation into a future version of the SIMBA package.

29) line 942: *It seems that the overall loss function includes some of the hyperparameters, so I don't understand how the hyperparameters can be selected based on the loss term. More generally, the model contains multiple hyperparameters. Please list them and show the grid of hyperparameters tuned.*

We have performed a grid search to select the optimal hyperparameters and use the same hyperparameters for all the experiments. See response to Reviewer 1 Comments 6 and 7. The only parameter that needs to be tuned based on our experiments is the “weight decay”. However, this can be empirically optimized based on the training sample size (i.e., the total number of edges) of the dataset, and therefore we provide an automatic procedure to determine the optimal weight decay parameter. All the experiments across 5 tasks in 15 datasets were performed with this procedure, and therefore we believe this is a robust procedure. In addition, we also provided several training metrics (they can be plotted using the function “`si.pl.pbg_metrics()`”) to monitor the training process so the users can further fine-tune the auto-generated weight decay parameter based on these training metrics, if needed.

We have incorporated these details into Supplementary Note 9.

“We have performed a grid search to select the optimal hyperparameters and use the same hyperparameters for all the experiments across 15 datasets. The only parameter that needs to be tuned during the training is the “weight decay”. However, this can be empirically optimized based on the training sample size (i.e., the total number of edges) of the dataset, and therefore we provide an automatic procedure to determine the optimal weight decay parameter. “`si.tl.pbg_train(auto_wd=True)`”. All the experiments across 5 tasks in 15 datasets were performed with this procedure.

In addition, we also provide several training metrics (they can be plotted using the function “`si.pl.pbg_metrics()`”) to monitor the training process so the users can further fine-tune the auto-generated weight decay parameter based on these training metrics, if needed.”

30) line 1006: *The max and gini scores seem somewhat arbitrary; please add citations or intuitions. Furthermore, for the cell-type specificity measurement, the gini index doesn't*

*make sense to me in this context,
i.e. the score is affected by i , yet there is no description of how the index i is determined.*

We thank the reviewer for the suggestion. We have expanded the text to justify the choice of the max and the Gini index and cited the related works that have employed them for similar tasks.

Briefly, our model calculates a probability score (represented as a dot product) of assigning a feature to a cell. Based on the distribution of probability scores, or its derived probability distribution for each feature across all cells (as shown in SIMBA barcode plots), the Gini index is calculated from a Softmax-transformed probability distribution to evaluate the deviation from a perfectly uniform distribution. Thus, features that show an imbalanced distribution (cell-type-specific) are assigned with a higher Gini index. The max score averaging the normalized probabilities of the top 50 cells serves as a metric of confidence towards cell-type assignment and aids in filtering noisy features. These metrics have been previously used to select cell-type-specific genes, including those of rare cell types.²¹.

Based on the equation of the Gini index originally described in line 1006, the Gini index is unaffected by i , which represents cell i . Rather, the Gini score of each feature is determined by the total number of cells, n , which is fixed within a dataset.

We have expanded the section **Metrics to assess cell-type specificity** as follows:

“SIMBA calculates a probability score (represented as a dot product) of assigning a feature to a cell and therefore generates a probability distribution of all cells for each feature. Based on this or its derived probability distribution (as shown in SIMBA barcode plots), four metrics are proposed to assess the cell type specificity of each feature from different aspects. The max score²¹ averaging the normalized probabilities of the top 50 cells serves as a metric of confidence towards cell-type assignment and aids in filtering noisy features. The Gini index²¹ is calculated from a Softmax-transformed probability distribution to evaluate the deviation from a perfectly uniform distribution and thus features that show an imbalanced distribution (i.e., cell-type-specific) are assigned with higher Gini index values. Standard deviation measures the amount of variation in the probability distribution, and a high value indicates a higher deviation within the cells (i.e., cell-type-specific). Entropy measures the information content and capture to what extent the cells are spread out over a Softmax-transformed probability distribution; a lower entropy indicates the distribution is nearly concentrated on one subset of cells (i.e., cell-type-specific). We observe that these four metrics generally give consistent results. For each SIMBA metric plot, by default, the Gini index is plotted against the max value.”

Other major comments:

31) The paper does not make a very convincing, systematic case that the co-embedding algorithm of genes, peaks, motifs and k-mers makes sense. Have you validated the gene-peak distance based on gene-peak proximity?

We appreciate this comment for its creativity. What the reviewer suggests, i.e. to validate the gene-peak distance based on gene-peak proximity, is currently outside the scope of SIMBA’s graph embedding procedure. The position of features in the embedding generated by SIMBA reflects cell-type specificity. Features that are exclusive to a subset of cells are

embedded with those cells, while non-cell type specific features are relegated to the middle of the embedding.

Genomic coordinates of features and the genomic distance between genes and peaks are not encoded in the graph. Within the SIMBA embedding procedure, cells are used as a “reference map” to which other features are then mapped. This procedure prioritizes cell type specific cell-feature relationships rather than relationships between features with nearby genomic coordinates.

However, as the reviewer suggested, we also investigated how the genomic proximity between genes and peaks is reflected in the SIMBA co-embedding space using the SHARE-seq multiomic dataset. In Response Fig. 1, we highlight marker genes including *Wnt3*, *Top2a*, *Krt27*, *Hoxc13*, and *Lef1* and their respective proximal peaks (in genomic coordinate space). Each peak is colored by genomic distance as well as by cell type specificity metrics including max value and Gini index. As shown in the left panel, we do not observe clear patterns reflected in the co-embedding with respect to the proximity of genomic coordinates. In contrast, cell type specific peaks (as indicated by the Gini Index and max value metrics) are embedded within specific cell clusters, while non-specific peaks are embedded without respect to cell type.

This comment inspires us to explore the potential extension of SIMBA to consider genomic coordinates as an encodable property and investigate the example proposed by the reviewer wherein one might be interested in encoding the genomic proximity of a peak to a gene. We speculate if such investigations may aid in three- dimensional genome organization analyses.

Response Figure 1. UMAP visualization of SIMBA embeddings of cells along with genes,

peaks, TF motifs, and k-mers. Genes including *Wnt3*, *Top2a*, *Krt27*, *Hoxc13*, and *Lef1* and their respective proximal peaks are highlighted. Genes are indicated with arrows. Peaks are indicated with large dots and are colored by genomic distance (left), max value (middle), and Gini index (right) respectively. Cells are indicated with small dots and colored by cell type.

32) Please add some biological explanation on what it means if a TF is located far away from its corresponding motif.

We have added the biological interpretation of the distance between a TF and its corresponding TF motif in the SIMBA space.

“The distance between a TF and its motif may suggest a lag between the expression of a TF and its binding activity in the accessible regions. For example, pioneer factors can bind initially to inaccessible regions and help in recruiting other factors to open these regions. In Figure 4c, the Hoxc13 gene appears at an earlier developmental stage than the Hoxc13 motif in the SIMBA co-embedding space. This agrees with a previous independent study showing Hoxc13 ability to bind inaccessible motifs²².”

33) The authors claim that SIMBA does clustering-free analysis (line 26 and in a lot of subsequent texts), which sounds interesting and reasonable. However, almost all of the example studies depend on previous cluster information. Can SIMBA find something surprising compared to previous methods -- e.g. a specific subset of cells driven by some unknown markers, or cell-specific regulation within a cell type?

We thank the reviewer for asking this question; upon consideration, many high-profile findings that use single-cell measurements and analysis report the discovery of previously undescribed heterogeneity, thus making this an important point of a required analysis capability.

In the current manuscript we have validated the quality of the SIMBA embeddings of cells and features by assessing its ability to recapitulate known biological information reported by previous studies using both computational methods (e.g., comparison with other computational methods, chromVAR for confirming TFs and k-mers, etc.) and experimental studies (e.g., the study of human hematopoietic regulatory landscape, the multimodal SHARE-seq study, FACS-sorting labels, etc.).

To address the reviewer’s point, which would not necessarily be addressed by recapitulation of known biology, we break assessment of these capabilities into general scenarios:

1. Discovery of potentially novel marker features based on their proximity to a specific subset of cells in the resulting SIMBA embedding. The cell locations may be arbitrarily defined or derived from a clustering solution or some external information.
2. Discovery and characterization of a subset of cells based solely on their proximity to a SIMBA- identified marker feature, independent of any clustering solution.

To demonstrate SIMBA's proficiency in the first scenario, we reanalyzed the 10x PBMCs scRNA-seq dataset. We then identified the centroid of the CD4 T cell population in the SIMBA embedding space. From the CD4 T cell centroid, we selected the top 500 nearest neighbor genes. This procedure recovers three "new" marker genes *UBASH3A*, *ITGA6*, *IL6ST* (**Response Figure 2**) that are not detected using Scanpy, which uses clustering and differential expression analysis. We next confirmed the biological significance of these three new marker genes, discovered by SIMBA but not by Scanpy's clustering and differential expression analysis. *UBASH3A* is a negative regulator of T cell receptors (TCRs). The expression of *UBASH3A* has been assessed in its regulatory mechanism in CD4+ T cells²³; Integrin alpha6 (*ITGA6*) is expressed more in CD4+ cells²⁴; Interleukin 6 (IL-6) plays a role in the generation of functional memory CD4+ T cells²⁵.

Response Figure 2: UMAP visualization of SIMBA embeddings of cells colored by cell type and gene expression respectively.

To demonstrate the second point, we highlight that SIMBA discovered a master regulator *Relb* and its target genes *Nfkbie*, *Nfkbia*, and *Nfkb1* in the mouse skin SHARE-seq dataset (**Fig.4d, Supplementary Fig. 7**). *Relb* was not reported in the original study but literature supports that this hair-follicle-specific master regulator plays an important role during the hair follicle development²⁶⁻²⁸. Importantly, by querying the neighboring cells of *Relb* in the SIMBA embedding space, it is possible to identify the cell subpopulations with high activity of *Relb*. This helps refine the classification of the cell types based on the global clustering solution proposed by alternative methods.

We show that a distinct cell subpopulation with high *Relb* activity in both *Relb* expression and *Relb* motif accessibility can be identified based on the proximity of cells to *Relb* motifs and genes (**Supplementary Fig. 28a**). This subpopulation with high *Relb* activity mainly corresponds to the subpopulation of TAC-2 in the original annotation, revealing the potentially unreported heterogeneity within the defined cell type (**Supplementary Fig. 28c-d**). We further showed the coordinated regulation of *Relb* and its target genes within the subpopulation (**Supplementary Fig. 28 e-f**). This example clearly shows that SIMBA can uncover cell state from a master regulator without clustering and leverage the co-embedding of cells and features to discover a novel subset of cells with co-expressed gene modules. In principle, any marker feature with high SIMBA cell type specificity scores can serve as a ‘location of interest’ in the SIMBA embedding space to query its proximal cells for further investigation by the users.

We again thank the reviewer for prompting us to improve the manuscript with these analyses. We have incorporated it into the Results section and documented the results in **Supplementary Note 6**.

Results:

*“Moreover, SIMBA identified a previously unreported master regulators *Relb*, and a novel *Relb*⁺ cell subpopulation within TAC-2 cells (**Supplementary Fig. 28** and **Supplementary Note 6**).”*

34) The potential advantage of using graph embeddings on multiple features, besides interpretation, seems to be improving embedding accuracy. Does the performance (cell clustering, marker gene detection, batch correction) improve with more types of features taken into account?

We agree with the reviewer’s suggestion; in principle, the inclusion of additional biologically distinguishing features could enhance one’s ability to resolve cellular heterogeneity and detect marker features. As batch correction has only been performed and benchmarked for scRNA-seq data, which contain a single feature (i.e., genes), we focused on evaluating the performance of SIMBA in marker gene detection and cell clustering.

For marker gene detection, we have compared the top 1,000 cell-type-specific genes identified by SIMBA using genes only and using both peaks and genes on the multiomic SHARE-seq data. We observed that ~70% (688) genes are mutually identified between the two analyses, indicating that SIMBA can identify roughly similar marker genes whether a single feature or multiple features are used as input.

For cell clustering, we have performed SIMBA analyses using different sets of features for scATAC-seq data (Supplementary Fig. 9). Qualitatively, these analyses do not indicate significant advantages gained by the inclusion of additional features. However, we supplement and corroborate this qualitative observation with additional quantitative analyses, wherein we supplied two sets of inputs to SIMBA: (1) peaks and (2) peaks and sequences (as in Supplementary Fig. 9), then performed clustering, which is orthogonal to

SIMBA's embedding solution and unrelated to its feature discovery module. We applied three common clustering methods: hierarchical clustering, K-means clustering, and Louvain clustering to each resulting embedding. We then, as in Supplementary Figure 10, use ARI, AMI, and Homogeneity as metrics to evaluate the clustering quality, compared to the ground truth labels, for each cell. Clustering resolution measured by these metrics is a previously implemented method¹⁸ by which we may judge the quality of the embedding generated by SIMBA with respect to each set of input features and their ability to resolve cell types. We applied this workflow to three real scATAC-seq datasets, 10x PBMCs (5k cells), a human hematopoiesis dataset with FACS-based cell type labels²⁹, and a sub-sampling of the sciATAC-seq mouse atlas³⁰, and two synthetic bone marrow ("clean" and "noisy") scATAC-seq datasets. For all datasets and evaluation metrics, only minimal or inconsistent differences were observed, indicating that there is negligible impact on the embeddings generated by SIMBA upon the inclusion of sequences as additional features (Supplementary Fig. 27). We speculate that there is inherent redundancy in the information contained by the peaks and their sequence contents, and that deeper investigation of feature contribution may be needed in future work. The results are shown in the following figure (Supplementary Fig. 27) and described in an addition to the manuscript via Supplementary Note 5. Finally, we have added the following description of the results to the manuscript by modifying the paragraph originally on lines 406-411:

"Our analyses show that SIMBA, overall, outperforms contemporary methods for scATAC-seq analysis, further demonstrating the wide utility of SIMBA (Supplementary Fig. 10; Supplementary Note 3). We also show that

there is negligible impact on the embeddings of cells generated by SIMBA upon the inclusion of sequences as additional features (Supplementary Figs. 9, 27, Supplementary Note 5)."

35) Their code works fine in anaconda.

We thank the reviewer for testing the SIMBA software and acknowledging its functionality.

=====

Minor comments:

36) What value of k is used for the k -mers?

We used $k=6$ for k -mers. This value is commonly used for scATAC-seq analysis.

37) Fig 1: bottom right: multi-omics integration figure, the transparent nodes are confusing.

We thank the reviewer for their feedback. We denoted the single-cell profiling technology used

through opacity. We have clarified the description of nodes in the legend (below).

“(Right) Common single-cell analysis tasks that may be accomplished using SIMBA. Different opacity levels indicate cells of different experimental batches or single-cell modalities. Solid lines indicate experimentally measured edges. Dashed lines indicate computationally inferred edges.”

38) line 907: Why normalize the sum of node embeddings by N_e (total number of edges)?

We do not normalize the sum of node embeddings by N_e . Rather, N_e is used to calculate the weight decay parameter (wd). wd is introduced to improve the training performance and to mitigate potential overfitting problems. This parameter is associated with sample size, i.e., the larger the sample size is, the smaller wd we should use. In SIMBA, wd is automatically calculated as a function of N_e .

References:

1. Cortal, A., Martignetti, L., Six, E. & Rausell, A. Gene signature extraction and cell identity recognition at the single-cell level with Cell-ID. *Nat Biotechnol* (2021).
2. Cao, Z.J. & Gao, G. Multi-omics single-cell data integration and regulatory inference with graph-linked embedding. *Nat Biotechnol* (2022).
3. Bilous, M. et al. Metacells untangle large and complex single-cell transcriptome networks. *BMC Bioinformatics* **23**, 336 (2022).
4. Zhao, Y., Cai, H., Zhang, Z., Tang, J. & Li, Y. Learning interpretable cellular and gene signature embeddings from single-cell transcriptomic data. *Nat Commun* **12**, 5261 (2021).
5. Baron, M. et al. A single-cell transcriptomic map of the human and mouse pancreas reveals inter-and intra-cell population structure. *Cell systems* **3**, 346-360. e344 (2016).
6. Tran, H.T.N. et al. A benchmark of batch-effect correction methods for single-cell RNA sequencing data. *Genome Biol* **21**, 12 (2020).
7. Stuart, T. et al. Comprehensive Integration of Single-Cell Data. *Cell* **177**, 1888-1902 e1821 (2019).
8. Welch, J.D. et al. Single-Cell Multi-omic Integration Compares and Contrasts Features of Brain Cell Identity. *Cell* **177**, 1873-1887 e1817 (2019).
9. Korsunsky, I. et al. Fast, sensitive and accurate integration of single-cell data with Harmony. *Nat Methods* **16**, 1289-1296 (2019).
10. Lerer, A. et al. Pytorch-biggraph: A large-scale graph embedding system. *arXiv preprint arXiv:1903.12287* (2019).
11. Vandenbon, A. & Diez, D. A clustering-independent method for finding differentially expressed genes in single-cell transcriptome data. *Nat Commun* **11**, 4318 (2020).
12. Franzén, O., Gan, L.-M. & Björkegren, J.L. PanglaoDB: a web server for exploration of mouse and human single-cell RNA sequencing data. *Database* **2019** (2019).
13. Wu, L.Y. et al. in Thirty-Second AAAI Conference on Artificial Intelligence (2018).
14. Ietswaart, R., Gyori, B.M., Bachman, J.A., Sorger, P.K. & Churchman, L.S. GeneWalk identifies relevant gene functions for a biological context using network representation learning. *Genome Biol* **22**, 55 (2021).
15. Yuan, H., Kshirsagar, M., Zamparo, L., Lu, Y. & Leslie, C.S. BindSpace decodes transcription factor binding signals by large-scale sequence embedding. *Nat Methods* (2019).

16. Li, H., Xiao, X., Wu, X., Ye, L. & Ji, G. scLINE: A multi-network integration framework based on network embedding for representation of single-cell RNA-seq data. *J Biomed Inform* **122**, 103899 (2021).
17. Buterez, D., Bica, I., Tariq, I., Andrés-Terré, H. & Liò, P. CELLVGAE: AN UNSUPERVISED SCRNA-SEQ ANALYSIS WORKFLOW WITH GRAPH ATTENTION NETWORKS. *bioRxiv 2020.12.20.423645v1* (2020).
18. Chen, H. et al. Assessment of computational methods for the analysis of single-cell ATAC-seq data. *Genome Biology* **20**, 241 (2019).
19. Ma, S. et al. Chromatin Potential Identified by Shared Single-Cell Profiling of RNA and Chromatin. *Cell* (2020).
20. Kadlec, R., Bajgar, O. & Kleindienst, J. Knowledge base completion: Baselines strike back. *arXiv preprint arXiv:1705.10744* (2017).
21. Jiang, L., Chen, H., Pinello, L. & Yuan, G.-C. GiniClust: detecting rare cell types from single-cell gene expression data with Gini index. *Genome biology* **17**, 144 (2016).
22. Bulajić, M. et al. Differential abilities to engage inaccessible chromatin diversify vertebrate Hox binding patterns. *Development* **147**, dev194761 (2020).
23. Yamagata, K. et al. IL-6 production through repression of UBASH3A gene via epigenetic dysregulation of super-enhancer in CD4+ T cells in rheumatoid arthritis. *Inflammation and Regeneration* **42**, 1-14 (2022).
24. Schweighoffer, T., Luce, G.E.G., Tanaka, Y. & Shaw, S. Differential expression of integrins $\alpha 6$ and $\alpha 4$ determines pathways in human peripheral CD4+ T cell differentiation. *Cell Adhesion and Communication* **2**, 403-415 (1994).
25. Nish, S.A. et al. T cell-intrinsic role of IL-6 signaling in primary and memory responses. *elife* **3** (2014).
26. Bellet, M.M., Zocchi, L. & Sassone-Corsi, P. The RelB subunit of NF κ B acts as a negative regulator of circadian gene expression. *Cell cycle* **11**, 3304-3311 (2012).
27. Krieger, K. et al. NF- κ B participates in mouse hair cycle control and plays distinct roles in the various pelage hair follicle types. *Journal of investigative dermatology* **138**, 256-264 (2018).
28. Gugasyan, R. et al. The transcription factors c-rel and RelA control epidermal development and homeostasis in embryonic and adult skin via distinct mechanisms. *Molecular and cellular biology* **24**, 5733-5745 (2004).
29. Buenrostro, J.D. et al. Integrated Single-Cell Analysis Maps the Continuous Regulatory Landscape of Human Hematopoietic Differentiation. *Cell* **173**, 1535-1548 e1516 (2018).

30. Cusanovich, D.A. et al. A Single-Cell Atlas of In Vivo Mammalian Chromatin Accessibility. *Cell* **174**, 1309- 1324 e1318 (2018).

Decision Letter, second revision:

Our ref: NMETH-A48860B

1st Feb 2023

Dear Dr. Pinello,

Thank you for submitting your revised manuscript "SIMBA: SIngle-cell eMBedding Along with features" (NMETH-A48860B). It has now been seen by the original referees and their comments are below. In light of our reviewers' comments, we'll be happy in principle to publish it in Nature Methods, pending minor revisions to satisfy the referees' final requests and to comply with our editorial and formatting guidelines.

After discussion within the editorial team, when revising the paper, please add qualitative comparison/discussion about the competing methods noted by the reviewers. Quantitative comparison would be nice but is not mandatory for publication. Please also discuss unique use cases of SIMBA though addition of new analysis/data is also not a prerequisite for publishing the paper.

TRANSPARENT PEER REVIEW

ORCID

Sincerely,

Lin Tang, PhD
Senior Editor
Nature Methods

Reviewer #1 (Remarks to the Author):

I thank the authors for the comprehensive revision and detailed answers to my remarks and comments. While the authors did a great job addressing those, I still have some remaining concerns and comments about the uniqueness of the algorithm (as picked up by the other reviewers) compared to existing methods.

Since the time of the submission and revision, a few papers are highly related are out or as a preprint; the first one is cell-space(Tayyebi, Pine, and Leslie 2022), a very similar algorithm (graph-based) requiring prior knowledge of TF motifs (as opposed to SIMBA) since these two approaches are very very similar (although Simba runs on multimodal). They also compare to SIMBA and show comparisons that there might be better performers than SIMBA. This is important since, in my opinion, the primary usage of SIMBA will be in the analysis of scRNA-seq&ATACseq or ATAC-seq. Related to previous comments, I strongly advise authors to technically compare their underlying algorithms against cell-space/starspace since both use PBG. What is the apparent novelty algorithmically, apart from softmax and other additions? While SIMB performs multimodal integration, knowing the exact novelties and being transparent is still essential. I would like to see an algorithmic comparison section in detail.

-The second exciting approach published in Nature methods is MIRA(Lynch et al. 2022). MIRA can integrate multimodal data, which probabilistically models regulatory factors. How does SIMBA compare to MIRA?

-Finally, while you have shown the comparison between SIMBA, scanpy, and Seurat, I still find the scale, extendability, and software quality of those software packages at another level which stems from years of development and active user base feedback. Lastly, according to the time and memory comparison, there is no clear advantage over that method. Yet, the clear advantage of SIMBA is feature-level embeddings which non of the software packages provide. I suggest authors find a use case if they find a biological discovery (or analysis) that could not be achieved with both those algorithms and other existing ones (uniqueness)

Lynch, Allen W., Christina V. Theodoris, Henry W. Long, Myles Brown, X. Shirley Liu, and Clifford A. Meyer. 2022. "MIRA: Joint Regulatory Modeling of Multimodal Expression and Chromatin Accessibility in Single Cells." *Nature Methods* 19 (9): 1097–1108.

Tayyebi, Zakieh, Allison R. Pine, and Christina S. Leslie. 2022. "Scalable Sequence-Informed Embedding of Single-Cell ATAC-Seq Data with CellSpace." *bioRxiv*.
<https://doi.org/10.1101/2022.05.02.490310>.

Reviewer #2 (Remarks to the Author):

I would like to thank the authors for their effort to address my comments. Here are my remaining comments in response to the revisions (listed based on reviewer 2's original indices):

- 1) I couldn't find StarSpace cited in the paper. As far as I can tell, the only mention of it is in the acknowledgments section.
- 2) It needs to be made clear that SIMBA's multi-omics integration relies on feature correspondence across modalities and hence may not work when integrating single-cell RNA-seq and single-cell imaging datasets, when there is no prior knowledge about gene and image feature correlations. The authors stated that "in the current implementation for features corresponding to gene expression we only use their numerical expression values, and the genomic coordinates of the corresponding transcript are not necessary," and it appears in Figure 6A that there is no feature correspondence needed. But in the corresponding description (line 608-611), it turns out that gene activity scores calculated by summing accessible regions to genes from scATAC-seq data are used to align scATAC-seq and scRNA-seq cells.
- 3) I still don't see a clear critique of previous methods. It seems that the authors first argue that one of the downsides of previous batch correction methods is that they don't incorporate multiple feature types, but the batch correction results shown in this paper (Figure 5) only use gene and cell information. Also, the statement in Line 77-81 seems incorrect: multiple methods (e.g. scVI) do not require preselection of gene markers in each batch before batch correction.
- 34) It seems a little bit unsatisfactory that incorporating sequence information makes the clustering performance of scATAC-seq decrease when using Louvain clustering, which is a common clustering method in single-cell analysis.

Regarding supplementary Table 1, the authors should include scGLUE (Gao et al. 2022), which works on scRNA-seq and scATAC-seq integration using a prior knowledge graph. Can the authors compare SIMBA with scGLUE in terms of multi-omics integration performance?

Author Rebuttal, second revision:

Reviewer #1:

I thank the authors for the comprehensive revision and detailed answers to my remarks and comments. While the authors did a great job addressing those, I still have some remaining concerns and comments about the uniqueness of the algorithm (as picked up by the other reviewers) compared to existing methods.

Since the time of the submission and revision, a few papers are highly related are out or as a preprint; the first one is cell-space(Tayyebi, Pine, and Leslie 2022), a very similar algorithm (graph-based) requiring prior knowledge of TF motifs (as opposed to SIMBA) since these two approaches are very very similar (although Simba runs on multimodal). They also compare to SIMBA and show comparisons that there might be better performers than SIMBA. This is important since, in my opinion, the primary usage of SIMBA will be in the analysis of scRNA-seq&ATACseq or ATAC-seq.

Related to previous comments, I strongly advise authors to technically compare their underlying algorithms against cell-space/starspace since both use PBG. What is the apparent novelty algorithmically, apart from softmax and other additions? While SIMB performs multimodal integration, knowing the exact novelties and being transparent is still essential. I would like to see an algorithmic comparison section in detail.

-The second exciting approach published in Nature methods is MIRA(Lynch et al. 2022). MIRA can integrate multimodal data, which probabilistically models regulatory factors. How does SIMBA compare to MIRA?

We thank the reviewer for suggesting these two methods. We did not include CellSpace[1] and MIRA[2] in the past version of our manuscript because they only came out while SIMBA was under revision. In fact, both CellSpace and MIRA cited SIMBA to acknowledge our contributions in their respective tasks.

The key differences between SIMBA and StarSpace-based methods, including CellSpace and BindSpace[3], have been described in detail in our response to the previous round of peer review (**Reviewer #2, Comment 14**). It is worth noting that while CellSpace uses StarSpace, SIMBA is built on PytorchBigGraph (PBG). The superior performance and scalability of PBG have been demonstrated in its own manuscript. Additionally, in contrast to CellSpace that directly applies StarSpace without further modification of its model, SIMBA has introduced several critical new procedures beyond the PBG model including the Softmax transformation, mitigation of overfitting through weight decay, and entity-type constraints, to tackle the unique challenges inherent to single-cell data.

MIRA is specifically designed for multimodal data analysis and utilizes a variational autoencoder to generate embeddings of cells by concatenating the low-dimensional representations of cells from scRNA-seq and scATAC-seq data. It is important to note that MIRA exclusively focuses on learning cell embeddings and does not support analyses based on co-embeddings of cells and features as in SIMBA.

In summary, we propose a generalizable graph embedding framework that focuses on learning node embeddings of a graph constructed from single-cell data. SIMBA explicitly learns low-dimensional representations of cells and features, and implicitly enables the possibility of clustering-free marker discovery, batch effect removal and multi-omics integration. SIMBA can also readily extend to new modalities and tasks and its generalizability and extensibility set it apart from existing methods. We are excited about the potential of SIMBA, and we believe that it will be an asset to the single-cell research community.

As suggested by the reviewer, to highlight the differences between SIMBA and the two additional methods we have expanded both the **Introduction** section and **Supplementary Table 1** as follows:

“CellSpace [1] is a scATAC-seq embedding method based on TagSpace, a non-graph-based model from StarSpace[4]. In this model sampled DNA sequences are represented as sentences with k-mers treated as words; cells are represented as hashtags. CellSpace is designed solely for scATAC-seq data, and the resulting embeddings contain cells and a single-type feature, i.e., DNA sequences. MIRA [2] is a multimodal analysis method that uses a VAE to learn embeddings of cells by concatenating low-dimensional representations of cells from scRNA-seq and scATAC-seq data. MIRA is designed for only single-cell multimodal data and the resulting embeddings contain only cells.”

-Finally, while you have shown the comparison between SIMBA, scanpy, and Seurat, I still find the scale, extendability, and software quality of those software packages at another level which stems from years of development and active user base feedback. Lastly, according to the time and memory comparison, there is no clear advantage over that method. Yet, the clear advantage of SIMBA is feature-level embeddings which non of the software packages provide. I suggest authors find a use case if they find a biological discovery (or analysis) that could not be achieved with both those algorithms and other existing ones (uniqueness)

We thank the reviewer for recognizing the novelty of SIMBA, i.e., generating embeddings of features, which none of current methods can achieve. We also acknowledge that in terms of software performance, SIMBA may not retain an advantage in running time or memory consumption, compared to the most widely used single-cell analysis packages such as Scanpy or Seurat.

We would like to emphasize that the focus of this manuscript centers around the development of a new single cell embedding method. We have demonstrated the advantages and novelty of SIMBA across multiple tasks and datasets. While this manuscript is not primarily oriented towards software development, we have provided a comprehensive and efficient Python package for various types of single-cell data analyses using graph embedding. It enables a seamless interaction between building graphs, training with PyTorch, and post-training analysis.

Our package is built on the commonly used AnnData data structure and therefore is compatible with popular single-cell tools like Scanpy. We also built a dedicated website (<https://simba-bio.readthedocs.io/>) with detailed documentation and tutorials to ensure practicality and easy adoption.

As SIMBA involves neural network training, during which it learns not only embeddings of cells but also embeddings of features, it is expected that it may require more time than Scanpy or Seurat, which neither performs neural network training nor learns feature-level embeddings. Aside from the aforementioned training step, SIMBA is actually comparably fast, memory efficient, and scalable to both Scanpy and Seurat, even for a large dataset with more than one million cells (Supplementary Figure 26). It is worth noting that we do acknowledge room for improvement and optimization of our software. Scanpy and Seurat have benefited from years of optimization with the support of large user bases. As SIMBA is adopted by the community, we anticipate feedback that will enable such optimization of both the method and the supporting software.

In regard to a potential use case of SIMBA for biological discovery, we believe our response to **Reviewer #2, Comment 33** in the previous round of peer review clearly illustrated a compelling example that was presented also in the new Supplementary Note 6. Briefly, this new analysis demonstrated how biological query in the SIMBA co-embedding space can be used for the discovery of both new marker features and cells populations.

Reviewer #2:

I would like to thank the authors for their effort to address my comments. Here are my remaining comments in response to the revisions (listed based on reviewer 2's original indices):

1) I couldn't find StarSpace cited in the paper. As far as I can tell, the only mention of it is in the acknowledgments section.

StarSpace (reference 13) has already been cited in multiple sections of our manuscript, including the **Introduction** (Line 93), **Results** (Line 165), and **Discussion** (Line 680).

2) It needs to be made clear that SIMBA's multi-omics integration relies on feature correspondence across modalities and hence may not work when integrating single-cell RNA-seq and single-cell imaging datasets, when there is no prior knowledge about gene and image feature correlations. The authors stated that "in the current implementation for features corresponding to gene expression we only use their numerical expression values, and the genomic coordinates of the corresponding transcript are not necessary," and it appears in Figure 6A that there is no feature correspondence needed. But in the corresponding description (line 608-611), it turns out that gene activity scores calculated by summing accessible regions to genes from scATAC-seq data are used to align scATAC-seq and scRNA-seq cells.

We thank the reviewer for sharing this concern. SIMBA is a general graph embedding method that can embed nodes of any connected graph. A connected graph may be constructed by adding edges in two ways:

- 1) Experimentally measured edges such as those between a cell and a gene, indicating gene expression in the cell, or between a cell and a chromatin region, indicating the region is open in that cell. Edges may also be added between a chromatin region and a TF-motif, indicating the presence of a transcription factor binding motif within that region.
- 2) Computationally inferred edges, such as edges between cells of different batches or modalities, indicating functional similarity between cells. Although these edges cannot be directly measured, they can be inferred computationally by summarizing features of the same or different types through the procedure we have introduced (see the sub-section **Graph Construction** in the **Methods** section).

For the integration of multi-omics data, we constructed two sub-graphs from scRNA-seq and scATAC-seq data separately and connected them through computationally inferred edges between cells. In our experiments, we have explored both approaches and we observed that the second approach, i.e., via computationally inferred edges, performs better in mitigating batch effects. As a result, the feature correspondence across modalities was not explicitly used to add edges when building the graph. Instead, they were summarized and modularized to infer the edges between cells of different modalities.

We appreciate the reviewer's suggestion regarding the integration of single-cell RNA-seq and single-cell imaging datasets. While beyond the scope of this manuscript, in the discussion section, we mention the spatial proximity between cells in imaging can be potentially employed to add edges between cells. In addition, genes can also serve as shared nodes to connect single-cell RNA-seq and imaging datasets. Therefore, it is possible to build a connected graph and utilize SIMBA graph embedding to integrate single-cell RNA-seq imaging datasets. SIMBA provides a generalizable framework, and it can readily extend to new single-cell modalities. But additional experiments and effort would be required to fully explore this foreseeable extension.

We have expanded the **Overview of SIMBA** section for improved clarity, as follows:

“In SIMBA, edges may be added in two ways: 1) measured experimentally; 2) inferred computationally. For edges that are measured experimentally ...”

*“For edges that cannot be directly measured, they are inferred computationally by summarizing features of the same or different types (**Methods**). Each edge between cells of different batches or modalities indicates the cellular functional or structural similarity.”*

“(cell-cell): cells of different batches or modalities are functionally or structurally similar.”

3) I still don't see a clear critique of previous methods. It seems that the authors first argue that one of the downsides of previous batch correction methods is that they don't incorporate multiple feature types, but the batch correction results shown in this paper (Figure 5) only use gene and cell information. Also, the statement in Line 77-81 seems incorrect: multiple methods (e.g. scVI) do not require preselection of gene markers in each batch before batch correction.

We thank the reviewer for the comment. We would like to clarify that our assertion regarding SIMBA's ability to incorporate multiple features refers to its potential for multi-modality and cross-modality analysis, in general. The batch correction of scRNA-seq datasets presented in Figure 5, which is based solely on a single-type feature (genes). In Figure 4, where we perform a multimodal analysis, it is evident that SIMBA effectively leverages the relationships between multiple types of cellular features, such as chromatin regions and DNA sequences. SIMBA is additionally well-suited for batch correction of two or more multimodal datasets. In this context, SIMBA can directly use both the inter-feature and inter-cell

relationships. This feature distinguishes SIMBA from existing methods, which may not be able to take advantage of these relationships.

To ameliorate any confusion or misinterpretation regarding lines 77-81, we here clarify that our statement was not intended to suggest pre-selection of marker genes for batch correction, and we fully agree that almost no methods require such pre-selection. Instead, as we noted in the manuscript, these lines refer to the standard process of marker gene detection after batch correction has been performed. Existing batch correction and multi-omics integration methods typically identify marker features through a two-step process: (1) detect marker features in each batch/modality and (2) concatenate them. We believe Lines 77-81 accurately describe a common approach to identifying marker genes following batch correction. However, we are willing to revise this statement if the confusion persists.

4) It seems a little bit unsatisfactory that incorporating sequence information makes the clustering performance of scATAC-seq decrease when using Louvain clustering, which is a common clustering method in single-cell analysis.

We appreciate the reviewer's comment. As highlighted in a prior benchmark study on computational scATAC-seq methods [5], the performance of clustering methods may vary depending on the space in which they are used. In order to address this potential source of bias, we followed the previously published scATAC-seq benchmarking framework [5] and employed three distinct clustering methods (k-means, hierarchical clustering, and Louvain) to evaluate the quality of cell embeddings. We believe that our results here are consistent with the previous study.

Regarding supplementary Table 1, the authors should include scGLUE (Gao et al. 2022), which works on scRNA-seq and scATAC-seq integration using a prior knowledge graph. Can the authors compare SIMBA with scGLUE in terms of multi-omics integration performance?

We would like to clarify that scGLUE (Gao et al. 2022) is already listed in Supplementary Table 1 under the name "GLUE," and was compared to SIMBA in the **Introduction** section. In the original manuscript, it used the name "GLUE" instead of "scGLUE". To maintain consistency, we also use the name "GLUE" in our manuscript.

1. Tayyebi, Z., A.R. Pine, and C.S. Leslie, *Scalable sequence-informed embedding of single-cell ATAC-seq data with CellSpace*. bioRxiv 2022.05.02.490310, 2022.
2. Lynch, A.W., et al., *MIRA: joint regulatory modeling of multimodal expression and chromatin accessibility in single cells*. Nat Methods, 2022. **19**(9): p. 1097-1108.
3. Yuan, H., et al., *BindSpace decodes transcription factor binding signals by large-scale sequence embedding*. Nat Methods, 2019.
4. Wu, L.Y., et al., *Starspace: Embed all the things!* Thirty-Second AAAI Conference on Artificial Intelligence, 2018.
5. Chen, H., et al., *Assessment of computational methods for the analysis of single-cell ATAC-seq data*. Genome Biology, 2019. **20**(1): p. 241.

Final Decision Letter:

26th Apr 2023

Dear Dr Pinello,

I am pleased to inform you that your Article, "SIMBA: SIngle-cell eMBedding Along with features", has now been accepted for publication in Nature Methods. Your paper is tentatively scheduled for publication in our July print issue, and will be published online prior to that. The received and accepted dates will be 4th Apr 2022 and 26th Apr 2023. This note is intended to let you know what to expect from us over the next month or so, and to let you know where to address any further questions.

Once your paper is typeset, you will receive an email with a link to choose the appropriate publishing options for your paper and our Author Services team will be in touch regarding any additional information that may be required.

Please note that *Nature Methods* is a Transformative Journal (TJ). Authors may publish their research with us through the traditional subscription access route or make their paper immediately open access through payment of an article-processing charge (APC). Authors will not be required to make a final decision about access to their article until it has been accepted. [Find out more about Transformative Journals](https://www.springernature.com/gp/open-research/transformative-journals)

Authors may need to take specific actions to achieve [compliance with funder and institutional open access mandates](https://www.springernature.com/gp/open-research/funding/policy-compliance-faqs). If your research is supported by a funder that requires immediate open access (e.g. according to [Plan S principles](https://www.springernature.com/gp/open-research/plan-s-compliance))

then you should select the gold OA route, and we will direct you to the compliant route where possible. For authors selecting the subscription publication route, the journal's standard licensing terms will need to be accepted, including [self-archiving policies](https://www.springernature.com/gp/open-research/policies/journal-policies). Those licensing terms will supersede any other terms that the author or any third party may assert apply to any version of the manuscript.

Your paper will now be copyedited to ensure that it conforms to Nature Methods style. Once proofs are generated, they will be sent to you electronically and you will be asked to send a corrected version within 24 hours. It is extremely important that you let us know now whether you will be difficult to contact over the next month. If this is the case, we ask that you send us the contact information (email, phone and fax) of someone who will be able to check the proofs and deal with any last-minute problems.

If, when you receive your proof, you cannot meet the deadline, please inform us at rjsproduction@springernature.com immediately.

Once your manuscript is typeset and you have completed the appropriate grant of rights, you will receive a link to your electronic proof via email with a request to make any corrections within 48 hours. If, when you receive your proof, you cannot meet this deadline, please inform us at rjsproduction@springernature.com immediately.

Once your paper has been scheduled for online publication, the Nature press office will be in touch to confirm the details.

Content is published online weekly on Mondays and Thursdays, and the embargo is set at 16:00 London time (GMT)/11:00 am US Eastern time (EST) on the day of publication. If you need to know the exact publication date or when the news embargo will be lifted, please contact our press office after you have submitted your proof corrections. Now is the time to inform your Public Relations or Press Office about your paper, as they might be interested in promoting its publication. This will allow them time to prepare an accurate and satisfactory press release. Include your manuscript tracking number NMETH-A48860C and the name of the journal, which they will need when they contact our office.

About one week before your paper is published online, we shall be distributing a press release to news organizations worldwide, which may include details of your work. We are happy for your institution or funding agency to prepare its own press release, but it must mention the embargo date and Nature Methods. Our Press Office will contact you closer to the time of publication, but if you or your Press Office have any inquiries in the meantime, please contact press@nature.com.

Nature Portfolio journals [encourage authors to share their step-by-step experimental protocols](https://www.nature.com/nature-research/editorial-policies/reporting-standards#protocols) on a protocol sharing platform of their choice. Nature Portfolio 's Protocol Exchange is a free-to-use and open resource for protocols; protocols deposited in Protocol Exchange are citable and can be linked from the published article. More details can found at www.nature.com/protocolexchange/about.

Please note that you and any of your coauthors will be able to order reprints and single copies of the issue containing your article through Nature Portfolio 's reprint website, which is located at <http://www.nature.com/reprints/author-reprints.html>. If there are any questions about reprints please send an email to author-reprints@nature.com and someone will assist you.

Please feel free to contact me if you have questions about any of these points. Thank you very much again for publishing your paper at Nature Methods!

Best regards,

Lin

Lin Tang, PhD
Senior Editor
Nature Methods